# Beyond Independence: Learning Correlated Views for Variational Incomplete Multi-View Clustering

**Zheming Xu** [1][2]  **Aiyue Tang** [1][2]  **Shidi Chen** [1][2]  **Xuechao Zou** [1][2]  **Congyan Lang** [1][2]  **Rogelio A. Mancisidor** [3]
**Michael Kampffmeyer** [4][5]

## Abstract

Incomplete multi-view clustering (IMVC) aims to uncover shared cluster structures from data with partially observed views. Although recent imputation-free methods based on variational inference demonstrate robustness to missing views, they commonly rely on a conditional independence assumption across views in the posterior aggregation stage, which fails to capture the inherently structured and potentially correlated nature of multi-view data. In this paper, we propose a variational framework that explicitly goes beyond this assumption by introducing a learnable cross-view correlation structure. Specifically, we explicitly model and learn correlations between views by utilizing the covariance structure of posterior estimation errors during aggregation. To facilitate robust and efficient learning, the correlation matrix is parameterized through a normalized Cholesky decomposition, ensuring positive definiteness and enabling the entire model to be trained jointly through a unified variational objective. Extensive experiments on multiple IMVC benchmarks demonstrate that our method consistently outperforms state-of-the-art approaches across diverse missing-view settings while introducing only a negligible number of learnable parameters. These results highlight the effectiveness of adaptive correlation modeling in variational IMVC, demonstrating the need to go beyond the independence assumption in IMVC. The code is available at https://github.com/zmxu196/ACOVA.

[1]School of Computer Science & Technology, Beijing Jiaotong University [2]Key Laboratory of Big Data & Artificial Intelligence in Transportation, Ministry of Education, China [3]Department of Data Science, BI Norwegian Business School, Oslo, Norway [4]Department of Physics and Technology, UiT The Arctic University of Norway [5]Norwegian Computing Center. Correspondence to: Congyan Lang <cylang@bjtu.edu.cn>.

*Proceedings of the 43rd International Conference on Machine Learning*, Seoul, South Korea. PMLR 306, 2026. Copyright 2026 by the author(s).

## 1. Introduction

Multi-view clustering (MVC) (Yu et al., 2025) aims to partition data objects that are characterized by multiple distinct feature sets or views into their natural groups. However, in real-world scenarios, missing views are prevalent due to factors such as sensor failures, data corruption, or privacy concerns, giving rise to the problem of incomplete multi-view clustering (IMVC) (Chao et al., 2025; Zhao et al., 2025b). Among existing approaches, imputation-based IMVC methods have gained significant attention by recovering missing data or latent features (Wang et al., 2022b; Lin et al., 2023; Wen et al., 2024; Gao & Pu, 2025; Huang et al., 2025; Zhang et al., 2025) before clustering. Nevertheless, the main challenge in imputation-based methods lies in the accurate recovery of missing data without ground-truth supervision, especially when the proportion of missing views is large.

To avoid this challenge, imputation-free IMVC methods have emerged to directly learn from the available data. These include graph-based methods (Wen et al., 2020b; 2023), cross-view mapping methods (Xu et al., 2022; Zhao et al., 2025a), and distribution alignment methods (Xu et al., 2023). Among them, DVIMC (Xu et al., 2024) pioneered the application of Product of Experts (PoE) for IMVC, standing out for its effectiveness in learning robust shared latent representations in the missing views settings. However, these methods treat each view-specific posterior as independent, which is an assumption that oversimplifies the rich dependencies that naturally exist among views. To address this issue, a straightforward way is to model encoder-level cross-view correlations directly. However, this would move beyond the mean-field variational family typically used in VAE-based IMVC methods, substantially complicating optimization. Effective fusion requires that views compensate for missing or unreliable information in one another, but also that the model account for correlations across views, since highly correlated views are prone to making similar errors. A recent supervised approach, CoDE (Mancisidor et al., 2025), revisits the classic consensus-of-experts formulation (Winkler, 1981) and shows that explicitly modeling inter-expert

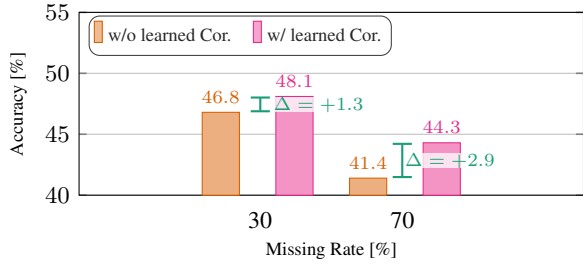

*Figure 1.* Benefit of modeling inter-view dependencies for different missing rates on Scene15.

correlations through their estimation errors is highly beneficial and provides a principled Bayesian approach. This perspective enables the model to capture cross-expert dependencies without explicitly parameterizing the relationships between the experts themselves, which is non-trivial as the joint posterior being approximated is a distribution of a latent variable. However, CoDE relies on a fixed scalar correlation obtained through a supervised grid search, rendering it impractical for IMVC, where labels are unavailable and views are incomplete. Moreover, its fixed correlation value fails to capture the diverse, pairwise dependencies across views, thereby compromising performance.

To tackle the aforementioned problems, we propose an **A**daptive **CO**rrelation-aware **V**ariational **A**ggregation framework for IMVC (ACOVA), which models inter-view dependencies via learning the estimation error covariance in the posterior aggregation stage. Specifically, we go beyond the strong independence assumption and extend the existing variational framework by (Xu et al., 2024) to a more realistic setting, in which the view-specific estimation errors are assumed to be correlated, allowing better capturing of the inherent dependencies between views. Furthermore, instead of relying on a supervised grid search to tune a global scalar correlation, which is infeasible in IMVC, we parameterize the correlation matrix via a normalized Cholesky decomposition, allowing it to be learned jointly with the model parameters through the overall training objective.

In summary, our major contributions are: (1) We introduce ACOVA, the first correlation-aware variational framework for incomplete multi-view clustering (IMVC) that explicitly models inter-view dependencies via the estimation error covariance, thereby relaxing the conditional-independence assumption underlying posterior aggregation in existing IMVC methods. (2) We propose an adaptive cross-view correlation learning mechanism that jointly estimates the correlation structure and model parameters within a unified variational objective, enabling ACOVA to dynamically capture view interactions and substantially improve clustering performance (Fig. 1). (3) We conduct extensive empirical evaluations on multiple IMVC benchmarks, where ACOVA

consistently achieves state-of-the-art or competitive performance across diverse missing-view settings with only a negligible increase in learnable parameters, demonstrating the importance of correlation-aware posterior modeling.

## 2. Related Work

**Incomplete Multi-View Clustering.** The development of deep neural networks has led to significant progress in deep IMVC (Wen et al., 2020a; Li et al., 2021; 2022; Wang et al., 2022a; Liu et al., 2024; Yu et al., 2024), resulting in two main types of methods: imputation-based and imputation-free approaches. Imputation-based methods aim to complete missing data through various strategies before performing clustering on the reconstructed multi-view dataset. Lin et al. (2021; 2023) utilize contrastive prediction combined with entropy-based optimization to recover missing views, while other methods adopt prototype-based alignment and imputation (Yuan et al., 2025) or apply diffusion models to generate missing samples (Wen et al., 2024; Zhang et al., 2025). However, these methods have the underlying issue that inaccurate or biased imputations will negatively affect the subsequent representation learning and clustering step. More recently, imputation-free methods (Jin et al., 2023; Wen et al., 2023) have therefore started to emerge, which avoid explicit missing data recovery and instead learn from available view-specific representations directly for downstream clustering. For example, CDIMC (Wen et al., 2020b) leverages graph embeddings and a self-paced learning strategy to enhance shared representations, while DIMVC (Xu et al., 2022) employs nonlinear mappings across views to exploit complementary information. APADC (Xu et al., 2023) emphasizes aligning the distributions of available data during feature learning, and DVIMC (Xu et al., 2024) incorporates variational autoencoders with coherence constraints to improve learning of a common representation.

**Variational Approaches for Incomplete Multi-View Clustering.** Variational Autoencoders (VAEs) (Kingma & Welling, 2014) provide an effective framework for learning latent distributions by incorporating diverse prior distributions and generative assumptions, and have been extensively applied to clustering (Jiang et al., 2017; Yang et al., 2019; Lim et al., 2020; Falck et al., 2021). Within multi-view clustering, numerous studies (Yin et al., 2020; Cui et al., 2023; Huang et al., 2023) have extended VAEs with principled approaches to aggregate information across views and derive joint representations. For instance, Multi-VAE (Xu et al., 2021) disentangles shared and view-specific features to improve clustering performance. However, most existing approaches are not designed for incomplete multi-view scenarios, limiting their practical applications. To address this challenge, a recent method DVIMC (Xu et al., 2024) employs a product-of-experts

strategy but assumes independence across views, limiting its ability to capture inter-view relationships. MVP (Gao & Pu, 2025) utilizes cyclic permutations within VAEs but relies on handcrafted priors and complex partitioning schemes that limit scalability. In contrast, our method adaptively learns a variational posterior with a cross-view correlation structure, enabling more expressive modeling of inter-view dependencies without explicit permutations.

## 3. Preliminaries and Motivation

In the following, we briefly review the preliminaries of variational IMVC. For completeness, we include derivations for these preliminaries in Appendix A.1. Building on these, we then delve into the core motivation of this work.

**Notations** Let $\{\mathcal{X}\}$ be an incomplete multi-view dataset, where we observe $\{\mathbf{x}_v\}_{v \in \mathcal{V}}$, and $\mathcal{V} \subseteq [V]$ denotes the set of available views. Here, $[V] = \{1, 2, \ldots, V\}$ represents the complete view set. Each $\mathbf{x}_v \in \mathbb{R}^{D_v}$ lies in a view-specific space. For brevity, we denote the observed view set of a generic instance by $\mathcal{V}$, with $\{\mathbf{x}_v\}_{v \in \mathcal{V}}$ representing the available multi-view inputs. In the missing-view setting, $\mathbf{x}_v$ denotes the data available for view $v$. The goal of IMVC is to partition the $N$ instances into $C$ clusters. For notation clarity, we use $\mathbf{M}^{(i)} \in \{0, 1\}^V$ for the per-sample view-availability mask and $\mathcal{M} \in \{0, 1\}^{N \times V}$ for the dataset-level mask matrix. When used with $\boldsymbol{\mu} \in \mathbb{R}^{VD \times 1}$ and $\boldsymbol{\Sigma} \in \mathbb{R}^{VD \times VD}$, $\mathbf{M}^{(i)}$ is repeated across latent dimensions to form an adapted mask in $\{0, 1\}^{VD \times 1}$.

**Generative Process** We assume that the multi-view data is generated from a random process involving latent variables $\mathbf{z} \in \mathbb{R}^D$ and discrete latent variables $\mathbf{c} \in \{1, \ldots, C\}$ that denote the common representation and cluster assignment, respectively. In the incomplete setting, assuming that the observed data are conditionally independent given the shared latent representation $\mathbf{z}$, the joint probability of the observed data and latent variables can be formulated as (Jiang et al., 2017; Yin et al., 2020; Xu et al., 2022):

$$p(\{\mathbf{x}_v\}_{v=1}^V, \mathbf{z}, \mathbf{c}) = p(\mathbf{c})\, p(\mathbf{z} \mid \mathbf{c})\, p(\{\mathbf{x}_v\}_{v=1}^V \mid \mathbf{z})$$
$$= p(\mathbf{c})\, p(\mathbf{z} \mid \mathbf{c}) \prod_{v=1}^V p(\mathbf{x}_v \mid \mathbf{z}). \quad (1)$$

**Inference Objective** Our goal is to infer the joint posterior distribution $p(\mathbf{z}, \mathbf{c} | \{\mathbf{x}_v\}_{v \in \mathcal{V}})$, which can be expressed through Bayes' theorem as:

$$p(\mathbf{z}, \mathbf{c} | \{\mathbf{x}_v\}_{v \in \mathcal{V}}) = \frac{p(\{\mathbf{x}_v\}_{v \in \mathcal{V}} | \mathbf{z}) p(\mathbf{z}, \mathbf{c})}{\int_{\mathbf{z}} \sum_{\mathbf{c}=1}^C p(\{\mathbf{x}_v\}_{v \in \mathcal{V}} | \mathbf{z}, \mathbf{c}) p(\mathbf{z}, \mathbf{c}) d\mathbf{z}}. \quad (2)$$

Due to the intractability of this posterior, variational inference introduces an approximate posterior $q_\phi(\mathbf{z}, \mathbf{c} | \{\mathbf{x}_v\}_{v \in \mathcal{V}})$ by minimizing the Kullback-Leibler (KL) divergence:

$$D_{\mathrm{KL}}(q_\phi(\mathbf{z}, \mathbf{c} | \{\mathbf{x}_v\}_{v \in \mathcal{V}}) || p(\mathbf{z}, \mathbf{c} | \{\mathbf{x}_v\}_{v \in \mathcal{V}})). \quad (3)$$

It is common to assume $\mathbf{z}$ and $\mathbf{c}$ are independent given $\{\mathbf{x}_v\}_{v \in \mathcal{V}}$, which allows to factorize the joint distribution as $q_\phi(\mathbf{z}, \mathbf{c} | \{\mathbf{x}_v\}_{v \in \mathcal{V}}) = q_\phi(\mathbf{z} | \{\mathbf{x}_v\}_{v \in \mathcal{V}}) q_\phi(\mathbf{c} | \{\mathbf{x}_v\}_{v \in \mathcal{V}})$.

**Joint Posterior Aggregation** With the VAE paradigm, we approximate the posterior $q_\phi(\mathbf{z} | \mathbf{x}_v) = \mathcal{N}(\mathbf{z} | \boldsymbol{\mu}_v, \boldsymbol{\Sigma}_v)$ of each view using a Gaussian distribution[1], where $\boldsymbol{\mu}_v$ and $\boldsymbol{\sigma}_v^2$ are the mean and variance parameters inferred by the $v$-th view-specific encoders. Based on this, the approximate posterior $q_\phi(\mathbf{z} | \{\mathbf{x}_v\}_{v \in \mathcal{V}})$ is typically obtained through an aggregation mechanism $\mathcal{A}$ that combines view-specific posteriors $q_\phi(\mathbf{z} | \mathbf{x}^{(v)})$:

$$\mathcal{A} : \{\boldsymbol{\mu}_v, \boldsymbol{\sigma}_v^2\}^{|\mathcal{V}|} \to \{\tilde{\boldsymbol{\mu}}, \tilde{\boldsymbol{\sigma}}^2\}. \quad (4)$$

**Motivation** Following this paradigm, recent multi-view clustering methods primarily differ in how they derive the joint posterior variational distribution $\mathcal{N}(\tilde{\boldsymbol{\mu}}, \tilde{\boldsymbol{\sigma}}^2)$. DMVC-VAE (Yin et al., 2020) emphasizes the varying importance of different views and proposes a weighted combination approach in the complete multi-view setting. A more relevant approach, DVIMC (Xu et al., 2024), tackles the incomplete multi-view setting using the Product of Experts (PoE) strategy, which treats each view as an independent expert and combines them through precision-weighted averaging. However, both methods rely on a strong independence assumption across views, which is often overly simplistic in real-world scenarios where views are semantically correlated. This limits the capacity of the model to capture cross-view interactions and express the true joint posterior. Besides, both approaches have drawbacks in the aggregation of the joint posterior. DMVCVAE (Yin et al., 2020) could be seen as a partial realization of the Mixture of Experts (MoE), where only the within-view variances are considered, omitting the variability arising from disagreements between views, leading to underestimated overall uncertainty. While in DVIMC (Xu et al., 2024), PoE is known to underestimate the variance of the consensus distribution due to its strong independence assumption. Let $\boldsymbol{\xi}_v = 1/\boldsymbol{\sigma}_v^2$ denote the precision of view $v$, the aggregated precision under PoE could then be denoted as $\tilde{\boldsymbol{\xi}}_{\mathrm{PoE}} = \sum_v \boldsymbol{\xi}_v$. As a result, the aggregated variance becomes $\tilde{\boldsymbol{\sigma}}^2 = 1/\tilde{\boldsymbol{\xi}}_{\mathrm{PoE}}$, which is always smaller than any individual $\boldsymbol{\sigma}_v^2$. This issue becomes especially concerning in the incomplete multi-view scenario, where limited observations naturally lead to higher uncertainty, requiring more cautious and calibrated posterior variance estimation.

## 4. Adaptive Correlation-aware Variational Aggregation

As discussed previously, IMVC requires posterior inference over samples with arbitrarily missing views. A central

---

[1]This research assumes a diagonal covariance matrix for all variational distributions, i.e. $\boldsymbol{\Sigma} = \boldsymbol{\sigma}^2 \mathbf{I}$.

challenge lies in how to aggregate view-specific posterior distributions to form a shared latent representation, particularly when the common independence assumption is overly simplistic and leads to underestimation of the posterior variance. Inspired by (Mancisidor et al., 2025; Winkler, 1981), we model the inter-view dependence through the error of estimation associated with each view's posterior distribution. However, directly applying these methods to IMVC poses significant limitations. First, they barely learn the covariance matrix of the error of estimation but cross-validate the implicit correlation, which is infeasible in unsupervised clustering settings where labels are unavailable. Second, based on grid-searching, they assume a fixed correlation, which restricts the model's ability to capture diverse and pairwise specific dependencies across views. To address these problems, we propose a novel approach (Fig. 2) where the correlation structure is learned jointly with the model parameters, enabling modeling of adaptive inter-view dependencies under missing-view scenarios without supervision.

## 4.1. Posterior with Adaptive Cross-view Correlation

Given the mean parameters $\{\boldsymbol{\mu}_v\}_{v\in\mathcal{V}}$ of the view-specific variational distributions, we define $\boldsymbol{\mu}^d = [\mu_1^d, \mu_2^d, \ldots, \mu_V^d]^\top \in \mathbb{R}^{V\times 1}$ as the vector of view-specific estimates for the $d$-th dimension of the latent variable $\mathbf{z}$, where $\mu_v^d$ represents the estimate of view $v$ for dimension $d$. The complete estimate vector is then constructed as $\boldsymbol{\mu} = [\boldsymbol{\mu}^{1\top}, \boldsymbol{\mu}^{2\top}, \ldots, \boldsymbol{\mu}^{D\top}]^\top \in \mathbb{R}^{VD\times 1}$ by concatenating all dimension-wise estimates. Given variance parameters $\{\sigma_v^2\}_{v\in\mathcal{V}}$, $\boldsymbol{\sigma}^2$ is defined similarly. To model the discrepancy between the estimate $\boldsymbol{\mu}$ and the true, but unknown, latent variable $\mathbf{z}$, Definition 4.1 introduces the error of estimation (Winkler, 1981; Mancisidor et al., 2025).

**Definition 4.1** (Error of Estimation). Given the estimate $\boldsymbol{\mu} = [\boldsymbol{\mu}^{1\top}, \boldsymbol{\mu}^{2\top}, \ldots, \boldsymbol{\mu}^{D\top}]^\top$ by the view-specific distributions $q_v(\mathbf{z}|\mathbf{x}_v) = \mathcal{N}(\boldsymbol{\mu}_v, \boldsymbol{\sigma}_v^2)$, the error of estimation of the latent variable $\boldsymbol{z}$ can be formulated as:

$$\boldsymbol{\epsilon} = \boldsymbol{\mu} - \mathbb{1}\mathbf{z}, \qquad (5)$$

where $\mathbb{1} \in \mathbb{R}^{VD\times D}$ is a block diagonal design matrix with structure $\mathbb{1} = \text{blkdiag}(\mathbf{1}_V, \mathbf{1}_V, \ldots, \mathbf{1}_V)$ containing $D$ blocks, each $\mathbf{1}_V \in \mathbb{R}^{V\times 1}$ being a column vector of ones, $\boldsymbol{\epsilon} \sim \mathcal{N}(\mathbf{0}, \boldsymbol{\Sigma})$, and $\boldsymbol{\Sigma} = \text{diag}[\boldsymbol{\Sigma}^1, \boldsymbol{\Sigma}^2, \cdots, \boldsymbol{\Sigma}^D] \in \mathbb{R}^{VD\times VD}$ consists of the error of estimation covariance matrices $\boldsymbol{\Sigma}^d \in \mathbb{R}^{V\times V}$ for each dimension $d$.

Based on the above definition and as shown in Fig. 2, each of the available view-specific distributions provides an assessment $\boldsymbol{\mu}^d$ of the joint latent variable $\mathbf{z}^d$. Therefore, the covariance matrix $\boldsymbol{\Sigma}^d$ reflects the interdependency between the error of estimation of the view-specific distributions. This dependency should not be ignored in previous aggregation methods, since all distributions are exposed to the

same underlying object. It is worth mentioning that (Mancisidor et al., 2025) learns the same dependency correlations across views, but relies on supervised cross-validation to grid-search the implicit correlation between the assessments of the view-specific distributions. However, this supervised approach is incompatible with unsupervised clustering tasks, where ground-truth labels are unavailable for tuning correlation parameters. Therefore, in this paper, we propose to adaptively learn the cross-view correlation structure through a novel differentiable parameterization:

**Definition 4.2** (Adaptive Cross-view Correlation Learning). Let $\mathbf{L}$ be a learnable lower-triangular matrix with positive diagonal elements ensured by exponentiation. We use the Cholesky decomposition of the covariance matrix $\boldsymbol{\Sigma}^d$ to guarantee the positive definiteness constraint and bounded entries of $\mathbf{R} = [\rho_{ij}]_{i,j=1}^V \succ 0$, a positive definite correlation matrix with $\rho_{ii} = 1$ and $\rho_{ij} \in [-1, 1]$ for $i \neq j$, as follows:

$$\mathbf{R}_{ij} = \frac{(\mathbf{L}\mathbf{L}^\top)_{ij}}{\sqrt{(\mathbf{L}\mathbf{L}^\top)_{ii}(\mathbf{L}\mathbf{L}^\top)_{jj}}}. \qquad (6)$$

We use the following decomposition of error covariance $\boldsymbol{\Sigma}^d$ for the $d$-th dimension

$$\boldsymbol{\Sigma}^d = \begin{bmatrix} \sigma_1^2 & \rho_{12}\sigma_1\sigma_2 & \cdots & \rho_{1V}\sigma_1\sigma_V \\ \rho_{21}\sigma_2\sigma_1 & \sigma_2^2 & \cdots & \rho_{2V}\sigma_2\sigma_V \\ \vdots & \vdots & \ddots & \vdots \\ \rho_{V1}\sigma_V\sigma_1 & \rho_{V2}\sigma_V\sigma_2 & \cdots & \sigma_V^2 \end{bmatrix} = \mathbf{DRD}, \qquad (7)$$

where $\mathbf{D} = \text{diag}(\sigma_1, \ldots, \sigma_V)$ contains the view-specific standard deviations, to learn pairwise dependencies between view-specific variational distributions.

Accordingly, by decomposing the error covariance $\boldsymbol{\Sigma}^d = \mathbf{DRD}$, we provide a general framework that unifies existing approaches through the separation of variance $\mathbf{D}$ and correlation matrix $\mathbf{R}$. When $\mathbf{R} = \mathbf{I}$, our method degenerates to DVIMC (Xu et al., 2024) with independence assumption; when $\mathbf{R} = \rho\mathbf{I}$ for a fixed scalar $\rho$, it degenerates to CoDE (Mancisidor et al., 2025) with a uniform correlation. Our approach generalizes beyond these limitations by modeling a full correlation matrix $\mathbf{R}$ that can adaptively model heterogeneous dependencies between views through a learnable Cholesky decomposition parameterization, enabling end-to-end optimization without manual tuning. Importantly, learning $\mathbf{R}$ models dependence among estimation errors across views after conditioning on the shared latent variable. It does not remove complementary information in $\mathbf{x}_v$ but calibrates uncertainty-aware fusion so that correlated errors are not treated as independent evidence, unlike PoE-style aggregation that assumes independence.

To understand $\mathbf{R}$'s capacity to deviate from independence-based aggregation, we provide a theoretical deviation bound from the identity matrix:

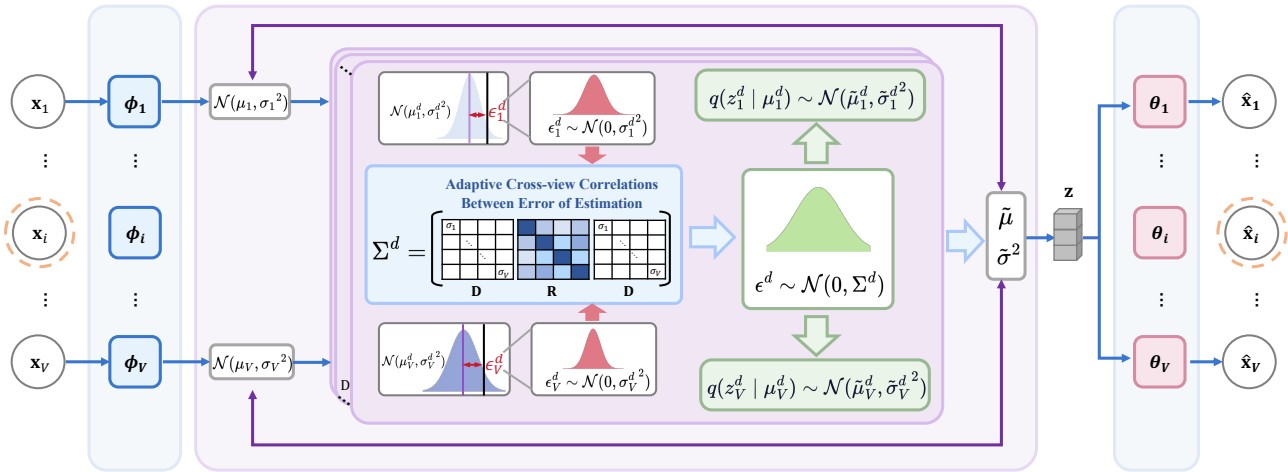

*Figure 2.* Overview of our proposed **ACOVA** framework. Given multi-view inputs $\{\mathbf{x}_v\}_{v=1}^V$, view-specific encoders $\{\boldsymbol{\phi}_v\}$ produce posterior distribution parameters $\{\boldsymbol{\mu}_v, \boldsymbol{\sigma}_v^2\}$ to characterize view-wise uncertainty. For each dimension $d \in [D]$, the estimation errors $\epsilon_v^d$ between the view-specific posteriors and the true latent variable are modeled with learnable cross-view correlations, captured by the correlation matrix $\mathbf{R}$. This structure, together with view-specific variances $\mathbf{D}$, forms a structured covariance matrix $\boldsymbol{\Sigma}^d = \mathbf{DRD}$, which governs the aggregation of posteriors into a consensus distribution with parameter values $\{\tilde{\mu}_v^d, \tilde{\sigma}_v^{d\,2}\}$. A latent representation $\mathbf{z} \in \mathbb{R}^D$ is sampled from this aggregated posterior and decoded via view-specific decoders $\{\boldsymbol{\theta}_v\}$ for reconstruction. Note that conditioning on $\boldsymbol{\mu}_v$ or on $\mathbf{x}_v$ is equivalent in this context.

**Theorem 4.3** (Frobenius Identity Deviation Bound). *Given* $\mathbf{R} = [\rho_{ij}]_{i,j=1}^V \succ 0$ *where* $\rho_{ii} = 1$ *and* $\rho_{ij} \in [-1, 1]$ *for* $i \neq j$, *the Frobenius deviation from identity matrix* $\mathbf{I}$ *is upper-bounded as*

$$\|\mathbf{R} - \mathbf{I}\|_F \leq \sqrt{V(V-1)}. \tag{8}$$

The proof of the bound can be found in Appendix A.2. The bound formally characterizes the maximal Frobenius deviation of a valid correlation matrix from the identity matrix, offering a theoretical guarantee on the extent of structural variability induced by cross-view dependency modeling. We further provide theoretical results demonstrating the identifiability of the learned correlations in Appendix A.3.

Based on the above analysis, denoting a binary mask vector as $\mathbf{M} \in \{0, 1\}^{VD \times 1}$ (adapted from the per-sample view indicator) to indicate the availability of view-specific observations, the joint posterior distribution for incomplete multi-view data can then be formulated as (see derivation in Appendix A.4):

$$q(\mathbf{z} \mid \{\mathbf{x}_v\}_{v \in \mathcal{V}}) \sim \mathcal{N}(\tilde{\boldsymbol{\mu}}, \tilde{\boldsymbol{\sigma}}^2) = \mathcal{N}(\mathbf{A}_\mathbf{M}^{-1}\mathbf{B}_\mathbf{M}, \mathbf{A}_\mathbf{M}^{-1}) \tag{9}$$

where the aggregation matrices are computed as:

$$\mathbf{A}_\mathbf{M} = \mathbb{1}^\top (\boldsymbol{\Sigma}^{-1} \odot \mathbf{M}\mathbf{M}^\top) \mathbb{1} \tag{10}$$

$$\mathbf{B}_\mathbf{M} = \mathbb{1}^\top (\boldsymbol{\Sigma}^{-1} \odot \mathbf{M}\mathbf{M}^\top)(\boldsymbol{\mu} \odot \mathbf{M}) \tag{11}$$

Here, the posterior aggregation is controlled by two matrices derived from the correlation structure and mask, where $\mathbf{A}_\mathbf{M}$ represents the precision matrix of the aggregated posterior, and $\mathbf{B}_\mathbf{M}$ contains the precision-weighted mean estimates.

The Hadamard product $\odot$ with the mask ensures that only available observations contribute to the aggregation process.[2]

### 4.2. ELBO for Incomplete Multi-view Clustering

The cluster assignment posterior $q_\phi(\mathbf{c}|\{\mathbf{x}_v\}_{v \in \mathcal{V}})$ can be approximated through the VaDE (Jiang et al., 2017) trick by $q_\phi(\mathbf{z}|\{\mathbf{x}_v\}_{v \in \mathcal{V}})$ via Bayes' rule:

$$q_\phi(\mathbf{c}|\{\mathbf{x}_v\}_{v \in \mathcal{V}}) = q(\mathbf{c} \mid \mathbf{z}) \equiv \frac{p(\mathbf{c})\, p(\mathbf{z} \mid \mathbf{c})}{\sum_\mathbf{c} p(\mathbf{c})\, p(\mathbf{z} \mid \mathbf{c})}, \tag{12}$$

where $p(\mathbf{c}) = \text{Cat}(\boldsymbol{\tau})$ and $p(\mathbf{z} \mid \mathbf{c}) \sim \mathcal{N}(\mathbf{z} \mid \boldsymbol{\mu}_c, \boldsymbol{\Sigma}_c)$ being a Gaussian mixture prior. Here, $\text{Cat}(\boldsymbol{\tau})$ refers to the categorical distribution with $\boldsymbol{\tau}_c$ ($\sum \boldsymbol{\tau}_c = 1$) being the prior probability for cluster $c$. Then the evidence lower bound (ELBO) derived from Eq. 3 for one sample can be expressed as[3]:

$$\mathcal{L}_{\text{ELBO}}(\{\mathbf{x}_v\}_{v \in \mathcal{V}}) = \mathbb{E}_{q_\phi(\mathbf{z}|\{\mathbf{x}_v\}_{v \in \mathcal{V}})} \left[ \sum_{v \in \mathcal{V}} \log p(\mathbf{x}_v \mid \mathbf{z}) \right]$$
$$- \mathbb{E}_{q_\phi(\mathbf{c}|\{\mathbf{x}_v\}_{v \in \mathcal{V}})} [D_{\text{KL}}(q_\phi(\mathbf{z} \mid \{\mathbf{x}_v\}_{v \in \mathcal{V}}) \,\|\, p(\mathbf{z} \mid \mathbf{c}))]$$
$$- D_{\text{KL}}(q_\phi(\mathbf{c} \mid \{\mathbf{x}_v\}_{v \in \mathcal{V}}) \,\|\, p(\mathbf{c})). \tag{13}$$

Note that the first reconstruction term in Eq. 13 ensures that the latent variables sampled from the joint posterior distribution are capable of generating all observed views. Therefore,

---

[2]Note, explicit inversion of $\boldsymbol{\Sigma}$ can be avoided if desired (see discussion in Appendix A.6).

[3]The full derivation is included in the Appendix A.5.

generative models encourage learning latent variables that also embed view-specific information, eliminating the need to factorize the latent space.

### 4.3. Training Objective

Following (Xu et al., 2024), we include a distribution alignment loss to encourage consistency between the aggregated posterior and the individual view-specific posteriors:

$$\mathcal{L}_{\text{align}} = -\frac{1}{|\mathcal{V}|} \sum_{v \in \mathcal{V}} D_{\text{KL}} \left( q_\phi \left( \mathbf{z} \mid \{\mathbf{x}_v\}_{v=1}^V \right) \big\| q_{\phi_v} \left( \mathbf{z} \mid \mathbf{x}_v \right) \right).$$

$$(14)$$

This alignment regularizer prevents the aggregated posterior from ignoring information encoded by each view-specific posterior $q_{\phi_v}(\mathbf{z} \mid \mathbf{x}_v)$, thereby retaining view-wise complementary cues in the fused latent space.

Overall, with balancing hyperparameter $\alpha$, the total training objective is [4]:

$$\mathcal{L}_{\text{total}} = \frac{1}{N} \sum_{i=1}^N \left( \mathcal{L}_{\text{ELBO}}^{(i)} + \alpha \mathcal{L}_{\text{align}}^{(i)} \right).$$

$$(15)$$

**Optimization of Correlation Matrix R.** In Eq. 15, both of the KL divergences terms $D_{\text{KL}} \left( q_\phi(\mathbf{z} \mid \{\mathbf{x}_v\}_{v \in \mathcal{V}}) \| p(\mathbf{z} \mid \mathbf{c}) \right)$ and $\sum_{v \in \mathcal{V}} D_{\text{KL}} \left( q_\phi \left( \mathbf{z} \mid \{\mathbf{x}_v\}_{v=1}^V \right) \big\| q_{\phi_v} \left( \mathbf{z} \mid \mathbf{x}_v \right) \right)$ can be seen as functions of $\mathbf{R}$. Assuming Gaussian distributions, their combination yields:

$$-\frac{1}{2} \Bigg\{ \underbrace{\text{tr} \left( \mathbf{\Sigma}_c^{-1} \mathbf{\Omega} \right) + \|\boldsymbol{\mu}_c - \boldsymbol{\eta}\|_{\mathbf{\Sigma}_c^{-1}}^2 + \ln |\mathbf{\Sigma}_c| - \ln |\mathbf{\Omega}|}_{\text{Cross-view Correlation Learning}}$$

$$+ \alpha \sum_{v \in \mathcal{V}} \underbrace{\left[ \text{tr} \left( \mathbf{\Sigma}_v^{-1} \mathbf{\Omega} \right) + \|\boldsymbol{\mu}_v - \boldsymbol{\eta}\|_{\mathbf{\Sigma}_v^{-1}}^2 + \ln |\mathbf{\Sigma}_v| - \ln |\mathbf{\Omega}| \right]}_{\text{View-specific Consistency Alignment}} \Bigg\},$$

$$(16)$$

which is the term being optimized in an end-to-end manner to learn the correlation matrix $\mathbf{R}$. For notational brevity, $\mathbf{\Omega} \triangleq \mathbf{A}_{\mathbf{M}}^{-1}$, $\boldsymbol{\eta} \triangleq \mathbf{\Omega} \mathbf{B}_{\mathbf{M}}$, and $\|\mathbf{x}\|_{\mathbf{S}}^2 \triangleq \mathbf{x}^\top \mathbf{S} \mathbf{x}$.

## 5. Experiment

### 5.1. Experimental Setup

**Datasets.** The experiments are carried out on the following six standard IMVC datasets. **Handwritten** (LeCun et al., 1989) contains 2,000 digit samples ranging from '0' to '9', where each sample is described using six types of features extracted from handwritten numerals. **Caltech5V** (Fei-Fei et al., 2004) consists of 1,400 images across 7 object categories, with each image represented by five types of

---

[4]Pseudo-code, complexity and time-cost analysis are discussed in Appendix A.7, A.8 and C.4, respectively.

visual features. **Scene15** (Fei-Fei & Perona, 2005) includes 4,485 images covering 15 scene categories. Each image is represented using three different visual features. **Fashion** (Xiao et al., 2017) is constructed from the Fashion-MNIST dataset, containing 10,000 images from 10 clothing categories. Each sample is represented from three different views. **NoisyMNIST** (Wang et al., 2015) is a large-scale dataset that consists of 2 views, 10 classes and 70,000 samples. **CUB Image-Caption** (Zhang et al., 2019) is a multi-modal dataset consisting of 600 images with corresponding text descriptions of 10 bird classes. Overall, the six datasets offer complementary coverage across scales, view counts, and both feature-based and raw-pixel views, as well as multi-modal inputs, facilitating a comprehensive evaluation under incomplete multi-view clustering.

**Incomplete multi-view data processing.** Following previous works (Xu et al., 2024), we construct the incomplete multi-view datasets by randomly selecting $p\%$ ($p = \{10, 30, 50, 70\}$) of the samples and randomly deleting views [5], under the constraint that every sample retains at least one available view. To ensure fair comparisons, results are averaged over five runs, with the same five masks being used across methods.

**Baselines.** We compare our ACOVA with eight recent state-of-the-art methods, namely BSV (Zhao et al., 2016), CONCAT (Zhao et al., 2016), DCP (Lin et al., 2023), DSIMVC (Tang & Liu, 2022), MVP (Gao & Pu, 2025), CPSPAN (Jin et al., 2023), DVIMC (Xu et al., 2024), and PMIMC (Yuan et al., 2025).

**Evaluation Measures.** We evaluate clustering performance using the four standard metrics for IMVC (Amigó et al., 2009): clustering accuracy (ACC), normalized mutual information (NMI), adjusted Rand index (ARI), and purity (PUR). Higher values indicate better clustering performance for all four metrics.

Additional setup details can be found in Appendix B.

### 5.2. Comparison with State-of-the-Art Methods

As shown in Table 1, we compare ACOVA with eight state-of-the-art IMVC methods across six real-world datasets. It can be observed that: (1) Compared to the rest of the methods, ACOVA consistently ranks first or second across all datasets and missing rate settings, validating the effectiveness of our adaptive correlation learning strategy. (2) Compared to existing variational IMVC methods, such as DVIMC, which builds upon a strong independence assumption, and MVP, which leverages cyclic permutation to impute latent variables, ACOVA achieves superior per-

---

[5]Extreme scenarios when $p = \{0, 90\}$ are discussed in Appendix C.2.

*Table 1.* Clustering performance under various missing-view rates on all datasets. Best results and second best are highlighted in **bold** and underlined, respectively. Note that all results are averaged over five runs.

| | Missing Rate | 10% | | | | 30% | | | | 50% | | | | 70% | | | |
|---|---|---|---|---|---|---|---|---|---|---|---|---|---|---|---|---|---|
| | Metrics | ACC↑ | NMI↑ | ARI↑ | PUR↑ | ACC↑ | NMI↑ | ARI↑ | PUR↑ | ACC↑ | NMI↑ | ARI↑ | PUR↑ | ACC↑ | NMI↑ | ARI↑ | PUR↑ |
| Scene15 | BSV | 0.3595 | 0.3577 | 0.1784 | 0.4002 | 0.3305 | 0.3206 | 0.1253 | 0.3630 | 0.2944 | 0.2905 | 0.0796 | 0.3321 | 0.2593 | 0.2568 | 0.0472 | 0.2903 |
| | Concat | 0.3703 | 0.3820 | 0.2037 | 0.4263 | 0.3360 | 0.3318 | 0.1540 | 0.3825 | 0.2943 | 0.2995 | 0.1201 | 0.3436 | 0.2596 | 0.2665 | 0.0982 | 0.3053 |
| | DCP | 0.4071 | 0.4414 | 0.2488 | 0.4342 | 0.3958 | 0.4264 | 0.2350 | 0.4265 | 0.3879 | 0.4095 | 0.2359 | 0.4166 | 0.3784 | 0.3849 | 0.2113 | 0.4026 |
| | CPSPAN | 0.3574 | 0.3457 | 0.1911 | 0.4103 | 0.3161 | 0.2783 | 0.1541 | 0.3439 | 0.2553 | 0.2070 | 0.1063 | 0.2786 | 0.2025 | 0.1423 | 0.0621 | 0.2198 |
| | DSIMVC | 0.2795 | 0.2976 | 0.1430 | 0.3218 | 0.2734 | 0.2896 | 0.1383 | 0.3131 | 0.2697 | 0.2828 | 0.1335 | 0.3133 | 0.2654 | 0.2741 | 0.1290 | 0.3062 |
| | DVIMC | 0.4863 | 0.4780 | 0.3192 | 0.5021 | 0.4678 | 0.4440 | 0.3097 | 0.4823 | 0.4371 | 0.4163 | 0.2737 | 0.4577 | 0.4136 | 0.3950 | 0.2519 | 0.4286 |
| | MVP | 0.4451 | 0.4361 | 0.2696 | 0.4830 | 0.4458 | 0.4341 | 0.2723 | 0.4866 | 0.4564 | 0.4314 | 0.2967 | 0.5170 | 0.4407 | 0.4285 | 0.2725 | 0.4895 |
| | PMIMC | 0.3317 | 0.3409 | 0.1778 | 0.3602 | 0.3071 | 0.3175 | 0.1492 | 0.3509 | 0.3125 | 0.3180 | 0.1477 | 0.3513 | 0.3178 | 0.3173 | 0.1492 | 0.3509 |
| | ACOVA (Ours) | 0.5040 | 0.4804 | 0.3273 | 0.5270 | 0.4807 | 0.4560 | 0.3106 | 0.5119 | 0.4710 | 0.4360 | 0.2973 | 0.4987 | 0.4428 | 0.4083 | 0.2733 | 0.4636 |
| Caltech5V | BSV | 0.5678 | 0.4364 | 0.3408 | 0.5827 | 0.5035 | 0.3699 | 0.2308 | 0.5170 | 0.4483 | 0.3235 | 0.1488 | 0.4617 | 0.3859 | 0.2798 | 0.0890 | 0.4019 |
| | Concat | 0.4750 | 0.3300 | 0.2519 | 0.4957 | 0.4450 | 0.2941 | 0.1998 | 0.4593 | 0.4279 | 0.2756 | 0.1595 | 0.4386 | 0.3600 | 0.2338 | 0.0891 | 0.3757 |
| | DCP | 0.5754 | 0.5719 | 0.4612 | 0.6054 | 0.5363 | 0.5269 | 0.4199 | 0.5778 | 0.5063 | 0.4729 | 0.3406 | 0.5393 | 0.3982 | 0.3363 | 0.1378 | 0.4056 |
| | CPSPAN | 0.7689 | 0.6677 | 0.6088 | 0.7750 | 0.6874 | 0.5653 | 0.4666 | 0.6888 | 0.6010 | 0.4789 | 0.3467 | 0.6030 | 0.4861 | 0.3664 | 0.2222 | 0.4904 |
| | DSIMVC | 0.7760 | 0.6998 | 0.6280 | 0.7816 | 0.7841 | 0.6947 | 0.6335 | 0.7841 | 0.7339 | 0.6868 | 0.5989 | 0.7413 | 0.7029 | 0.5838 | 0.5083 | 0.7047 |
| | DVIMC | 0.9013 | 0.8207 | 0.7999 | 0.9013 | 0.8921 | 0.8011 | 0.7850 | 0.8921 | 0.8609 | 0.7507 | 0.7322 | 0.8609 | 0.8514 | 0.7345 | 0.7195 | 0.8514 |
| | MVP | 0.8083 | 0.6969 | 0.6594 | 0.8083 | 0.8070 | 0.7139 | 0.6723 | 0.8146 | 0.8103 | 0.7063 | 0.6700 | 0.8157 | 0.7677 | 0.6483 | 0.5934 | 0.7693 |
| | PMIMC | 0.8963 | 0.8205 | 0.7966 | 0.8963 | 0.8917 | 0.8124 | 0.7858 | 0.8917 | 0.8601 | 0.7591 | 0.7381 | 0.8601 | 0.8570 | 0.7522 | 0.7304 | 0.8570 |
| | ACOVA (Ours) | 0.9149 | 0.8507 | 0.8287 | 0.9149 | 0.9131 | 0.8435 | 0.8226 | 0.9131 | 0.8953 | 0.8078 | 0.7918 | 0.8953 | 0.8660 | 0.7531 | 0.7340 | 0.8660 |
| Handwritten | BSV | 0.7087 | 0.6647 | 0.5625 | 0.7433 | 0.7105 | 0.6344 | 0.4716 | 0.7171 | 0.5842 | 0.5444 | 0.3028 | 0.5970 | 0.5159 | 0.4802 | 0.1874 | 0.5208 |
| | Concat | 0.7684 | 0.7214 | 0.6412 | 0.7837 | 0.7147 | 0.6441 | 0.5012 | 0.7190 | 0.6108 | 0.5437 | 0.3492 | 0.6166 | 0.5021 | 0.4498 | 0.2511 | 0.5105 |
| | DCP | 0.6459 | 0.7094 | 0.5663 | 0.6737 | 0.6639 | 0.6951 | 0.5189 | 0.6877 | 0.7045 | 0.7018 | 0.5651 | 0.7334 | 0.6913 | 0.6398 | 0.4856 | 0.7083 |
| | CPSPAN | 0.8710 | 0.7969 | 0.7586 | 0.8727 | 0.7524 | 0.6927 | 0.5603 | 0.7638 | 0.6430 | 0.6050 | 0.3862 | 0.6595 | 0.6470 | 0.5828 | 0.3289 | 0.6486 |
| | DSIMVC | 0.8014 | 0.7960 | 0.7282 | 0.8158 | 0.8180 | 0.8082 | 0.7455 | 0.8275 | 0.8304 | 0.8122 | 0.7543 | 0.8342 | 0.8284 | 0.7879 | 0.7312 | 0.8284 |
| | DVIMC | 0.9169 | 0.8604 | 0.8547 | 0.9201 | 0.8954 | 0.8470 | 0.8210 | 0.9020 | 0.8980 | 0.8373 | 0.8175 | 0.9054 | 0.7808 | 0.7905 | 0.7197 | 0.8099 |
| | MVP | 0.7833 | 0.8002 | 0.7114 | 0.8119 | 0.8095 | 0.8000 | 0.7270 | 0.8208 | 0.7474 | 0.7853 | 0.6834 | 0.7715 | 0.7653 | 0.7863 | 0.6956 | 0.7848 |
| | PMIMC | 0.8826 | 0.8188 | 0.7846 | 0.8834 | 0.8998 | 0.8213 | 0.7962 | 0.8998 | 0.8613 | 0.8096 | 0.7637 | 0.8695 | 0.8522 | 0.7994 | 0.7505 | 0.8587 |
| | ACOVA (Ours) | 0.9363 | 0.8692 | 0.8645 | 0.9363 | 0.9376 | 0.8709 | 0.8669 | 0.9376 | 0.9146 | 0.8489 | 0.8325 | 0.9146 | 0.8999 | 0.8351 | 0.8176 | 0.9021 |
| Fashion | BSV | 0.4832 | 0.4606 | 0.2986 | 0.5210 | 0.4424 | 0.4114 | 0.2210 | 0.4779 | 0.4100 | 0.3734 | 0.1626 | 0.4361 | 0.3645 | 0.3348 | 0.1040 | 0.3916 |
| | Concat | 0.6546 | 0.6518 | 0.5451 | 0.7028 | 0.5002 | 0.4618 | 0.3646 | 0.5272 | 0.4146 | 0.3433 | 0.2509 | 0.4350 | 0.2988 | 0.1976 | 0.1307 | 0.3247 |
| | DCP | 0.8547 | 0.8864 | 0.8119 | 0.8652 | 0.7663 | 0.8340 | 0.7285 | 0.7899 | 0.7533 | 0.7995 | 0.6904 | 0.7643 | 0.6717 | 0.7430 | 0.6079 | 0.6879 |
| | CPSPAN | 0.7130 | 0.7566 | 0.6303 | 0.7548 | 0.6981 | 0.7524 | 0.6179 | 0.7446 | 0.6886 | 0.7527 | 0.6237 | 0.7436 | 0.6672 | 0.7416 | 0.6043 | 0.7273 |
| | DSIMVC | 0.8993 | 0.8573 | 0.8178 | 0.8993 | 0.8810 | 0.8772 | 0.8162 | 0.8810 | 0.8342 | 0.8076 | 0.7380 | 0.8342 | 0.8048 | 0.7733 | 0.6938 | 0.8056 |
| | DVIMC | 0.8856 | 0.8799 | 0.8284 | 0.9024 | 0.8806 | 0.8718 | 0.8236 | 0.8986 | 0.8629 | 0.8449 | 0.7965 | 0.8867 | 0.8579 | 0.8290 | 0.7729 | 0.8725 |
| | MVP | 0.8691 | 0.8743 | 0.8175 | 0.8955 | 0.8259 | 0.8537 | 0.7744 | 0.8722 | 0.7982 | 0.8372 | 0.7484 | 0.8485 | 0.8386 | 0.8325 | 0.7758 | 0.8665 |
| | PMIMC | 0.6986 | 0.7563 | 0.6282 | 0.7427 | 0.7039 | 0.7586 | 0.6365 | 0.7540 | 0.7118 | 0.7586 | 0.6295 | 0.7577 | 0.7059 | 0.7583 | 0.6362 | 0.7540 |
| | ACOVA (Ours) | 0.9056 | 0.8876 | 0.8516 | 0.9129 | 0.8848 | 0.8735 | 0.8266 | 0.9009 | 0.8793 | 0.8479 | 0.8002 | 0.8794 | 0.8722 | 0.8312 | 0.7833 | 0.8745 |
| NoisyMNIST | BSV | 0.5144 | 0.4542 | 0.3258 | 0.5618 | 0.4184 | 0.3343 | 0.1896 | 0.4456 | 0.2939 | 0.1912 | 0.0754 | 0.3090 | 0.2672 | 0.1566 | 0.0452 | 0.2693 |
| | Concat | 0.4338 | 0.3910 | 0.2727 | 0.4418 | 0.3873 | 0.3229 | 0.1891 | 0.4036 | 0.3463 | 0.2592 | 0.1252 | 0.3561 | 0.2805 | 0.1832 | 0.0615 | 0.2864 |
| | DCP | 0.8945 | 0.9149 | 0.8786 | 0.9319 | 0.8997 | 0.8952 | 0.8608 | 0.9198 | 0.9150 | 0.8586 | 0.8401 | 0.9150 | 0.9225 | 0.8335 | 0.8431 | 0.9225 |
| | CPSPAN | 0.5665 | 0.5870 | 0.4402 | 0.6058 | 0.6103 | 0.6050 | 0.4724 | 0.6357 | 0.5774 | 0.5824 | 0.4439 | 0.6097 | 0.5486 | 0.5766 | 0.4271 | 0.5932 |
| | DSIMVC | 0.8207 | 0.7735 | 0.7276 | 0.8327 | 0.7337 | 0.6790 | 0.6148 | 0.7464 | 0.6152 | 0.5769 | 0.4848 | 0.6361 | 0.6245 | 0.5822 | 0.4867 | 0.6343 |
| | DVIMC | 0.9372 | 0.9345 | 0.9157 | 0.9466 | 0.9365 | 0.9060 | 0.8987 | 0.9368 | 0.8998 | 0.8699 | 0.8538 | 0.9019 | 0.8741 | 0.8184 | 0.7999 | 0.8763 |
| | MVP | 0.8657 | 0.8778 | 0.7864 | 0.8667 | 0.8590 | 0.8635 | 0.8222 | 0.8777 | 0.8355 | 0.8195 | 0.7792 | 0.8464 | 0.6660 | 0.6744 | 0.5721 | 0.6694 |
| | PMIMC | 0.7127 | 0.6985 | 0.5927 | 0.7432 | 0.7187 | 0.6593 | 0.5623 | 0.7458 | 0.6616 | 0.5739 | 0.4947 | 0.6695 | 0.6044 | 0.5344 | 0.4393 | 0.6220 |
| | ACOVA (Ours) | 0.9663 | 0.9478 | 0.9440 | 0.9664 | 0.9344 | 0.9125 | 0.9016 | 0.9348 | 0.9599 | 0.9001 | 0.9144 | 0.9599 | 0.9377 | 0.8579 | 0.8697 | 0.9377 |
| CUB Image-Caption | BSV | 0.6723 | 0.6604 | 0.4818 | 0.6823 | 0.6080 | 0.5858 | 0.3376 | 0.6163 | 0.5270 | 0.5241 | 0.2295 | 0.5387 | 0.4493 | 0.4474 | 0.1266 | 0.4630 |
| | Concat | 0.6890 | 0.6685 | 0.4979 | 0.6947 | 0.5837 | 0.5790 | 0.3322 | 0.5987 | 0.5187 | 0.5107 | 0.2068 | 0.5323 | 0.4663 | 0.4658 | 0.1391 | 0.4770 |
| | DCP | 0.5227 | 0.5822 | 0.2867 | 0.5833 | 0.5163 | 0.5781 | 0.3008 | 0.5797 | 0.4040 | 0.4430 | 0.1196 | 0.4587 | 0.2653 | 0.2337 | 0.0571 | 0.2867 |
| | CPSPAN | 0.6973 | 0.6766 | 0.5420 | 0.7153 | 0.6067 | 0.5914 | 0.4153 | 0.6160 | 0.5407 | 0.5198 | 0.2762 | 0.5487 | 0.4957 | 0.4797 | 0.1953 | 0.5033 |
| | DSIMVC | 0.6380 | 0.5744 | 0.4280 | 0.6427 | 0.6420 | 0.5663 | 0.4236 | 0.6420 | 0.5810 | 0.5419 | 0.3824 | 0.5860 | 0.5410 | 0.4965 | 0.3275 | 0.5440 |
| | DVIMC | 0.6573 | 0.6585 | 0.5246 | 0.6807 | 0.6703 | 0.6502 | 0.5131 | 0.6807 | 0.6057 | 0.6026 | 0.4538 | 0.6160 | 0.4710 | 0.4694 | 0.2795 | 0.4803 |
| | MVP | 0.6497 | 0.6317 | 0.5051 | 0.6667 | 0.6840 | 0.6545 | 0.5256 | 0.6903 | 0.6163 | 0.5750 | 0.4401 | 0.6287 | 0.6440 | 0.6036 | 0.4655 | 0.6510 |
| | PMIMC | 0.7183 | 0.7131 | 0.5795 | 0.7383 | 0.6820 | 0.6847 | 0.5360 | 0.6977 | 0.6517 | 0.6198 | 0.4846 | 0.6590 | 0.5907 | 0.5480 | 0.4034 | 0.6060 |
| | ACOVA (Ours) | 0.7970 | 0.7695 | 0.6581 | 0.7970 | 0.7563 | 0.7335 | 0.6215 | 0.7590 | 0.6823 | 0.6568 | 0.5267 | 0.6917 | 0.6554 | 0.6136 | 0.4730 | 0.6584 |

formance in most cases, demonstrating the effectiveness of our cross-view correlation modeling and learning mechanism. Notably, this performance gain comes with negligible additional complexity. ACOVA introduces only 3 and 21 extra learnable parameters in the 2-view and 6-view settings, respectively, relative to DVIMC backbones with over 6 million and 16 million parameters.

## 5.3. Ablation Study

**Effectiveness of Learning the Correlation.** To investigate the effectiveness of learning cross-view correlations, we compare our method (Learn **R** (*w/* corr.)) to an ablated version (*w/o* corr.), where we assume view independence during posterior aggregation. Note, ACOVA in this case sim-

plifies to baseline (Xu et al., 2024). As shown in Table 2, we consistently outperform '*w/o* corr.' across all missing rates on both datasets, indicating the effectiveness and robustness of adaptively modeling and learning view correlations during posterior aggregation. To provide a more comprehensive analysis, we further conduct the grid search over fixed correlation coefficients in [0.1, 0.3, 0.5, 0.7, 0.9]. Note, while the fixed grid search sometimes yields competitive results, its optimal value varies significantly across datasets and missing rates, making it impractical to determine without ground-truth supervision in real clustering scenarios, especially for more severe missing conditions. In contrast, our correlation learning strategy adaptively captures the intrinsic relationships between views without manual tuning or supervision, demonstrating both effectiveness and robustness

*Table 2.* Ablation results on Handwritten and Scene15 datasets with different missing-view rates. '*w/*' and '*w/o*' mean '*with*' and '*without*', respectively. 'corr.' refers to considering cross-view correlations. The best result in each column is denoted in **bold**. Note that all results are averaged over five runs.

| | Variant | 10% | | | | 30% | | | | 50% | | | | 70% | | | |
|---|---|---|---|---|---|---|---|---|---|---|---|---|---|---|---|---|---|
| | | ACC↑ | NMI↑ | ARI↑ | PUR↑ | ACC↑ | NMI↑ | ARI↑ | PUR↑ | ACC↑ | NMI↑ | ARI↑ | PUR↑ | ACC↑ | NMI↑ | ARI↑ | PUR↑ |
| Handwritten | Learn R (*w/* corr.) | **0.9363** | 0.8692 | 0.8645 | **0.9363** | **0.9376** | **0.8709** | **0.8669** | **0.9376** | **0.9146** | **0.8489** | **0.8325** | **0.9146** | 0.8999 | **0.8351** | **0.8176** | 0.9021 |
| | *w/o* corr. | 0.9169 | 0.8604 | 0.8547 | 0.9201 | 0.8954 | 0.8470 | 0.8210 | 0.9020 | 0.8980 | 0.8373 | 0.8175 | 0.9054 | 0.7808 | 0.7905 | 0.7197 | 0.8099 |
| | fixed $\rho = 0.1$ | 0.9339 | 0.8676 | 0.8595 | 0.9339 | 0.9249 | 0.8534 | 0.8418 | 0.9249 | 0.8860 | 0.8288 | 0.8031 | 0.8898 | 0.8881 | 0.8231 | 0.7986 | 0.8918 |
| | fixed $\rho = 0.3$ | 0.8874 | 0.8513 | 0.8231 | 0.8945 | 0.9217 | 0.8509 | 0.8355 | 0.9217 | 0.8929 | 0.8335 | 0.8093 | 0.8972 | 0.8859 | 0.8219 | 0.7962 | 0.8903 |
| | fixed $\rho = 0.5$ | 0.9088 | 0.8590 | 0.8391 | 0.9118 | 0.9301 | 0.8612 | 0.8510 | 0.9301 | 0.8710 | 0.8291 | 0.7957 | 0.8821 | 0.8867 | 0.8239 | 0.7962 | 0.8908 |
| | fixed $\rho = 0.7$ | 0.9069 | 0.8568 | 0.8322 | 0.9091 | 0.9317 | 0.8638 | 0.8545 | 0.9317 | 0.8963 | 0.8410 | 0.8161 | 0.9001 | **0.9115** | 0.8319 | 0.8145 | **0.9115** |
| | fixed $\rho = 0.9$ | 0.9362 | **0.8749** | **0.8649** | 0.9362 | 0.9172 | 0.8445 | 0.8263 | 0.9172 | 0.8786 | 0.8330 | 0.8036 | 0.8828 | 0.8977 | 0.8140 | 0.7934 | 0.8977 |
| | *w/o* $\mathcal{L}_{\text{ELBO}}$ | 0.6240 | 0.5284 | 0.3673 | 0.6240 | 0.5980 | 0.4548 | 0.3227 | 0.5980 | 0.5495 | 0.3924 | 0.2874 | 0.5545 | 0.4845 | 0.3357 | 0.2301 | 0.5020 |
| | *w/o* $\mathcal{L}_{\text{align}}$ | 0.8235 | 0.7768 | 0.7036 | 0.8235 | 0.6930 | 0.6378 | 0.5179 | 0.6930 | 0.6645 | 0.6259 | 0.5179 | 0.6675 | 0.6090 | 0.5664 | 0.4574 | 0.6100 |
| Scene15 | Learn R (*w/* corr.) | 0.5040 | **0.4804** | 0.3273 | 0.5270 | **0.4807** | **0.4560** | **0.3106** | **0.5119** | **0.4710** | **0.4360** | **0.2973** | **0.4987** | **0.4428** | **0.4083** | **0.2733** | **0.4636** |
| | *w/o* corr. | 0.4863 | 0.4780 | 0.3192 | 0.5021 | 0.4678 | 0.4440 | 0.3097 | 0.4823 | 0.4371 | 0.4163 | 0.2737 | 0.4577 | 0.4136 | 0.3950 | 0.2519 | 0.4286 |
| | fixed $\rho = 0.1$ | 0.4963 | 0.4722 | 0.3237 | 0.5220 | 0.4702 | 0.4466 | 0.2977 | 0.4941 | 0.4653 | 0.4292 | 0.2955 | 0.4966 | 0.4181 | 0.3878 | 0.2518 | 0.4287 |
| | fixed $\rho = 0.3$ | **0.5057** | 0.4743 | 0.3272 | **0.5281** | 0.4746 | 0.4458 | 0.3011 | 0.4986 | 0.4598 | 0.4275 | 0.2901 | 0.4885 | 0.4165 | 0.3882 | 0.2508 | 0.4309 |
| | fixed $\rho = 0.5$ | 0.4963 | 0.4673 | 0.3214 | 0.5218 | 0.4689 | 0.4414 | 0.3005 | 0.4891 | 0.4569 | 0.4224 | 0.2863 | 0.4815 | 0.4105 | 0.3893 | 0.2521 | 0.4228 |
| | fixed $\rho = 0.7$ | 0.4761 | 0.4543 | 0.3023 | 0.5031 | 0.4742 | 0.4255 | 0.2907 | 0.4940 | 0.4590 | 0.4295 | 0.2818 | 0.4765 | 0.4192 | 0.3827 | 0.2503 | 0.4353 |
| | fixed $\rho = 0.9$ | 0.4649 | 0.4505 | 0.2938 | 0.4939 | 0.4609 | 0.4187 | 0.2710 | 0.4727 | 0.3918 | 0.3877 | 0.2083 | 0.4144 | 0.3656 | 0.3575 | 0.2239 | 0.3765 |
| | *w/o* $\mathcal{L}_{\text{ELBO}}$ | 0.1599 | 0.1605 | 0.0519 | 0.1632 | 0.1407 | 0.0870 | 0.0050 | 0.1431 | 0.1405 | 0.0854 | 0.0220 | 0.1411 | 0.1394 | 0.0746 | 0.0170 | 0.1402 |
| | *w/o* $\mathcal{L}_{\text{align}}$ | 0.4566 | 0.4624 | 0.2926 | 0.4941 | 0.3974 | 0.4000 | 0.2335 | 0.4193 | 0.3393 | 0.3412 | 0.1902 | 0.3454 | 0.2105 | 0.1798 | 0.0821 | 0.2157 |

of our method.

**Effectiveness of Training Objectives.** Our objective function consists of two components: $\mathcal{L}_{\text{ELBO}}$ and $\mathcal{L}_{\text{align}}$. As demonstrated in Table 2. Removing either of them leads to significant performance drops, verifying their complementary roles, where the $\mathcal{L}_{\text{ELBO}}$ ensures effective representation learning and $\mathcal{L}_{\text{align}}$ promotes cross-view consistency.

### 5.4. Correlation Learning Analysis

To further demonstrate the view relationships captured by the correlation matrix **R** and validate our motivation, we conducted controlled simulation experiments on the Fashion dataset. Specifically, we created two views by duplicating the original images. To simulate heterogeneous views, we adopted a mutually exclusive noise injection strategy where each sample had exactly one corrupted view (view 1 for randomly selected 50% of samples, view 2 for the rest). We introduced pepper noise rates $\in \{0.00, 0.50, 1.00\}$ to control the degree of view heterogeneity, and missing rates $\in \{0\%, 10\%, 30\%, 50\%, 70\%\}$ to simulate incomplete data scenarios.

Fig. 3 visualizes the learned off-diagonal correlations in **R** across all combinations of noise and missing rates. A clear bi-directional declining trend emerges: as the noise rate increases from 0.00 to 1.00, the two views gradually become less correlated due to increasing heterogeneity. Similarly, as the missing rate increases, the correlation values in **R** progressively decrease across all noise conditions, as incomplete information limits the shared structure between views. These results validate that our learned **R** adaptively captures meaningful cross-view relationships under both heterogeneous and incomplete conditions, supporting the

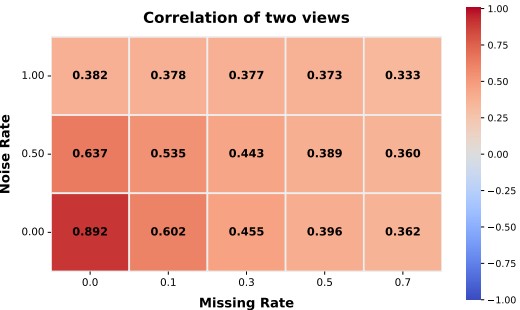

*Figure 3.* Learned correlations under different noise and missing rates.

necessity of adaptive correlation learning for realistic multi-view clustering. Notably, even under maximal noise and high missingness, the off-diagonal correlations do not collapse fully to zero (the prior). We hypothesize that this arises from the generative model itself, which requires the latent variables to retain some meaningful information to generate the views. More discussion on the correlation matrix can be found in Appendix C.

### 5.5. Visualization and Parameter Sensitivity Analysis

**Visualization.** Fig. 4(a-c) compares the t-SNE visualization between CONCAT (Zhao et al., 2016), DVIMC (Xu et al., 2024) and our method with a missing rate of 10%. Our approach shows clearer separation and fewer outliers, indicating more discriminative representations by learning cross-view correlations adaptively from incomplete data. In addition, we plot the correlation matrix **R** learned from the incomplete Handwritten datasets with 50% missing rates in Fig. 4(d), where we observe meaningful view dependencies. The views for this dataset corresponding to: Fourier coefficients, Profile correlations, Karhunen-Loève coefficients,

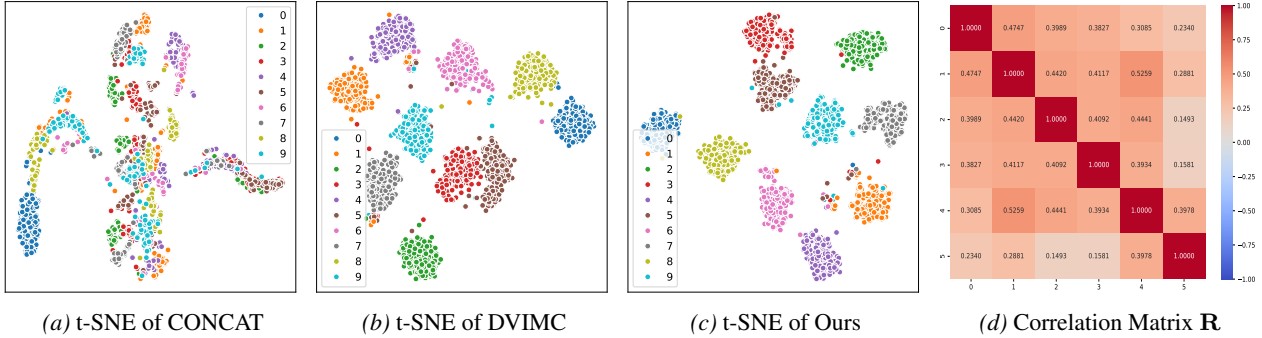

(a) t-SNE of CONCAT      (b) t-SNE of DVIMC      (c) t-SNE of Ours      (d) Correlation Matrix **R**

*Figure 4.* t-SNE and learned correlation matrix visualization on the Handwritten dataset.

Zernike moments, Pixel averages, and morphological features (in that order). From Fig. 4(d), we notice that most views have the least correlation with the last view, which intuitively makes sense, as the morphological features are significantly different from the frequency- or intensity-based features in the other views. Further, we notice high correlation between profile correlations and pixel averages, where both are linked to the underlying intensity distributions. We also observe high correlation between Karhunen-Loève coefficients and pixel averages, which indicates that much of the variation in the dataset is due to average intensities, which also aligns well with what we expect for this dataset. More visualization on different datasets and missing rates are included in Appendix C.

**Parameter Sensitivity Analysis.** This section empirically analyzes the effect on ACC of the trade-off parameter $\alpha$ in Eq. 15 across different missing rates for the Scene15 and Caltech5V datasets. As shown in Fig. 5, $\alpha$ in the range of [5,20] achieves higher performance. Additional sensitivity studies and analyses can be found in Appendix C.

## 6. Discussion and Conclusion

In this work, we propose ACOVA, an imputation-free variational framework that adaptively learns inter-view correlations for IMVC. By going beyond the conventional independence assumption and modeling cross-view correlation through view-specific estimation error, our method captures the cross-view dependencies. Besides, we propose to jointly optimize the posterior representations and cross-view correlation, enabling more effective latent embeddings and significantly boosting clustering performance, especially under high missing-view conditions. Extensive experiments on multiple benchmarks verify the effectiveness and robustness of our approach compared to prior IMVC methods.

## Impact Statement

This work advances a correlation-aware variational aggregation perspective for incomplete multi-view learning, demon-

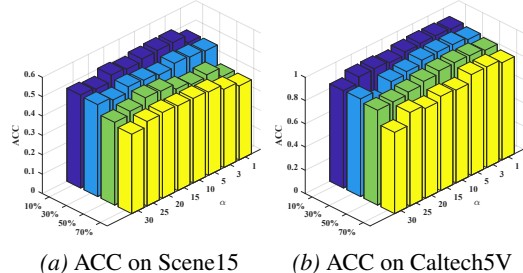

(a) ACC on Scene15      (b) ACC on Caltech5V

*Figure 5.* Clustering performance with different parameter $\alpha$ and missing rate settings on Scene15 and Caltech5V dataset.

strating that explicitly modeling inter-view dependencies (rather than assuming conditional independence) can lead to substantial performance gains. By adaptively learning cross-view correlations within a unified variational formulation, the approach mitigates the need for fixed correlation assumptions or costly hyperparameter searches. Although evaluated in incomplete multi-view clustering, where data missingness amplifies the importance of correlation modeling, our framework is general and can be used as a drop-in replacement for CoDE or the popular Mixture/Product of Experts, when dependency modeling is beneficial. This includes both other unsupervised and supervised settings, some natural follow-ups being image synthesis for fusing modalities in generative adversarial networks (Huang et al., 2022), comparative text assessment in large language models (Liusie et al., 2024), and distillation aggregation for diffusion policies in diffusion models (Zhou et al., 2024).

## Acknowledgment

This work was supported by the Beijing Natural Science-Xiaomi Innovation Joint Foundation (No. L253007). It was further supported by the Research Council of Norway (NFR), through its Center for Research-based Innovation (grant no. 309439) and FRIPRO (grant no. 303514 and 360068) funding schemes.

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

# A. Theoretical Derivations and Analysis

## A.1. Mathematical Background

This section provides the mathematical foundations (Jiang et al., 2017; Yin et al., 2020; Xu et al., 2022) for the key equations in variational incomplete multi-view clustering preliminaries in Section 3.

### A.1.1. Derivation of the Joint Probability (Equation 1)

The generative process assumes a probabilistic model where cluster assignment $\mathbf{c}$ is drawn from a categorical distribution, shared latent representation $\mathbf{z}$ depends on the cluster assignment, and each view $\mathbf{x}_v$ is conditionally independent of other views given $\mathbf{z}$. For the joint probability $p(\{\mathbf{x}_v\}_{v=1}^V, \mathbf{z}, \mathbf{c})$, we could first apply the chain rule of probability to get:

$$p(\{\mathbf{x}_v\}_{v=1}^V, \mathbf{z}, \mathbf{c}) = p(\mathbf{c}) \cdot p(\mathbf{z} \mid \mathbf{c}) \cdot p(\{\mathbf{x}_v\}_{v=1}^V \mid \mathbf{z}, \mathbf{c}). \tag{17}$$

Given the latent representation $\mathbf{z}$, we assume the observations are independent of the cluster assignment: $\mathbf{x}_v \perp \mathbf{c}|\mathbf{z}$, which gives $p(\{\mathbf{x}_v\}_{v=1}^V \mid \mathbf{z}, \mathbf{c}) = p(\{\mathbf{x}_v\}_{v=1}^V \mid \mathbf{z})$. Besides, given the shared representation $\mathbf{z}$, we assume different views are conditionally independent: $\mathbf{x}_i \perp \mathbf{x}_j|\mathbf{z}$ for $i \neq j$, yielding $p(\{\mathbf{x}_v\}_{v=1}^V \mid \mathbf{z}) = \prod_{v=1}^V p(\mathbf{x}_v \mid \mathbf{z})$. This leads directly to Eq.1:

$$p(\{\mathbf{x}_v\}_{v=1}^V, \mathbf{z}, \mathbf{c}) = p(\mathbf{c}) \, p(\mathbf{z} \mid \mathbf{c}) \prod_{v=1}^V p(\mathbf{x}_v \mid \mathbf{z}). \tag{18}$$

### A.1.2. Derivation of the Posterior Distribution (Equation 2)

For incomplete multi-view data where only views $v \in \mathcal{V}$ are observed, the posterior distribution follows from Bayes' theorem:

$$p(\mathbf{z}, \mathbf{c}|\{\mathbf{x}_v\}_{v \in \mathcal{V}}) = \frac{p(\{\mathbf{x}_v\}_{v \in \mathcal{V}}, \mathbf{z}, \mathbf{c})}{p(\{\mathbf{x}_v\}_{v \in \mathcal{V}})}. \tag{19}$$

For the incomplete setting, missing views are implicitly marginalized out. Under the conditional independence assumption, the joint likelihood for observed views becomes:

$$p(\{\mathbf{x}_v\}_{v \in \mathcal{V}} \mid \mathbf{z}) = \prod_{v \in \mathcal{V}} p(\mathbf{x}_v \mid \mathbf{z}). \tag{20}$$

The marginal likelihood requires summing over all latent variables, yielding Eq.2:

$$p(\mathbf{z}, \mathbf{c}|\{\mathbf{x}_v\}_{v \in \mathcal{V}}) = \frac{p(\{\mathbf{x}_v\}_{v \in \mathcal{V}}|\mathbf{z})p(\mathbf{z}, \mathbf{c})}{\int_{\mathbf{z}} \sum_{\mathbf{c}=1}^C p(\{\mathbf{x}_v\}_{v \in \mathcal{V}}|\mathbf{z}, \mathbf{c})p(\mathbf{z}, \mathbf{c})d\mathbf{z}}. \tag{21}$$

### A.1.3. Variational Approximation (Equation 3)

The exact posterior is computationally intractable due to the complex integration and summation in the denominator. Variational inference approximates this with a tractable distribution $q_\phi(\mathbf{z}, \mathbf{c}|\{\mathbf{x}_v\}_{v \in \mathcal{V}})$ by minimizing the Kullback-Leibler divergence (Eq.3):

$$D_{\mathrm{KL}}(q_\phi(\mathbf{z}, \mathbf{c}|\{\mathbf{x}_v\}_{v \in \mathcal{V}})||p(\mathbf{z}, \mathbf{c}|\{\mathbf{x}_v\}_{v \in \mathcal{V}})). \tag{22}$$

As for optimization, minimizing the KL divergence is equivalent to maximizing the Evidence Lower BOund (ELBO), and please refer to A.5 for more detailed derivations.

## A.2. Proof of the Frobenius Identity Deviation Bound

We provide the proof of the Theorem of Frobenius Identity Deviation Bound, which establishes an upper bound on the Frobenius norm deviation of a correlation matrix from the identity.

*Proof.* Let $\mathbf{R} \in \mathbb{R}^{V \times V}$ be a correlation matrix, i.e., a symmetric matrix with unit diagonal entries $R_{ii} = 1$ for all $i$, and off-diagonal entries satisfying $|R_{ij}| \leq 1$ for all $i \neq j$. We compute the Frobenius norm of the deviation from the identity:

$$\|\mathbf{R} - \mathbf{I}\|_F^2 = \sum_{i,j}(R_{ij} - \delta_{ij})^2 = \sum_{i \neq j} R_{ij}^2, \tag{23}$$

where $\delta_{ij}$ is the Kronecker delta, and we used the fact that $R_{ii} - 1 = 0$. Since $|R_{ij}| \leq 1$ for all $i \neq j$, it follows that:

$$\sum_{i \neq j} R_{ij}^2 \leq \sum_{i \neq j} 1 = V(V-1). \tag{24}$$

Taking the square root of both sides yields the desired result:

$$\|\mathbf{R} - \mathbf{I}\|_F \leq \sqrt{V(V-1)}. \tag{25}$$

$\square$

## A.3. Proof of Identifiability of the Covariance Parameterization

In the following, we demonstrate the identifiability of the proposed covariance parameterization.

**Theorem A.1** (Identifiability of the Covariance Parameterization). *Let $\mathbf{D} = \mathrm{diag}(\boldsymbol{\sigma}_1, \ldots, \boldsymbol{\sigma}_V)$ and $\mathbf{D}' = \mathrm{diag}(\sigma_1', \ldots, \sigma_V')$ be diagonal matrices with strictly positive entries. Let $\mathbf{L}$ and $\mathbf{L}'$ be lower–triangular matrices with strictly positive diagonal entries, and define correlation matrices*

$$\mathbf{R} = \mathrm{corr}(\mathbf{L}\mathbf{L}^\top), \qquad \mathbf{R}' = \mathrm{corr}(\mathbf{L}'\mathbf{L}'^\top), \tag{26}$$

*i.e. $\mathbf{R}_{ij} = (\mathbf{L}\mathbf{L}^\top)_{ij} / \sqrt{(\mathbf{L}\mathbf{L}^\top)_{ii}(\mathbf{L}\mathbf{L}^\top)_{jj}}$ (and analogously for $\mathbf{R}'$). For the $d$-th dimension, define the covariance matrices*

$$\boldsymbol{\Sigma}_d = \mathbf{D}\mathbf{R}\mathbf{D}, \qquad \boldsymbol{\Sigma}_d' = \mathbf{D}'\mathbf{R}'\mathbf{D}'. \tag{27}$$

*If $\boldsymbol{\Sigma}_d = \boldsymbol{\Sigma}_d'$, then*

$$\mathbf{D} = \mathbf{D}', \qquad \mathbf{R} = \mathbf{R}', \qquad \mathbf{L} = \mathbf{L}'. \tag{28}$$

*In particular, the mapping $(\mathbf{L}, \mathbf{D}) \mapsto \boldsymbol{\Sigma}_d$ is identifiable.*

*Proof.* Assume $\boldsymbol{\Sigma}_d = \boldsymbol{\Sigma}_d'$, i.e.,

$$\mathbf{D}\mathbf{R}\mathbf{D} = \mathbf{D}'\mathbf{R}'\mathbf{D}'. \tag{29}$$

Since $\mathbf{D}$ is invertible, left- and right-multiplying Eq. 29 by $\mathbf{D}^{-1}$ yields

$$\mathbf{R} = \mathbf{A}\mathbf{R}'\mathbf{A}, \qquad \mathbf{A} := \mathbf{D}^{-1}\mathbf{D}'. \tag{30}$$

Here $\mathbf{A} = \mathrm{diag}(\mathbf{a}_1, \ldots, \mathbf{a}_V)$ with all $\mathbf{a}_i > 0$. Because $\mathbf{R}$ and $\mathbf{R}'$ are correlation matrices, we have $\mathbf{R}_{ii} = \mathbf{R}_{ii}' = 1$ for all $i$. Taking the diagonal entries of the equation $\mathbf{R} = \mathbf{A}\mathbf{R}'\mathbf{A}$ gives

$$1 = \mathbf{R}_{ii} = (\mathbf{A}\mathbf{R}'\mathbf{A})_{ii} = \mathbf{a}_i^2 \mathbf{R}_{ii}' = \mathbf{a}_i^2, \qquad i = 1, \ldots, V. \tag{31}$$

Hence $\mathbf{a}_i = 1$ for all $i$, implying $\mathbf{A} = \mathbf{I}$ and thus $\mathbf{D}' = \mathbf{D}$. Substituting $\mathbf{D}' = \mathbf{D}$ into Eq. 29 gives

$$\mathbf{D}\mathbf{R}\mathbf{D} = \mathbf{D}\mathbf{R}'\mathbf{D} \quad \Rightarrow \quad \mathbf{R} = \mathbf{R}'. \tag{32}$$

By construction, $\mathbf{R}$ and $\mathbf{R}'$ are correlation matrices obtained from $\mathbf{L}\mathbf{L}^\top$ and $\mathbf{L}'\mathbf{L}'^\top$, respectively. Equality $\mathbf{R} = \mathbf{R}'$ implies $\mathbf{L}\mathbf{L}^\top = \mathbf{L}'\mathbf{L}'^\top$. For any positive definite matrix, the Cholesky factor that is lower triangular with strictly positive diagonal entries is unique. Therefore, $\mathbf{L} = \mathbf{L}'$. Thus $\boldsymbol{\Sigma}_d = \boldsymbol{\Sigma}_d'$ implies $\mathbf{D} = \mathbf{D}'$, $\mathbf{R} = \mathbf{R}'$, and $\mathbf{L} = \mathbf{L}'$, proving identifiability of the parameterization. $\square$

Note, the only non-identifiable quantities are the internal neural-network weights themselves, which is expected and does not affect the identifiability of the covariance structure.

## A.4. Complete Derivation of Posterior Distribution

In this section, we provide a detailed derivation of the aggregated posterior distribution under incomplete multi-view settings. Specifically, given a subset of available modalities indexed by a binary mask $\mathbf{M} \in \{0,1\}^V$ for each sample, we derive the variational posterior $q(\mathbf{z} \mid \{\mathbf{x}_v\}_{v \in \mathcal{V}})$. This derivation is presented for one latent dimension, and stacking all $D$ dimensions recovers the main-text $VD$-dimensional form. Note that since our inference model produces view-specific latent predictions $\boldsymbol{\mu}$, this posterior can be equivalently expressed as $q(\mathbf{z}|\boldsymbol{\mu}, \mathbf{M})$, where $\boldsymbol{\mu}$ contains the sufficient statistics from the encoder networks.

$$q(\mathbf{z} \mid \{\mathbf{x}_v\}_{v \in \mathcal{V}}) = q(\mathbf{z}|\boldsymbol{\mu}, \mathbf{M}) \sim \mathcal{N}(\tilde{\boldsymbol{\mu}}, \tilde{\boldsymbol{\sigma}}^2) = \mathcal{N}(\mathbf{A}_{\mathbf{M}}^{-1}\mathbf{B}_{\mathbf{M}}, \mathbf{A}_{\mathbf{M}}^{-1}) \tag{33}$$

We begin by modeling the latent estimation as:

$$\boldsymbol{\mu} = \mathbb{1} \cdot \mathbf{z} + \boldsymbol{\epsilon}, \quad \boldsymbol{\epsilon} \sim \mathcal{N}(\mathbf{0}, \boldsymbol{\Sigma}), \tag{34}$$

where $\boldsymbol{\mu} \in \mathbb{R}^V$ is the vector of view-specific latent predictions, $\mathbf{z} \in \mathbb{R}$ is the shared latent variable for this one-dimensional derivation, and $\boldsymbol{\Sigma} \in \mathbb{R}^{V \times V}$ is the correlated noise covariance. Here, $\mathbb{1}$ is a binary design matrix to enable dimensional alignment. The expectation and covariance of $\boldsymbol{\mu}$ are thus:

$$\mathbb{E}[\boldsymbol{\mu}] = \mathbb{E}[\mathbb{1} \cdot \mathbf{z} + \boldsymbol{\epsilon}] = \mathbb{1} \cdot \mathbf{z},$$
$$\mathrm{Cov}[\boldsymbol{\mu}] = \mathrm{Cov}[\boldsymbol{\epsilon}] = \boldsymbol{\Sigma} \tag{35}$$

That is,

$$\boldsymbol{\mu} \sim \mathcal{N}(\mathbb{1} \cdot \mathbf{z}, \boldsymbol{\Sigma}) \tag{36}$$

When only a subset of views are observed, represented by mask $\mathbf{M}$, we define masked observations as:

$$\boldsymbol{\mu}_{\mathbf{M}} = \boldsymbol{\mu} \odot \mathbf{M},$$
$$\boldsymbol{\mu}_{\mathbf{M}} \sim \mathcal{N}(\mathbb{1} \cdot \mathbf{z} \odot \mathbf{M}, \boldsymbol{\Sigma}_{\mathbf{M}}), \tag{37}$$

where the masked covariance is given by:

$$\boldsymbol{\Sigma}_{\mathbf{M}} = \boldsymbol{\Sigma} \odot (\mathbf{M}\mathbf{M}^{\top}) \tag{38}$$

Assuming an improper flat prior distribution on $\mathbf{z}$, the variational posterior is proportional to the likelihood function. Therefore,

$$
\begin{aligned}
q(\mathbf{z}|\boldsymbol{\mu}, \mathbf{M}) &\propto \exp\left[-\frac{1}{2}(\boldsymbol{\mu} \odot \mathbf{M} - \mathbb{1}\mathbf{z})^{\top}(\boldsymbol{\Sigma}^{-1} \odot \mathbf{M}\mathbf{M}^{\top})(\boldsymbol{\mu} \odot \mathbf{M} - \mathbb{1}\mathbf{z})\right] \\
&= \exp\left[-\frac{1}{2}[(\boldsymbol{\mu} \odot \mathbf{M})^{\top}(\boldsymbol{\Sigma}^{-1} \odot \mathbf{M}\mathbf{M}^{\top})(\boldsymbol{\mu} \odot \mathbf{M}) - 2(\boldsymbol{\mu} \odot \mathbf{M})^{\top}(\boldsymbol{\Sigma}^{-1} \odot \mathbf{M}\mathbf{M}^{\top})\mathbb{1}\mathbf{z}\right. \\
&\quad \left. + \mathbf{z}^{\top}\mathbb{1}^{\top}(\boldsymbol{\Sigma}^{-1} \odot \mathbf{M}\mathbf{M}^{\top})\mathbb{1}\mathbf{z}]\right] \\
&= \exp\left[-\frac{1}{2}[\mathbf{z}^{\top}\mathbf{A}_{\mathbf{M}}\mathbf{z} - 2\mathbf{z}^{\top}\mathbf{B}_{\mathbf{M}} + \mathbf{C}]\right] \\
&= \exp\left[-\frac{1}{2}[(\mathbf{z} - \mathbf{A}_{\mathbf{M}}^{-1}\mathbf{B}_{\mathbf{M}})^{\top}\mathbf{A}_{\mathbf{M}}(\mathbf{z} - \mathbf{A}_{\mathbf{M}}^{-1}\mathbf{B}_{\mathbf{M}}) - \mathbf{B}_{\mathbf{M}}^{\top}\mathbf{A}_{\mathbf{M}}^{-1}\mathbf{B}_{\mathbf{M}} + \mathbf{C}]\right] \\
&\propto \exp\left[-\frac{1}{2}(\mathbf{z} - \mathbf{A}_{\mathbf{M}}^{-1}\mathbf{B}_{\mathbf{M}})^{\top}\mathbf{A}_{\mathbf{M}}(\mathbf{z} - \mathbf{A}_{\mathbf{M}}^{-1}\mathbf{B}_{\mathbf{M}})\right],
\end{aligned} \tag{39}
$$

where

The gradient and Hessian of $\log q(\mathbf{z}|\boldsymbol{\mu}, \mathbf{M})$ with respect to $\mathbf{z}$ are:

$$\nabla_{\mathbf{z}} \log q(\mathbf{z}|\boldsymbol{\mu}, \mathbf{M}) = -\mathbf{A_M}\mathbf{z} + \mathbf{B_M}$$
$$\nabla_{\mathbf{z}}^2 \log q(\mathbf{z}|\boldsymbol{\mu}, \mathbf{M}) = -\mathbf{A_M} \tag{40}$$

Therefore, we obtain the final form of the aggregated variational posterior distribution:

$$q(\mathbf{z} \mid \{\mathbf{x}_v\}_{v \in \mathcal{V}}) = q(\mathbf{z}|\boldsymbol{\mu}, \mathbf{M}) \sim \mathcal{N}(\mathbf{A_M}^{-1}\mathbf{B_M}, \mathbf{A_M}^{-1}) \qquad \square$$

## A.5. Complete Derivation of ELBO

In this section, we provide a detailed derivation of the evidence lower bound (ELBO), *i.e.*:

$$\mathcal{L}_{\text{ELBO}}(\{\mathbf{x}_v\}_{v \in \mathcal{V}}) = \mathbb{E}_{q_\phi(\mathbf{z}|\{\mathbf{x}_v\}_{v \in \mathcal{V}})} \left[ \sum_{v \in \mathcal{V}} \log p(\mathbf{x}_v \mid \mathbf{z}) \right]$$
$$- \mathbb{E}_{q_\phi(\mathbf{c}|\{\mathbf{x}_v\}_{v \in \mathcal{V}})} \left[ D_{\text{KL}} \left( q_\phi(\mathbf{z} \mid \{\mathbf{x}_v\}_{v \in \mathcal{V}}) \| p(\mathbf{z} \mid \mathbf{c}) \right) \right] - D_{\text{KL}} \left( q_\phi(\mathbf{c} \mid \{\mathbf{x}_v\}_{v \in \mathcal{V}}) \| p(\mathbf{c}) \right) \tag{41}$$

We begin with the Kullback-Leibler (KL) divergence between the variational joint distribution $q_\phi(\mathbf{z}, \mathbf{c}|\{\mathbf{x}_v\}_{v \in \mathcal{V}})$ and the true joint posterior $p(\mathbf{z}, \mathbf{c}|\{\mathbf{x}_v\}_{v \in \mathcal{V}})$:

$$D_{\text{KL}}(q_\phi(\mathbf{z}, \mathbf{c}|\{\mathbf{x}_v\}_{v \in \mathcal{V}})\|p(\mathbf{z}, \mathbf{c}|\{\mathbf{x}_v\}_{v \in \mathcal{V}}))$$
$$= \mathbb{E}_{q_\phi(\mathbf{z}, \mathbf{c}|\{\mathbf{x}_v\}_{v \in \mathcal{V}})} \left[ \log \frac{q_\phi(\mathbf{z}, \mathbf{c}|\{\mathbf{x}_v\}_{v \in \mathcal{V}})}{p(\mathbf{z}, \mathbf{c}|\{\mathbf{x}_v\}_{v \in \mathcal{V}})} \right]$$
$$= \mathbb{E}_{q_\phi(\mathbf{z}, \mathbf{c}|\{\mathbf{x}_v\}_{v \in \mathcal{V}})} \left[ \log \frac{q_\phi(\mathbf{z}, \mathbf{c}|\{\mathbf{x}_v\}_{v \in \mathcal{V}}) \cdot p(\{\mathbf{x}_v\}_{v \in \mathcal{V}})}{p(\mathbf{z}, \mathbf{c}, \{\mathbf{x}_v\}_{v \in \mathcal{V}})} \right]$$
$$= \mathbb{E}_{q_\phi(\mathbf{z}, \mathbf{c}|\{\mathbf{x}_v\}_{v \in \mathcal{V}})} \left[ \log \frac{q_\phi(\mathbf{z}, \mathbf{c}|\{\mathbf{x}_v\}_{v \in \mathcal{V}})}{p(\mathbf{z}, \mathbf{c}, \{\mathbf{x}_v\}_{v \in \mathcal{V}})} \right] + \log p(\{\mathbf{x}_v\}_{v \in \mathcal{V}}) \tag{42}$$

Since KL divergence is always non-negative, we have:

$$\log p(\{\mathbf{x}_v\}_{v \in \mathcal{V}}) \geq \mathbb{E}_{q_\phi(\mathbf{z}, \mathbf{c}|\{\mathbf{x}_v\}_{v \in \mathcal{V}})} \left[ \log \frac{p(\mathbf{z}, \mathbf{c}, \{\mathbf{x}_v\}_{v \in \mathcal{V}})}{q_\phi(\mathbf{z}, \mathbf{c}|\{\mathbf{x}_v\}_{v \in \mathcal{V}})} \right]$$
$$= \mathcal{L}_{\text{ELBO}}(\{\mathbf{x}_v\}_{v \in \mathcal{V}}) \tag{43}$$

Given the factorization $q_\phi(\mathbf{z}, \mathbf{c}|\{\mathbf{x}_v\}_{v \in \mathcal{V}}) = q_\phi(\mathbf{z}|\{\mathbf{x}_v\}_{v \in \mathcal{V}})q_\phi(\mathbf{c}|\{\mathbf{x}_v\}_{v \in \mathcal{V}})$ and the joint distribution:

$$p(\mathbf{z}, \mathbf{c}, \{\mathbf{x}_v\}_{v \in \mathcal{V}}) = p(\mathbf{c})p(\mathbf{z}|\mathbf{c}) \prod_{v \in \mathcal{V}} p(\mathbf{x}_v|\mathbf{z}), \tag{44}$$

we can then derive the Evidence Lower Bound (ELBO):

$$\mathcal{L}_{\text{ELBO}}(\{\mathbf{x}_v\}_{v\in\mathcal{V}})$$

$$= \mathbb{E}_{q_\phi(\mathbf{z},\mathbf{c}|\{\mathbf{x}_v\}_{v\in\mathcal{V}})} \left[ \log \frac{p(\mathbf{z},\mathbf{c},\{\mathbf{x}_v\}_{v\in\mathcal{V}})}{q_\phi(\mathbf{z},\mathbf{c}|\{\mathbf{x}_v\}_{v\in\mathcal{V}})} \right]$$

$$= \mathbb{E}_{q_\phi(\mathbf{z},\mathbf{c}|\{\mathbf{x}_v\}_{v\in\mathcal{V}})} \left[ \log \frac{p(\mathbf{c})p(\mathbf{z}|\mathbf{c}) \prod_{v\in\mathcal{V}} p(\mathbf{x}_v|\mathbf{z})}{q_\phi(\mathbf{z}|\{\mathbf{x}_v\}_{v\in\mathcal{V}})q_\phi(\mathbf{c}|\{\mathbf{x}_v\}_{v\in\mathcal{V}})} \right]$$

$$= \mathbb{E}_{q_\phi(\mathbf{z},\mathbf{c}|\{\mathbf{x}_v\}_{v\in\mathcal{V}})} \left[ \sum_{v\in\mathcal{V}} \log p(\mathbf{x}_v|\mathbf{z}) + \log p(\mathbf{c}) \right.$$

$$\left. + \log p(\mathbf{z}|\mathbf{c}) - \log q_\phi(\mathbf{z}|\{\mathbf{x}_v\}_{v\in\mathcal{V}}) - \log q_\phi(\mathbf{c}|\{\mathbf{x}_v\}_{v\in\mathcal{V}}) \right]$$

$$= \mathbb{E}_{q_\phi(\mathbf{z}|\{\mathbf{x}_v\}_{v\in\mathcal{V}})} \left[ \sum_{v\in\mathcal{V}} \log p(\mathbf{x}_v|\mathbf{z}) \right]$$

$$- \mathbb{E}_{q_\phi(\mathbf{c}|\{\mathbf{x}_v\}_{v\in\mathcal{V}})} \left[ D_{\text{KL}}(q_\phi(\mathbf{z}|\{\mathbf{x}_v\}_{v\in\mathcal{V}})\|p(\mathbf{z}|\mathbf{c})) \right] - D_{\text{KL}}(q_\phi(\mathbf{c}|\{\mathbf{x}_v\}_{v\in\mathcal{V}})\|p(\mathbf{c})) \qquad \square$$

### A.6. Note on Efficient and Stable Inversion

Our formulation expresses $\boldsymbol{\Sigma}^d = \mathbf{DRD}$, with $\mathbf{R}_{ij} = \frac{(\mathbf{LL}^\top)_{ij}}{\sqrt{(\mathbf{LL}^\top)_{ii}(\mathbf{LL}^\top)_{jj}}}$, where $\mathbf{L}$ is a learned lower-triangular matrix with strictly positive diagonal entries. This parameterization guarantees $\mathbf{R} \succ 0$. Directly forming or inverting $\mathbf{R}^{-1}$ or $\boldsymbol{\Sigma}^{d^{-1}}$ could in certain cases be numerically unstable. Although we have not observed this in our experiments, this can be avoided altogether as we do not have to explicitly compute the inverse and can instead exploit the structure of the Cholesky decomposition using forward and backward substitution, as outlined in the following:

Given our learned $\mathbf{L}$, the correlation matrix and its inverse are $\mathbf{R} = \mathbf{D}_R^{-1}(\mathbf{LL}^\top)\mathbf{D}_R^{-1}$ and $\mathbf{R}^{-1} = \mathbf{D}_R(\mathbf{LL}^\top)^{-1}\mathbf{D}_R$, where $\mathbf{D}_R = \text{diag}(\sqrt{(\mathbf{LL}^\top)_{11}}, \ldots, \sqrt{(\mathbf{LL}^\top)_{VV}})$. Consequently, $\boldsymbol{\Sigma}^{d^{-1}} = \mathbf{D}^{-1}\mathbf{R}^{-1}\mathbf{D}^{-1} = (\mathbf{D}^{-1}\mathbf{D}_R)\mathbf{L}^{-\top}\mathbf{L}^{-1}(\mathbf{D}_R\mathbf{D}^{-1})$, in which $\mathbf{L}^{-\top}$ and $\mathbf{L}^{-1}$ can be handled via forward and backward substitution instead of explicit matrix inversion, being numerically more stable as well as requiring $\mathcal{O}(V^2)$ rather than $\mathcal{O}(V^3)$.

### A.7. Algorithm Description

The pseudo-code of optimizing our ACOVA is summarized in Algorithm 1.

---

**Algorithm 1** Optimizing process of **ACOVA**

---

**Require:** Incomplete multi-view data $\{\mathcal{X}\}$, missing-view indicator matrix $\mathcal{M}$, number of clusters $C$, trade-off parameters $\alpha$, and total training epoch $T$.
**Ensure:** The clustering result $\bar{\mathbf{Y}}$.
1: Initialize model parameters $\{\boldsymbol{\phi}_v, \boldsymbol{\theta}_v\}$, cluster distribution parameters $\{\boldsymbol{\tau}_c, \boldsymbol{\mu}_c, \boldsymbol{\Sigma}_c^2\}$, correlation matrix $\mathbf{R}$.
2: **while** not reaching $T$ **do**
3:     Compute the distribution parameters $\{\boldsymbol{\mu}_v, \boldsymbol{\sigma}_v^2\}_{v\in\mathcal{V}}$ through view-specific encoders.
4:     Compute the inter-view error covariance matrix $\boldsymbol{\Sigma}^d$ by Eq. 7.
5:     Compute the mean and variance $\{\tilde{\boldsymbol{\mu}}, \tilde{\boldsymbol{\sigma}}^2\}$ and obtain the aggregated posterior distribution by Eq. 9.
6:     Compute the cluster assignment posterior $q_\phi(\mathbf{c}|\{\mathbf{x}_v\}_{v\in\mathcal{V}})$ by Eq. 12.
7:     Compute $\{p_{\boldsymbol{\theta}_v}(\mathbf{x}_v \mid \mathbf{z}, \mathbf{c})\}_{v\in\mathcal{V}}$ using the view-specific decoders with reparameterization.
8:     Update $\{\boldsymbol{\phi}_v, \boldsymbol{\theta}_v\}, \{\boldsymbol{\tau}_c, \boldsymbol{\mu}_c, \boldsymbol{\Sigma}_c^2\}$ and $\mathbf{R}$ by optimizing Eq. 15.
9: **end while**
10: Calculate the clustering assignment of each sample by Eq. 12 and predict the cluster by the max probability.

---

## A.8. Complexity Analysis

In our framework, the primary computational operations include feature encoding, cross-view aggregation, and posterior inference. Let $N$ be the number of samples, $V$ the number of views. For simplicity, we assume that all views are mapped to a common latent dimension $D$ after encoding, regardless of their original dimensionality $d_v$. The view-specific feature encoding process thus takes $\mathcal{O}(NVD)$. The cross-view correlation modeling, which is the core of our method, involves two steps: (1) constructing $V \times V$ covariance matrices for each sample and latent dimension with complexity $\mathcal{O}(NDV^2)$, and (2) performing matrix inversion on these covariance matrices, which dominates the computational cost with $\mathcal{O}(NDV^3)$ when computing the inverse explicitly, but can be reduced to $\mathcal{O}(NDV^2)$ by exploiting the Cholesky factorization and performing forward and backward substitutions instead of explicitly computing matrix inverses (see Appendix A.6). Besides, other reconstruction and KL divergence terms in posterior inference cost $\mathcal{O}(NVD)$. Therefore, the overall time complexity for our naive implementation is $\mathcal{O}(NDV^3)$ but can be optimized to $\mathcal{O}(NDV^2)$. Since $V$ is usually small (*e.g.*, $V < 10$ and $V \ll N$), our method remains highly efficient in practical incomplete multi-view clustering scenarios, scaling linearly with the number of samples $N$. See Appendix C.4 for an empirical comparison of run-times.

## A.9. Discussion on Information Imbalance in Aggregation

In line with standard VAE training under partial observations, we learn a shared latent z for each sample using only its observed views. Missing modalities are not fed to encoders and are not treated as observed evidence in the ELBO. More specifically, the approximate posterior $q_\phi(\mathbf{z} \mid \{\mathbf{x}_v\}_{v \in \mathcal{V}})$ is obtained through an aggregation mechanism combining the available view-specific posteriors $q_{\phi_v}(\mathbf{z} \mid \mathbf{x}_v)$. This paradigm is shared by IMVC methods such as DVIMC, which in this way learn a unified representation despite missing views.

Here we provide a concrete analysis of how the missing-view mask $\mathbf{M}$ impacts our proposed aggregation through $q(\mathbf{z} \mid \{\mathbf{x}_v\}_{v \in \mathcal{V}}) \sim \mathcal{N}(\tilde{\boldsymbol{\mu}}, \tilde{\boldsymbol{\sigma}}^2) = \mathcal{N}(\mathbf{A}_\mathbf{M}^{-1}\mathbf{B}_\mathbf{M}, \mathbf{A}_\mathbf{M}^{-1})$ where $\mathbf{A}_\mathbf{M} = \mathbb{1}^\top(\boldsymbol{\Sigma}^{-1} \odot \mathbf{M}\mathbf{M}^\top)\mathbb{1}$ and $\mathbf{B}_\mathbf{M} = \mathbb{1}^\top(\boldsymbol{\Sigma}^{-1} \odot \mathbf{M}\mathbf{M}^\top)(\boldsymbol{\mu} \odot \mathbf{M})$. For simplicity, here we consider a single latent dimension, where this reduces to the quadratic form $\mathbf{A}_\mathbf{M} = \mathbf{M}^\top\boldsymbol{\Sigma}^{-1}\mathbf{M}$. Fewer observed views (more zeros in $\mathbf{M}$) remove corresponding rows/columns from this quadratic form, so $\mathbf{A}_\mathbf{M}$ cannot increase when views are missing. This means that if there are fewer views, the missing-view mask ensures that $\mathbf{A}_\mathbf{M}$ will be smaller, leading to larger posterior variance, explicitly preventing overconfidence when evidence is scarce. This effectively addresses the problem that samples with fewer views can lead to overconfidence and ensures that the information imbalance is taken into account (more views, more information, more confident).

Another factor of information imbalance is related to the quality of the views (how redundant/noisy). To address this, each encoder produces $\sigma_v^2$, such that a view with higher uncertainty contributes less to $\mathbf{A}_\mathbf{M}$ and $\mathbf{B}_\mathbf{M}$ as $\boldsymbol{\Sigma} = \mathbf{D}\mathbf{R}\mathbf{D}$, with $\mathbf{D} = \mathrm{diag}(\sigma_1, \ldots, \sigma_V)$. This reduces the influence of weak/noisy views and mitigates imbalance caused by heterogeneous view quality. Finally, the learned cross-view correlations address one key problem of PoE, where PoE sums precisions and becomes overconfident. We instead avoid this problem, where similar or duplicate views outvote single distinct views by accounting for dependency, yielding calibrated results.

Here we consider a simple example in which we have two views $x_1$ and $x_2$, and the true joint posterior variable is $z = 8$. The two view-specific encoders estimate $\mu_1 = 4$ and $\mu_2 = 8$, with uncertainties $\sigma_1^2 = 3$ and $\sigma_2^2 = 1$, and correlation $\rho_{12} = 0.55$. Following (Winkler, 1981), the expectation of the joint posterior distribution (the estimate of $z$) can be written as the weighted average of the view-specific estimates $\tilde{\mu} = \omega_1\mu_1 + \omega_2\mu_2$, where the weights are $\omega_1 = (\sigma_2^2 - \rho_{12}\sigma_1\sigma_2)/(\sigma_1^2 + \sigma_2^2 - 2\rho_{12}\sigma_1\sigma_2)$ and $\omega_2 = (\sigma_1^2 - \rho_{12}\sigma_1\sigma_2)/(\sigma_1^2 + \sigma_2^2 - 2\rho_{12}\sigma_1\sigma_2)$. Therefore, the estimate of the joint posterior distribution according to our proposed aggregation method is $\tilde{\mu} = 0.02 \times 4 + 0.98 \times 8$, leaning towards the relatively more accurate and less uncertain estimate. In contrast, neglecting the correlation between view-specific encoders underestimates the true joint posterior variable as $\tilde{\mu} = 0.25 \times 4 + 0.75 \times 8$.

# B. Experiment Settings

## B.1. Statistics for Datasets

In this section, we provide details of the six real-world datasets used in the experiment in Table 3.

*Table 3.* Detailed statistics for the six multi-view clustering datasets used in our experiments.

| Dataset | #Views | #View Dimensions | #Instances | #Categories |
|---|---|---|---|---|
| Handwritten (LeCun et al., 1989) | 6 | 76, 216, 64, 47, 240, 6 | 2,000 | 10 |
| Caltech5V (Fei-Fei et al., 2004) | 5 | 40, 254, 1984, 512, 928 | 1,400 | 7 |
| Scene15 (Fei-Fei & Perona, 2005) | 3 | 20, 59, 40 | 4,485 | 15 |
| Fashion (Xiao et al., 2017) | 3 | 784, 784, 784 | 10,000 | 10 |
| NoisyMNIST (Wang et al., 2015) | 2 | 784, 784 | 70,000 | 10 |
| CUB Image-Captions (Zhang et al., 2019) | 2 | 1024, 300 | 600 | 10 |

*Table 4.* Statistics and brief summary of eight comparison methods.

| Method | Venue | #View | Description |
|---|---|---|---|
| BSV (Zhao et al., 2016) | IJCAI | Multiple | Imputes missing views using the view-wise average values and performs clustering using the best-performing single view. |
| CONCAT (Zhao et al., 2016) | IJCAI | Multiple | Similar to BSV that imputes missing views using the view-wise average values, but concatenates features across all views after imputation and then performs $K$-Means clustering. |
| DSIMVC (Tang & Liu, 2022) | ICML | Multiple | A bi-level optimization-based safe incomplete multi-view clustering framework that dynamically imputes missing views from semantic neighbors and automatically selects reliable imputed samples. |
| DCP (Lin et al., 2023) | PAMI | Two | An information-theoretical framework that unifies cross-view consistency learning and missing-view recovery via dual contrastive learning and dual prediction. |
| CPSPAN (Jin et al., 2023) | CVPR | Multiple | Jointly performs cross-view partial sample alignment and prototype-level alignment to address the challenges of prototype shift under missing-view settings. |
| DVIMC (Xu et al., 2024) | AAAI | Multiple | Employ PoE-based VAEs to construct a shared latent space and mitigate information imbalance via a coherence loss. |
| MVP (Gao & Pu, 2025) | ICLR | Multiple | A variational framework that explicitly infers missing views in latent space and enhances consistency via cyclic permutation-based regularization. |
| PMIMC (Yuan et al., 2025) | TIP | Multiple | An IMVC method that solves the prototype-unaligned problem and performance instability problem by prototype contrastive learning loss and prototype-based imputation strategies. |

### B.2. Incomplete Multi-view Data Generation

Following prior IMVC works (Xu et al., 2024; Tang & Liu, 2022), we generate incomplete multi-view data by randomly masking a proportion $p\%$ of the samples. Formally, given a dataset with $N$ samples and $V$ views, we define a binary mask matrix $\mathcal{M} \in \{0, 1\}^{N \times V}$, where $\mathcal{M}_{i,v} = 1$ if the $i$-th sample is observed in view $v$, and $\mathcal{M}_{i,v} = 0$ otherwise. The generation process is as follows: (1) randomly select $p\%$ of the samples as incomplete; (2) for each selected sample, randomly assign a non-empty proper subset of views to be retained, ensuring that at least one view is observed and at least one is missing; (3) all remaining $(1 - p)\%$ of the samples are kept fully observed with $\mathcal{M}_{i,v} = 1$ for all $v \in \{1, \ldots, V\}$. For example, in a three-view setting with views $\{A, B, C\}$, incomplete samples may have only one view (e.g., $A$, $B$, or $C$) or two views (e.g., $\{A, B\}$, $\{B, C\}$, or $\{A, C\}$).

### B.3. Comparison Methods

In this section, we provide details on the eight comparison approaches that are considered in the experiments (see Table 4).

*Table 5.* Hyperparameter settings for all datasets. "ExpLR" is short for ExponentialLR.

| Hyperparameters | Handwritten | Caltech5V | Scene15 | Fashion | NoisyMNIST | CUB |
|---|---|---|---|---|---|---|
| Pretraining Epochs | 200 | 200 | 200 | 200 | 200 | 200 |
| Training Epochs | 300 | 300 | 300 | 300 | 300 | 300 |
| Batch Size | 256 | 256 | 256 | 256 | 512 | 256 |
| Optimizer | Adam | Adam | Adam | Adam | Adam | Adam |
| Scheduler | ExpLR | ExpLR | ExpLR | ExpLR | ExpLR | ExpLR |
| Network Learning Rate | $3e^{-4}$ | $1e^{-3}$ | $1e^{-3}$ | $1e^{-3}$ | $1e^{-3}$ | $5e^{-4}$ |
| Prior Distribution Learning Rate | $1e^{-2}$ | $1e^{-1}$ | $5e^{-2}$ | $5e^{-2}$ | $5e^{-2}$ | $5e^{-2}$ |
| Correlation Learning Rate | $1e^{-2}$ | $1e^{-2}$ | $1e^{-2}$ | $1e^{-2}$ | $1e^{-2}$ | $1e^{-2}$ |
| Balance parameter $\alpha$ | 15 | 5 | 20 | 20 | 10 | 60 |
| Latent feature dimension $D$ | 10 | 15 | 10 | 10 | 10 | 10 |

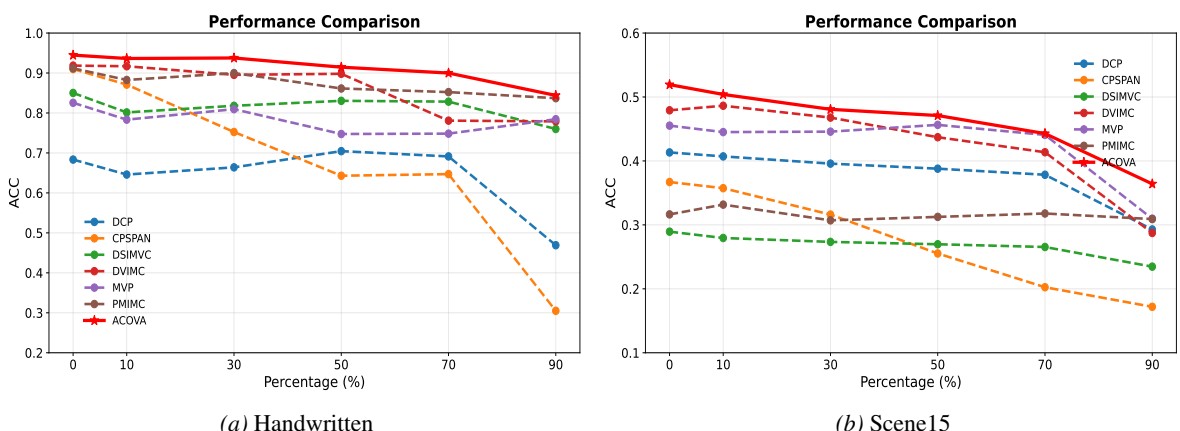

*(a)* Handwritten         *(b)* Scene15

*Figure 6.* Performance curves vs. missing rates. The proposed approach consistently outperforms prior approaches even in the extreme scenarios.

## B.4. Implementation Details

All experiments are conducted on the PyTorch platform, running on a Linux server equipped with an Intel(R) Xeon(R) Gold 5218R CPU @ 2.10GHz and an NVIDIA GeForce RTX 3090 GPU. For fair comparison, we run five experimental trials for each method and report the average results. All compared methods are implemented with their recommended hyperparameter settings or through a parameter search for better performance. For network architecture, each view-specific encoder consists of three fully-connected layers with dimensions [500, 500, 2,000], followed by ReLU activation functions, while the dimensions of the decoder are reversed. The hyperparameters for the six datasets are listed in Table 5.

## C. Additional Experimental Results

### C.1. Complete Results on the Six Datasets

In Table 6, we report the complete results, including the mean and standard deviation over five runs for all methods across the six datasets. In Table 7 and Table 8, we also report the complete results of the ablation study on the Handwritten and Scene15 datasets, respectively.

### C.2. Discussion on Extreme Scenarios

To further evaluate the robustness of our proposed approach, we also consider more extreme missing-rate scenarios. Fig. 6 provides performance curves *vs.* missing-rate for an extended range of [0,90] for the Handwritten and Scene15 datasets. Results demonstrate that the proposed approach consistently outperforms the baselines even in the two extreme cases (missing rate of 90% and of 0% (complete)), further supporting the need to go beyond the independence assumption.

*Table 6.* Comparison of clustering performance on the benchmark datasets under different missing-view rates (MR) of 10%, 30%, 50%, 70%. The best results are highlighted in **bold**, while the second-best are underlined. Each experiment is conducted five times, with the mean and standard deviation reported.

**Scene15**

| MR | 10% | | | | 30% | | | | 50% | | | | 70% | | | |
|---|---|---|---|---|---|---|---|---|---|---|---|---|---|---|---|---|
| Metrics | ACC↑ | NMI↑ | ARI↑ | PUR↑ | ACC↑ | NMI↑ | ARI↑ | PUR↑ | ACC↑ | NMI↑ | ARI↑ | PUR↑ | ACC↑ | NMI↑ | ARI↑ | PUR↑ |
| BSV | 0.3595±0.0061 | 0.3577±0.0054 | 0.1784±0.041 | 0.4002±0.0092 | 0.3305±0.0086 | 0.3206±0.0071 | 0.1253±0.0051 | 0.3630±0.0097 | 0.2944±0.0100 | 0.2905±0.0063 | 0.0796±0.0046 | 0.3321±0.0132 | 0.2593±0.0085 | 0.2568±0.0030 | 0.0472±0.0015 | 0.2903±0.0029 |
| Concat | 0.3703±0.0038 | 0.3820±0.0022 | 0.2037±0.0024 | 0.4263±0.0031 | 0.3360±0.0067 | 0.3318±0.0035 | 0.1540±0.0036 | 0.3825±0.0059 | 0.2943±0.0139 | 0.2995±0.0083 | 0.1201±0.0057 | 0.3436±0.0089 | 0.2596±0.0137 | 0.2665±0.0020 | 0.0982±0.0042 | 0.3053±0.0084 |
| DCP | 0.4071±0.0109 | 0.4414±0.0098 | 0.2488±0.0110 | 0.4342±0.0158 | 0.3958±0.0072 | 0.4264±0.0052 | 0.2350±0.0062 | 0.4265±0.0084 | 0.3879±0.0216 | 0.4095±0.0101 | 0.2359±0.0063 | 0.4166±0.0161 | 0.3784±0.0084 | 0.3849±0.0125 | 0.2113±0.0113 | 0.4026±0.0100 |
| CPSPAN | 0.3574±0.0201 | 0.3457±0.0148 | 0.1911±0.0142 | 0.4103±0.0188 | 0.3161±0.0137 | 0.2783±0.0118 | 0.1541±0.0076 | 0.3439±0.0173 | 0.2553±0.0071 | 0.2070±0.0076 | 0.1063±0.0049 | 0.2786±0.0058 | 0.2025±0.0117 | 0.1423±0.0138 | 0.0621±0.0101 | 0.2198±0.0126 |
| DSIMVC | 0.2795±0.0142 | 0.2976±0.0107 | 0.1430±0.0063 | 0.3218±0.0110 | 0.2734±0.0111 | 0.2896±0.0087 | 0.1383±0.0052 | 0.3131±0.0067 | 0.2697±0.0059 | 0.2828±0.0069 | 0.1355±0.0043 | 0.3133±0.0074 | 0.2654±0.0090 | 0.2741±0.0082 | 0.1290±0.0067 | 0.3062±0.0079 |
| DVIMC | 0.4863±0.0103 | 0.4780±0.0059 | 0.3192±0.0085 | 0.5021±0.0109 | 0.4678±0.0054 | 0.4440±0.0033 | 0.3097±0.0089 | 0.4823±0.0089 | 0.4371±0.0568 | 0.4163±0.0402 | 0.2737±0.0399 | 0.4577±0.0654 | 0.4136±0.0266 | 0.3950±0.0228 | 0.2519±0.0189 | 0.4286±0.0293 |
| MVP | 0.4451±0.0188 | 0.4361±0.0112 | 0.2696±0.0126 | 0.4830±0.0161 | 0.4458±0.0064 | 0.4341±0.0061 | 0.2723±0.0076 | 0.4866±0.0099 | 0.4564±0.0068 | 0.4314±0.0074 | 0.2967±0.0095 | 0.5170±0.0130 | 0.4407±0.0163 | 0.4285±0.0091 | 0.2725±0.0117 | 0.4895±0.0171 |
| PMIMC | 0.3317±0.0032 | 0.3409±0.0053 | 0.1778±0.0032 | 0.3602±0.0073 | 0.3071±0.0118 | 0.3175±0.0028 | 0.1484±0.0036 | 0.3500±0.0086 | 0.3125±0.0000 | 0.3180±0.0014 | 0.1477±0.0026 | 0.3513±0.0061 | 0.3178±0.0071 | 0.3173±0.0041 | 0.1492±0.0036 | 0.3509±0.0064 |
| ACOVA (Ours) | 0.5040±0.0153 | 0.4804±0.0050 | 0.3273±0.0088 | 0.5276±0.0202 | 0.4807±0.0136 | 0.4560±0.0090 | 0.3106±0.0090 | 0.5119±0.0164 | 0.4710±0.0138 | 0.4360±0.0034 | 0.2973±0.0084 | 0.4987±0.0126 | 0.4428±0.0140 | 0.4083±0.0055 | 0.2733±0.0031 | 0.4636±0.0161 |

**Caltech5V**

| Method | ACC↑ | NMI↑ | ARI↑ | PUR↑ | ACC↑ | NMI↑ | ARI↑ | PUR↑ | ACC↑ | NMI↑ | ARI↑ | PUR↑ | ACC↑ | NMI↑ | ARI↑ | PUR↑ |
|---|---|---|---|---|---|---|---|---|---|---|---|---|---|---|---|---|
| BSV | 0.5678±0.0051 | 0.4364±0.0045 | 0.3408±0.0046 | 0.5827±0.0050 | 0.5035±0.0086 | 0.3699±0.0049 | 0.2308±0.0032 | 0.5170±0.0064 | 0.4483±0.0062 | 0.3235±0.0046 | 0.1488±0.0027 | 0.4617±0.0053 | 0.3859±0.0043 | 0.2798±0.0055 | 0.0890±0.0032 | 0.4019±0.0062 |
| Concat | 0.4750±0.0337 | 0.3300±0.0136 | 0.2519±0.0232 | 0.4957±0.0186 | 0.4450±0.0128 | 0.2941±0.0076 | 0.1998±0.0077 | 0.4593±0.0091 | 0.4279±0.0140 | 0.2756±0.0133 | 0.1595±0.0081 | 0.4386±0.0132 | 0.3600±0.0241 | 0.2338±0.0201 | 0.0891±0.0110 | 0.3757±0.0172 |
| DCP | 0.5754±0.0187 | 0.5719±0.0175 | 0.4612±0.0339 | 0.6054±0.0171 | 0.5363±0.0296 | 0.5269±0.0132 | 0.4199±0.0212 | 0.5751±0.0113 | 0.5163±0.0596 | 0.4729±0.0255 | 0.3406±0.0454 | 0.5393±0.0454 | 0.3982±0.0289 | 0.3362±0.0156 | 0.1378±0.0234 | 0.4056±0.0276 |
| CPSPAN | 0.7689±0.0185 | 0.6677±0.0221 | 0.6088±0.0298 | 0.7750±0.0176 | 0.6874±0.0359 | 0.5653±0.0265 | 0.4666±0.0310 | 0.6888±0.0343 | 0.6010±0.0249 | 0.4789±0.0343 | 0.3467±0.0184 | 0.6030±0.0249 | 0.4861±0.0389 | 0.3664±0.0329 | 0.2222±0.0309 | 0.4904±0.0373 |
| DSIMVC | 0.7760±0.0536 | 0.6998±0.0245 | 0.6280±0.0395 | 0.7816±0.0440 | 0.7841±0.0261 | 0.6947±0.0270 | 0.6335±0.0289 | 0.7841±0.0261 | 0.7339±0.0370 | 0.6868±0.0088 | 0.5989±0.0172 | 0.7413±0.0270 | 0.7029±0.0193 | 0.5838±0.0212 | 0.5083±0.0227 | 0.7047±0.0162 |
| DVIMC | 0.9013±0.0103 | 0.8207±0.0157 | 0.7999±0.0182 | 0.9013±0.0103 | 0.8921±0.0093 | 0.8011±0.0166 | 0.7850±0.0170 | 0.8921±0.0093 | 0.8609±0.0123 | 0.7507±0.0186 | 0.7322±0.0186 | 0.8609±0.0123 | 0.8514±0.0050 | 0.7345±0.0103 | 0.7195±0.0080 | 0.8514±0.0050 |
| MVP | 0.8083±0.0148 | 0.6969±0.0212 | 0.6594±0.0242 | 0.8083±0.0148 | 0.8070±0.0213 | 0.7139±0.0208 | 0.6723±0.0208 | 0.8146±0.0132 | 0.8103±0.0231 | 0.7063±0.0235 | 0.6700±0.0283 | 0.8157±0.0184 | 0.7677±0.0354 | 0.6483±0.0333 | 0.5934±0.0432 | 0.7693±0.0337 |
| PMIMC | 0.8963±0.0120 | 0.8205±0.0171 | 0.7996±0.0251 | 0.8963±0.0120 | 0.8917±0.0128 | 0.8124±0.0171 | 0.7858±0.0251 | 0.8917±0.0128 | 0.8601±0.0071 | 0.7591±0.0137 | 0.7381±0.0141 | 0.8601±0.0071 | 0.8570±0.0119 | 0.7522±0.0092 | 0.7304±0.0141 | 0.8570±0.0119 |
| ACOVA (Ours) | 0.9149±0.0129 | 0.8507±0.0190 | 0.8287±0.0237 | 0.9149±0.0129 | 0.9131±0.0186 | 0.8435±0.0233 | 0.8226±0.0334 | 0.9131±0.0186 | 0.8953±0.0151 | 0.8078±0.0230 | 0.7918±0.0270 | 0.8953±0.0151 | 0.8660±0.0075 | 0.7531±0.0146 | 0.7340±0.0146 | 0.8660±0.0075 |

**Handwritten**

| Method | ACC↑ | NMI↑ | ARI↑ | PUR↑ | ACC↑ | NMI↑ | ARI↑ | PUR↑ | ACC↑ | NMI↑ | ARI↑ | PUR↑ | ACC↑ | NMI↑ | ARI↑ | PUR↑ |
|---|---|---|---|---|---|---|---|---|---|---|---|---|---|---|---|---|
| BSV | 0.7087±0.0113 | 0.6647±0.0036 | 0.5625±0.0083 | 0.7433±0.0022 | 0.7105±0.0435 | 0.6344±0.0336 | 0.4716±0.0336 | 0.7171±0.0368 | 0.5842±0.0214 | 0.5444±0.0145 | 0.3028±0.0193 | 0.5970±0.0182 | — | — | — | — |
| Concat | 0.7684±0.0337 | 0.7214±0.0112 | 0.6412±0.0149 | 0.7837±0.0174 | 0.7147±0.0202 | 0.6441±0.0094 | 0.5012±0.0131 | 0.7190±0.0120 | 0.6108±0.0596 | 0.5437±0.0466 | 0.3492±0.0172 | 0.6166±0.0524 | — | — | — | — |
| DCP | 0.6459±0.0434 | 0.7094±0.0154 | 0.5663±0.0251 | 0.6737±0.0296 | 0.6639±0.0462 | 0.6951±0.0270 | 0.5189±0.0638 | 0.6877±0.0357 | 0.7045±0.0411 | 0.7018±0.0218 | 0.5651±0.0397 | 0.7334±0.0354 | — | — | — | — |
| CPSPAN | 0.8710±0.0485 | 0.7969±0.0267 | 0.7586±0.0462 | 0.8727±0.0452 | 0.7524±0.0423 | 0.6927±0.0183 | 0.5603±0.0301 | 0.7638±0.0327 | 0.6430±0.0415 | 0.6050±0.0145 | 0.3862±0.0130 | 0.6595±0.0331 | — | — | — | — |
| DSIMVC | 0.8014±0.0429 | 0.7960±0.0355 | 0.7282±0.0471 | 0.8158±0.0301 | 0.8180±0.0466 | 0.8082±0.0427 | 0.7455±0.0570 | 0.8275±0.0352 | 0.8304±0.0327 | 0.8122±0.0256 | 0.7543±0.0358 | 0.8342±0.0252 | — | — | — | — |
| DVIMC | 0.9169±0.0545 | 0.8604±0.0268 | 0.8547±0.0514 | 0.9201±0.0481 | 0.8954±0.0515 | 0.8470±0.0336 | 0.8210±0.0527 | 0.9020±0.0409 | 0.8980±0.0486 | 0.8373±0.0199 | 0.8175±0.0397 | 0.9054±0.0340 | — | — | — | — |
| MVP | 0.7833±0.0440 | 0.8002±0.0174 | 0.7114±0.0256 | 0.8119±0.0297 | 0.8095±0.0169 | 0.8000±0.0267 | 0.7270±0.0265 | 0.8208±0.0117 | 0.7474±0.0597 | 0.7853±0.0111 | 0.6834±0.0371 | 0.7715±0.0408 | — | — | — | — |
| PMIMC | 0.8826±0.0367 | 0.8188±0.0132 | 0.7846±0.0266 | 0.8834±0.0352 | 0.8998±0.0115 | 0.8213±0.0129 | 0.7962±0.0194 | 0.8998±0.0115 | 0.8613±0.0557 | 0.8096±0.0254 | 0.7637±0.0519 | 0.8695±0.0427 | — | — | — | — |
| ACOVA (Ours) | 0.9363±0.0067 | 0.8692±0.0113 | 0.8645±0.0133 | 0.9363±0.0067 | 0.9376±0.0073 | 0.8709±0.0125 | 0.8669±0.0144 | 0.9376±0.0073 | 0.9146±0.0321 | 0.8489±0.0280 | 0.8325±0.0453 | 0.9146±0.0321 | — | — | — | — |

**Fashion**

| Method | ACC↑ | NMI↑ | ARI↑ | PUR↑ | ACC↑ | NMI↑ | ARI↑ | PUR↑ | ACC↑ | NMI↑ | ARI↑ | PUR↑ | ACC↑ | NMI↑ | ARI↑ | PUR↑ |
|---|---|---|---|---|---|---|---|---|---|---|---|---|---|---|---|---|
| BSV | 0.4324±0.0163 | 0.4606±0.0100 | 0.2986±0.0076 | 0.5210±0.0113 | 0.4424±0.0167 | 0.4114±0.0080 | 0.2210±0.0048 | 0.4779±0.0094 | 0.4100±0.0245 | 0.3734±0.0074 | 0.1626±0.0050 | 0.4361±0.0121 | — | — | — | — |
| Concat | 0.6546±0.0575 | 0.6518±0.0254 | 0.5451±0.0369 | 0.7028±0.0499 | 0.5002±0.0302 | 0.4618±0.0177 | 0.3646±0.0198 | 0.5272±0.0214 | 0.4618±0.0177 | 0.3433±0.0120 | 0.2509±0.0114 | 0.4350±0.0089 | — | — | — | — |
| DCP | 0.8547±0.0618 | 0.8866±0.0188 | 0.8119±0.0519 | 0.8652±0.0547 | 0.7663±0.0494 | 0.8340±0.0177 | 0.7285±0.0371 | 0.7899±0.0473 | 0.7533±0.0497 | 0.7995±0.0166 | 0.6904±0.0356 | 0.7643±0.0461 | — | — | — | — |
| CPSPAN | 0.7130±0.0547 | 0.7566±0.0345 | 0.6303±0.0415 | 0.7548±0.0517 | 0.6981±0.0457 | 0.7524±0.0342 | 0.6179±0.0342 | 0.7446±0.0431 | 0.6886±0.0784 | 0.7527±0.0378 | 0.6237±0.0473 | 0.7436±0.0686 | — | — | — | — |
| DSIMVC | 0.8993±0.0236 | 0.8573±0.0152 | 0.8178±0.0317 | 0.8993±0.0236 | 0.8810±0.0298 | 0.8162±0.0374 | 0.8236±0.0372 | 0.8810±0.0298 | 0.8342±0.0374 | 0.8070±0.0048 | 0.7380±0.0183 | 0.8359±0.0256 | — | — | — | — |
| DVIMC | 0.8856±0.0541 | 0.8799±0.0126 | 0.8284±0.0444 | 0.9024±0.0337 | 0.8806±0.0461 | 0.8718±0.0098 | 0.8537±0.0021 | 0.8986±0.0301 | 0.8629±0.0627 | 0.8449±0.0217 | 0.7965±0.0482 | 0.8867±0.0505 | — | — | — | — |
| MVP | 0.8691±0.0550 | 0.8743±0.0175 | 0.8175±0.0473 | 0.8955±0.0339 | 0.8259±0.0025 | 0.8537±0.0029 | 0.7744±0.0029 | 0.8722±0.0016 | 0.7982±0.0441 | 0.8372±0.0155 | 0.7484±0.0338 | 0.8485±0.0384 | — | — | — | — |
| PMIMC | 0.6986±0.0797 | 0.7563±0.0237 | 0.6282±0.0535 | 0.7427±0.0540 | 0.7039±0.0319 | 0.7615±0.0175 | 0.6365±0.0268 | 0.7535±0.0300 | 0.7118±0.0368 | 0.7586±0.0208 | 0.6295±0.0321 | 0.7577±0.0373 | — | — | — | — |
| ACOVA (Ours) | 0.9056±0.0627 | 0.8876±0.0208 | 0.8516±0.0494 | 0.9129±0.0481 | 0.8848±0.0562 | 0.8735±0.0141 | 0.8266±0.0460 | 0.9009±0.0365 | 0.8793±0.0395 | 0.8479±0.0151 | 0.8002±0.0378 | 0.8794±0.0394 | — | — | — | — |

**NoisyMNIST**

| Method | ACC↑ | NMI↑ | ARI↑ | PUR↑ | ACC↑ | NMI↑ | ARI↑ | PUR↑ | ACC↑ | NMI↑ | ARI↑ | PUR↑ | ACC↑ | NMI↑ | ARI↑ | PUR↑ |
|---|---|---|---|---|---|---|---|---|---|---|---|---|---|---|---|---|
| BSV | 0.5144±0.0037 | 0.4542±0.0087 | 0.3258±0.0035 | 0.5618±0.0047 | 0.4184±0.0645 | 0.3343±0.0899 | 0.1896±0.0604 | 0.4456±0.0837 | 0.2939±0.0051 | 0.1912±0.0024 | 0.0754±0.0022 | 0.3090±0.0043 | 0.2672±0.0048 | 0.1566±0.0031 | 0.0452±0.0007 | 0.2693±0.0056 |
| Concat | 0.4338±0.0016 | 0.3910±0.0012 | 0.2727±0.0009 | 0.4418±0.0009 | 0.3873±0.0057 | 0.3229±0.0065 | 0.1891±0.0047 | 0.4036±0.0022 | 0.3463±0.0210 | 0.2592±0.0245 | 0.1252±0.0278 | 0.3561±0.0187 | 0.2805±0.0248 | 0.1832±0.0278 | 0.0615±0.0299 | 0.2864±0.0216 |
| DCP | 0.8945±0.0663 | 0.9149±0.0355 | 0.8786±0.0749 | 0.9319±0.0439 | 0.8997±0.0755 | 0.8952±0.0316 | 0.8608±0.0845 | 0.9198±0.0522 | 0.9150±0.0286 | 0.8586±0.0222 | 0.8401±0.0496 | 0.9150±0.0286 | 0.9225±0.0185 | 0.8335±0.0127 | 0.8431±0.0274 | 0.9225±0.0185 |
| CPSPAN | 0.5665±0.0149 | 0.5870±0.0250 | 0.4402±0.0277 | 0.6447±0.0146 | 0.6103±0.0311 | 0.6050±0.0193 | 0.4724±0.0295 | 0.6357±0.0180 | 0.5774±0.0170 | 0.5824±0.0231 | 0.4439±0.0201 | 0.6097±0.0166 | 0.5486±0.0166 | 0.5766±0.0185 | 0.4271±0.0291 | 0.5932±0.0213 |
| DSIMVC | 0.8207±0.0963 | 0.7735±0.0826 | 0.7276±0.1157 | 0.8327±0.0843 | 0.7337±0.1270 | 0.6790±0.1017 | 0.6148±0.1211 | 0.7464±0.1211 | 0.6152±0.1458 | 0.5769±0.1145 | 0.4848±0.1589 | 0.6361±0.1331 | 0.6245±0.0276 | 0.5822±0.0283 | 0.4867±0.0321 | 0.6343±0.0347 |
| DVIMC | 0.9372±0.0541 | 0.9345±0.0121 | 0.9157±0.0533 | 0.9466±0.0123 | 0.9365±0.0395 | 0.9060±0.0174 | 0.8987±0.0409 | 0.9368±0.0390 | 0.8899±0.0479 | 0.8699±0.0232 | 0.8538±0.0469 | 0.9019±0.0460 | 0.8741±0.0562 | 0.8184±0.0560 | 0.7999±0.0560 | 0.8763±0.0520 |
| MVP | 0.8657±0.0344 | 0.8778±0.0383 | 0.7864±0.0571 | 0.8667±0.0428 | 0.8590±0.1030 | 0.8635±0.0521 | 0.8222±0.1028 | 0.8777±0.0821 | 0.8355±0.0337 | 0.8195±0.0318 | 0.7792±0.0407 | 0.8464±0.0321 | 0.6660±0.0712 | 0.6744±0.1043 | 0.5721±0.1088 | 0.6694±0.0713 |
| PMIMC | 0.7127±0.0953 | 0.6985±0.0616 | 0.5927±0.0961 | 0.7432±0.0848 | 0.7187±0.0147 | 0.6593±0.0250 | 0.5623±0.0119 | 0.7458±0.0163 | 0.6616±0.0362 | 0.5739±0.0194 | 0.4947±0.0240 | 0.6695±0.0301 | 0.6044±0.0243 | 0.5344±0.0253 | 0.4393±0.0259 | 0.6220±0.0233 |
| ACOVA (Ours) | 0.9663±0.0393 | 0.9478±0.0240 | 0.9440±0.0508 | 0.9664±0.0393 | 0.9344±0.0471 | 0.9125±0.0195 | 0.9016±0.0195 | 0.9348±0.0465 | 0.9599±0.0010 | 0.9001±0.0023 | 0.9144±0.0020 | 0.9599±0.0010 | 0.9377±0.0080 | 0.8579±0.0111 | 0.8697±0.0152 | 0.9377±0.0080 |

**CUB Image-Caption**

| Method | ACC↑ | NMI↑ | ARI↑ | PUR↑ | ACC↑ | NMI↑ | ARI↑ | PUR↑ | ACC↑ | NMI↑ | ARI↑ | PUR↑ | ACC↑ | NMI↑ | ARI↑ | PUR↑ |
|---|---|---|---|---|---|---|---|---|---|---|---|---|---|---|---|---|
| BSV | 0.6723±0.0381 | 0.6604±0.0149 | 0.4818±0.0237 | 0.6823±0.0247 | 0.6080±0.0294 | 0.5858±0.0118 | 0.3376±0.0252 | 0.6163±0.0208 | 0.5270±0.0188 | 0.5241±0.0062 | 0.2295±0.0102 | 0.5387±0.0193 | 0.4493±0.0407 | 0.4474±0.0243 | 0.1266±0.0166 | 0.4630±0.0335 |
| Concat | 0.6890±0.0291 | 0.6947±0.0201 | 0.4979±0.0228 | 0.6947±0.0212 | 0.5837±0.0370 | 0.5790±0.0427 | 0.3322±0.0400 | 0.5987±0.0269 | 0.5187±0.0295 | 0.5107±0.0187 | 0.2068±0.0102 | 0.5323±0.0157 | 0.4663±0.0214 | 0.4658±0.0146 | 0.1391±0.0157 | 0.4770±0.0234 |
| DCP | 0.5227±0.0296 | 0.5822±0.0087 | 0.2867±0.0139 | 0.5833±0.0118 | 0.5163±0.0580 | 0.5781±0.0371 | 0.3008±0.0559 | 0.5797±0.0533 | 0.4040±0.0235 | 0.4430±0.0180 | 0.1196±0.0180 | 0.4587±0.0202 | 0.2653±0.0078 | 0.2337±0.0201 | 0.0571±0.0080 | 0.2867±0.0166 |
| CPSPAN | 0.6973±0.0206 | 0.6766±0.0101 | 0.5420±0.0104 | 0.7153±0.0264 | 0.6067±0.0171 | 0.5914±0.0124 | 0.4153±0.0125 | 0.6160±0.0237 | 0.5407±0.0100 | 0.5198±0.0088 | 0.2762±0.0093 | 0.5487±0.0040 | 0.4957±0.0077 | 0.4797±0.0101 | 0.1953±0.0203 | 0.5033±0.0064 |
| DSIMVC | 0.6380±0.0188 | 0.5744±0.0157 | 0.4280±0.0159 | 0.6427±0.0146 | 0.6420±0.0105 | 0.5663±0.0085 | 0.4236±0.0102 | 0.6420±0.0105 | 0.5419±0.0287 | 0.5419±0.0287 | 0.3824±0.0193 | 0.5419±0.0287 | 0.5410±0.0573 | 0.4965±0.0419 | 0.3275±0.0544 | 0.5440±0.0541 |
| DVIMC | 0.6573±0.0427 | 0.6585±0.0266 | 0.5246±0.0366 | 0.6807±0.0383 | 0.6703±0.0372 | 0.6502±0.0409 | 0.5131±0.0448 | 0.6807±0.0334 | 0.6057±0.0341 | 0.6026±0.0252 | 0.4538±0.0280 | 0.6160±0.0233 | 0.4710±0.0438 | 0.4694±0.0452 | 0.2795±0.0477 | 0.4803±0.0489 |
| MVP | 0.6497±0.0246 | 0.6317±0.0154 | 0.5051±0.0139 | 0.6667±0.0176 | 0.6840±0.0338 | 0.6545±0.0263 | 0.5256±0.0310 | 0.6903±0.0325 | 0.6517±0.0315 | 0.5750±0.0178 | 0.4401±0.0285 | 0.6287±0.0373 | 0.6440±0.0274 | 0.6036±0.0398 | 0.4655±0.0221 | 0.6510±0.0211 |
| PMIMC | 0.7183±0.0069 | 0.7131±0.0108 | 0.5795±0.0072 | 0.7383±0.0053 | 0.6667±0.0176 | 0.6847±0.0221 | 0.5360±0.0411 | 0.6977±0.0420 | 0.5360±0.0163 | 0.6198±0.0163 | 0.4846±0.0228 | 0.6590±0.0420 | 0.5907±0.0515 | 0.5480±0.0315 | 0.4034±0.0367 | 0.6060±0.0475 |
| ACOVA (Ours) | 0.7970±0.0143 | 0.7695±0.0139 | 0.6581±0.0195 | 0.7970±0.0143 | 0.7563±0.0242 | 0.7335±0.0085 | 0.6215±0.0182 | 0.7590±0.0216 | 0.6823±0.0295 | 0.6568±0.0193 | 0.5267±0.0235 | 0.6917±0.0252 | 0.6554±0.0310 | 0.6136±0.0223 | 0.4730±0.0216 | 0.6584±0.0229 |

*Table 7.* Ablation results on Handwritten dataset with different missing-view rates. 'w/' and 'w/o' mean 'with' and 'without', respectively. 'corr.' refers to considering cross-view correlations. The best result in each column is denoted in **bold**. Note that all results are averaged over five runs, with the mean and standard deviation reported.

| | Variant | 10% | | | | 30% | | | |
|---|---|---|---|---|---|---|---|---|---|
| | | ACC↑ | NMI↑ | ARI↑ | PUR↑ | ACC↑ | NMI↑ | ARI↑ | PUR↑ |
| | Learn **R** (w/ corr.) | **0.9363 ± 0.0067** | 0.8692 ± 0.0113 | 0.8645 ± 0.0133 | **0.9363 ± 0.0067** | **0.9376 ± 0.0073** | **0.8709 ± 0.0125** | **0.8669 ± 0.0144** | **0.9376 ± 0.0073** |
| | w/o corr. | 0.9169 ± 0.0545 | 0.8604 ± 0.0268 | 0.8547 ± 0.0514 | 0.9201 ± 0.0481 | 0.8954 ± 0.0515 | 0.8470 ± 0.0336 | 0.8210 ± 0.0527 | 0.9020 ± 0.0409 |
| | fixed $\rho = 0.1$ | 0.9339 ± 0.0081 | 0.8676 ± 0.0097 | 0.8595 ± 0.0163 | 0.9339 ± 0.0081 | 0.9249 ± 0.0165 | 0.8534 ± 0.0235 | 0.8418 ± 0.0315 | 0.9249 ± 0.0165 |
| | fixed $\rho = 0.3$ | 0.8874 ± 0.0614 | 0.8513 ± 0.0229 | 0.8231 ± 0.0532 | 0.8945 ± 0.0527 | 0.9217 ± 0.0177 | 0.8509 ± 0.0248 | 0.8355 ± 0.0340 | 0.9217 ± 0.0177 |
| | fixed $\rho = 0.5$ | 0.9088 ± 0.0498 | 0.8590 ± 0.0246 | 0.8391 ± 0.0454 | 0.9118 ± 0.0439 | 0.9301 ± 0.0104 | 0.8612 ± 0.0168 | 0.8510 ± 0.0208 | 0.9301 ± 0.0104 |
| | fixed $\rho = 0.7$ | 0.9069 ± 0.0410 | 0.8568 ± 0.0160 | 0.8322 ± 0.0359 | 0.9091 ± 0.0368 | 0.9317 ± 0.0117 | 0.8638 ± 0.0190 | 0.8545 ± 0.0229 | 0.9317 ± 0.0117 |
| | fixed $\rho = 0.9$ | 0.9362 ± 0.0096 | **0.8749 ± 0.0126** | **0.8649 ± 0.0190** | 0.9362 ± 0.0096 | 0.9172 ± 0.0137 | 0.8445 ± 0.0210 | 0.8263 ± 0.0268 | 0.9172 ± 0.0137 |
| | w/o $\mathcal{L}_{\text{ELBO}}$ | 0.6240 ± 0.0611 | 0.5284 ± 0.0574 | 0.3673 ± 0.0494 | 0.6240 ± 0.0417 | 0.5980 ± 0.0447 | 0.4548 ± 0.0340 | 0.3227 ± 0.0423 | 0.5980 ± 0.0414 |
| | w/o $\mathcal{L}_{\text{align}}$ | 0.8235 ± 0.0635 | 0.7768 ± 0.0477 | 0.7036 ± 0.0666 | 0.8235 ± 0.0675 | 0.6930 ± 0.0521 | 0.6378 ± 0.0296 | 0.5179 ± 0.0397 | 0.6930 ± 0.0286 |
| | Variant | 50% | | | | 70% | | | |
| | | ACC↑ | NMI↑ | ARI↑ | PUR↑ | ACC↑ | NMI↑ | ARI↑ | PUR↑ |
| Handwritten | Learn **R** (w/ corr.) | **0.9146 ± 0.0321** | **0.8489 ± 0.0280** | **0.8325 ± 0.0453** | **0.9146 ± 0.0321** | 0.8999 ± 0.0526 | **0.8351 ± 0.0325** | **0.8176 ± 0.0541** | 0.9021 ± 0.0483 |
| | w/o corr. | 0.8980 ± 0.0486 | 0.8373 ± 0.0199 | 0.8175 ± 0.0397 | 0.9054 ± 0.0340 | 0.7808 ± 0.0560 | 0.7905 ± 0.0253 | 0.7197 ± 0.0412 | 0.8099 ± 0.0403 |
| | fixed $\rho = 0.1$ | 0.8860 ± 0.0542 | 0.8288 ± 0.0345 | 0.8031 ± 0.0554 | 0.8898 ± 0.0478 | 0.8881 ± 0.0524 | 0.8231 ± 0.0268 | 0.7986 ± 0.0461 | 0.8918 ± 0.0451 |
| | fixed $\rho = 0.3$ | 0.8929 ± 0.0515 | 0.8335 ± 0.0311 | 0.8093 ± 0.0514 | 0.8972 ± 0.0435 | 0.8859 ± 0.0565 | 0.8219 ± 0.0269 | 0.7962 ± 0.0489 | 0.8903 ± 0.0477 |
| | fixed $\rho = 0.5$ | 0.8710 ± 0.0661 | 0.8291 ± 0.0332 | 0.7957 ± 0.0603 | 0.8821 ± 0.0527 | 0.8867 ± 0.0514 | 0.8239 ± 0.0238 | 0.7962 ± 0.0438 | 0.8908 ± 0.0434 |
| | fixed $\rho = 0.7$ | 0.8963 ± 0.0464 | 0.8410 ± 0.0163 | 0.8161 ± 0.0380 | 0.9001 ± 0.0390 | **0.9115 ± 0.0102** | 0.8319 ± 0.0150 | 0.8145 ± 0.0207 | **0.9115 ± 0.0102** |
| | fixed $\rho = 0.9$ | 0.8786 ± 0.0545 | 0.8330 ± 0.0165 | 0.8036 ± 0.0427 | 0.8828 ± 0.0495 | 0.8977 ± 0.0255 | 0.8140 ± 0.0282 | 0.7934 ± 0.0407 | 0.8977 ± 0.0255 |
| | w/o $\mathcal{L}_{\text{ELBO}}$ | 0.5495 ± 0.0583 | 0.3924 ± 0.0491 | 0.2874 ± 0.0545 | 0.5545 ± 0.0372 | 0.4845 ± 0.0587 | 0.3357 ± 0.0551 | 0.2301 ± 0.0483 | 0.5020 ± 0.0425 |
| | w/o $\mathcal{L}_{\text{align}}$ | 0.6645 ± 0.0775 | 0.6259 ± 0.0530 | 0.5179 ± 0.0684 | 0.6675 ± 0.0589 | 0.6090 ± 0.0550 | 0.5664 ± 0.0463 | 0.4574 ± 0.0592 | 0.6100 ± 0.0610 |

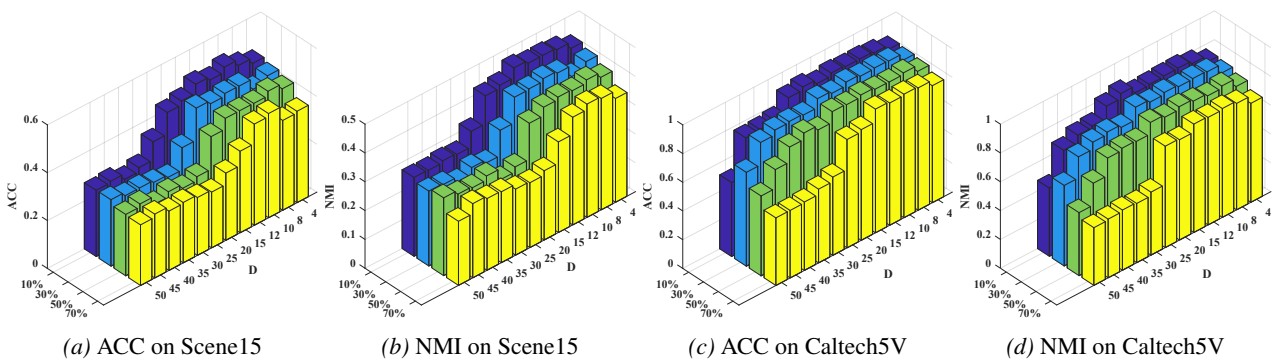

*(a) ACC on Scene15*    *(b) NMI on Scene15*    *(c) ACC on Caltech5V*    *(d) NMI on Caltech5V*

*Figure 7.* Clustering performance with different latent dimension $D$ and missing rate settings on Scene15 and Caltech5V dataset.

## C.3. Impact of Latent Dimension $D$

As illustrated in Fig. 7, we empirically investigate the impact of latent dimension $D$ on clustering performance across different missing rates on Scene15 and Caltech5V datasets. We examine $D$ ranging from {4, 8, 10, 12, 15, 20, 25, 30, 35, 40, 45, 50 }, and observe that increasing $D$ from 4 to 15 leads to performance improvements, while further increases yield metric degradation. This trend remains consistent across different missing ratios and datasets. Empirically, we recommend setting $D$ in the range of 10-15 for optimal clustering performance.

## C.4. Time Cost Analysis

In Table 9 we report the training time and clustering accuracy (ACC) of all compared methods on the Scene15 dataset with a 70% missing ratio. Note that BSV and CONCAT do not require any training procedures, and thus we omit their time cost in the table. Furthermore, for fair comparison, we report the training time for the main algorithm and exclude the pretraining stage. The results demonstrate that our ACOVA achieves an effective trade-off between computational cost and clustering performance, exhibiting moderate training time while maintaining superior clustering accuracy. We further repeat the analysis on the NoisyMNIST dataset, due to its larger scale, demonstrating explicitly the beneficial scaling of our approach when it comes to the number of samples $N$. To show the impact of $V$, we also conducted the runtime analysis for different numbers of views on the Caltech5V dataset in Table 10. It can be observed that the overhead of the cubic scaling with respect to $V$ is negligible for most common settings.

*Table 8.* Ablation results on Scene15 dataset with different missing-view rates. '*w/*' and '*w/o*' mean '*with*' and '*without*', respectively. 'corr.' refers to considering cross-view correlations. The best result in each column is denoted in **bold**. Note that all results are averaged over five runs, with the mean and standard deviation reported.

| | Variant | 10% | | | | 30% | | | |
|---|---|---|---|---|---|---|---|---|---|
| | | ACC↑ | NMI↑ | ARI↑ | PUR↑ | ACC↑ | NMI↑ | ARI↑ | PUR↑ |
| | Learn $\mathbf{R}$ (*w/* corr.) | $0.5040 \pm 0.0153$ | $\mathbf{0.4804} \pm 0.0050$ | $\mathbf{0.3273} \pm 0.0088$ | $0.5270 \pm 0.0202$ | $\mathbf{0.4807} \pm 0.0136$ | $\mathbf{0.4560} \pm 0.0090$ | $0.3106 \pm 0.0079$ | $\mathbf{0.5119} \pm 0.0164$ |
| | *w/o* corr. | $0.4863 \pm 0.0103$ | $0.4780 \pm 0.0059$ | $0.3192 \pm 0.0085$ | $0.5021 \pm 0.0109$ | $0.4678 \pm 0.0054$ | $0.4440 \pm 0.0033$ | $0.3097 \pm 0.0033$ | $0.4823 \pm 0.0089$ |
| | fixed $\rho = 0.1$ | $0.4963 \pm 0.0142$ | $0.4722 \pm 0.0069$ | $0.3237 \pm 0.0105$ | $0.5220 \pm 0.0128$ | $0.4702 \pm 0.0077$ | $0.4466 \pm 0.0037$ | $0.2977 \pm 0.0041$ | $0.4941 \pm 0.0095$ |
| | fixed $\rho = 0.3$ | $\mathbf{0.5057} \pm 0.0062$ | $0.4743 \pm 0.0099$ | $0.3272 \pm 0.0060$ | $\mathbf{0.5281} \pm 0.0075$ | $0.4746 \pm 0.0202$ | $0.4458 \pm 0.0128$ | $0.3011 \pm 0.0136$ | $0.4986 \pm 0.0193$ |
| | fixed $\rho = 0.5$ | $0.4963 \pm 0.0190$ | $0.4673 \pm 0.0113$ | $0.3214 \pm 0.0099$ | $0.5218 \pm 0.0200$ | $0.4689 \pm 0.0175$ | $0.4414 \pm 0.0111$ | $0.3005 \pm 0.0129$ | $0.4891 \pm 0.0221$ |
| | fixed $\rho = 0.7$ | $0.4761 \pm 0.0261$ | $0.4543 \pm 0.0220$ | $0.3023 \pm 0.0235$ | $0.5031 \pm 0.0278$ | $0.4742 \pm 0.0088$ | $0.4255 \pm 0.0104$ | $0.2907 \pm 0.0067$ | $0.4940 \pm 0.0106$ |
| | fixed $\rho = 0.9$ | $0.4649 \pm 0.0181$ | $0.4505 \pm 0.0115$ | $0.2938 \pm 0.0179$ | $0.4939 \pm 0.0158$ | $0.4609 \pm 0.0146$ | $0.4187 \pm 0.0064$ | $0.2710 \pm 0.0148$ | $0.4727 \pm 0.0007$ |
| | *w/o* $\mathcal{L}_{\text{ELBO}}$ | $0.1599 \pm 0.0214$ | $0.1605 \pm 0.0192$ | $0.0519 \pm 0.0178$ | $0.1632 \pm 0.0259$ | $0.1407 \pm 0.0508$ | $0.0870 \pm 0.0315$ | $0.0050 \pm 0.0187$ | $0.1431 \pm 0.0495$ |
| | *w/o* $\mathcal{L}_{\text{align}}$ | $0.4566 \pm 0.0270$ | $0.4624 \pm 0.0098$ | $0.2926 \pm 0.0179$ | $0.4941 \pm 0.0158$ | $0.3974 \pm 0.0148$ | $0.4000 \pm 0.0146$ | $0.2335 \pm 0.0195$ | $0.4193 \pm 0.0310$ |
| | Variant | 50% | | | | 70% | | | |
| | | ACC↑ | NMI↑ | ARI↑ | PUR↑ | ACC↑ | NMI↑ | ARI↑ | PUR↑ |
| Scene15 | Learn $\mathbf{R}$ (*w/* corr.) | $\mathbf{0.4710} \pm 0.0138$ | $\mathbf{0.4360} \pm 0.0034$ | $\mathbf{0.2973} \pm 0.0084$ | $\mathbf{0.4987} \pm 0.0126$ | $\mathbf{0.4428} \pm 0.0140$ | $\mathbf{0.4083} \pm 0.0055$ | $\mathbf{0.2733} \pm 0.0031$ | $\mathbf{0.4636} \pm 0.0161$ |
| | *w/o* corr. | $0.4371 \pm 0.0568$ | $0.4163 \pm 0.0402$ | $0.2737 \pm 0.0399$ | $0.4577 \pm 0.0654$ | $0.4136 \pm 0.0266$ | $0.3950 \pm 0.0228$ | $0.2519 \pm 0.0189$ | $0.4286 \pm 0.0293$ |
| | fixed $\rho = 0.1$ | $0.4653 \pm 0.0222$ | $0.4292 \pm 0.0054$ | $0.2955 \pm 0.0128$ | $0.4966 \pm 0.0138$ | $0.4181 \pm 0.0100$ | $0.3878 \pm 0.0112$ | $0.2518 \pm 0.0132$ | $0.4287 \pm 0.0159$ |
| | fixed $\rho = 0.3$ | $0.4598 \pm 0.0268$ | $0.4275 \pm 0.0138$ | $0.2901 \pm 0.0163$ | $0.4885 \pm 0.0227$ | $0.4165 \pm 0.0130$ | $0.3882 \pm 0.0105$ | $0.2508 \pm 0.0085$ | $0.4309 \pm 0.0147$ |
| | fixed $\rho = 0.5$ | $0.4569 \pm 0.0057$ | $0.4224 \pm 0.0109$ | $0.2863 \pm 0.0092$ | $0.4815 \pm 0.0110$ | $0.4105 \pm 0.0291$ | $0.3893 \pm 0.0061$ | $0.2521 \pm 0.0137$ | $0.4228 \pm 0.0263$ |
| | fixed $\rho = 0.7$ | $0.4590 \pm 0.0082$ | $0.4295 \pm 0.0077$ | $0.2818 \pm 0.0096$ | $0.4765 \pm 0.0096$ | $0.4192 \pm 0.0219$ | $0.3827 \pm 0.0136$ | $0.2503 \pm 0.0175$ | $0.4353 \pm 0.0245$ |
| | fixed $\rho = 0.9$ | $0.3918 \pm 0.0364$ | $0.3877 \pm 0.0318$ | $0.2083 \pm 0.0349$ | $0.4144 \pm 0.0370$ | $0.3656 \pm 0.0111$ | $0.3575 \pm 0.0067$ | $0.2239 \pm 0.0090$ | $0.3765 \pm 0.0056$ |
| | *w/o* $\mathcal{L}_{\text{ELBO}}$ | $0.1405 \pm 0.0189$ | $0.0854 \pm 0.0095$ | $0.0220 \pm 0.0084$ | $0.1411 \pm 0.0195$ | $0.1394 \pm 0.0224$ | $0.0746 \pm 0.0192$ | $0.0170 \pm 0.0158$ | $0.1402 \pm 0.0276$ |
| | *w/o* $\mathcal{L}_{\text{align}}$ | $0.3393 \pm 0.0701$ | $0.3412 \pm 0.0603$ | $0.1902 \pm 0.0465$ | $0.3454 \pm 0.0687$ | $0.2105 \pm 0.0588$ | $0.1798 \pm 0.0931$ | $0.0821 \pm 0.0595$ | $0.2157 \pm 0.0575$ |

*Table 9.* Runtime comparison on the Scene15 and NoisyMNIST datasets.

| Dataset | Metric | Methods | | | | | | | | |
|---|---|---|---|---|---|---|---|---|---|---|
| | | BSV | CONCAT | DSIMVC | DCP | CPSPAN | DVIMC | MVP | PMIMC | ACOVA |
| Scene15 | Time | - | - | $2.35 \times 10^{-2}$ | $1.47 \times 10^{1}$ | $6.62 \times 10^{0}$ | $1.12 \times 10^{-1}$ | $1.84 \times 10^{0}$ | $4.63 \times 10^{0}$ | $1.16 \times 10^{-1}$ |
| | ACC | 0.2593 | 0.2596 | 0.2654 | 0.3784 | 0.2025 | 0.4136 | 0.4407 | 0.3178 | 0.4428 |
| NoisyMNIST | Time | - | - | $3.06 \times 10^{0}$ | $1.48 \times 10^{1}$ | $2.22 \times 10^{2}$ | $4.98 \times 10^{0}$ | $7.61 \times 10^{1}$ | $1.14 \times 10^{2}$ | $5.59 \times 10^{0}$ |
| | ACC | 0.2672 | 0.2805 | 0.6245 | 0.9225 | 0.5486 | 0.8741 | 0.6660 | 0.6044 | 0.9377 |

## C.5. Convergence Analysis

To verify the convergence, we analyze both the training objective (Loss) and clustering performance (Accuracy) against epochs on the Scene15 and Handwritten datasets with a 10% missing ratio. As illustrated in Fig. 8, the training objective exhibits consistent monotonic descent until convergence, while the clustering accuracy steadily increases and eventually saturates. These results all confirm good convergence properties of the proposed ACOVA.

## C.6. Learning Multiple Correlation Matrices

As our proposed framework enables end-to-end optimization of $\mathbf{R}$, we can further extend our framework to learn multiple correlation matrices. In Fig. 9, we compare the performance of learning one shared $\mathbf{R}$ to learning a separate $\mathbf{R}$ for each dimension or for groups of dimensions. More specifically, "grouped_2" and "grouped_5" correspond to learning two and five $\mathbf{R}$ matrices for groups of $d/2$ and $d/5$ dimensions, respectively. As can be seen in Fig. 9, while performance is relatively similar for the different approaches, we notice that increasing the number of $\mathbf{R}$ matrices does not necessarily increase performance, and can instead lead to performance degradation. This is not surprising as we are considering the clustering setting, where we lack strong supervised guidance, and increasing complexity and model flexibility could potentially result in degenerate solutions. We thus recommend learning a single shared $\mathbf{R}$.

## C.7. Analysis of Learned Correlation Matrix

In this section, we further analyze the process of learning the correlation matrix. In Fig. 10, we monitor $\|\mathbf{R}-\mathbf{I}\|_F$ throughout training on the Caltech5V and Handwritten datasets. From these results, we observe that $\|\mathbf{R}-\mathbf{I}\|_F$ converges to values between 1.2 and 2.9 for the Caltech5V dataset and values between 1.4 and 3 for the Handwritten dataset. This lies well within the theoretical range of $[0, \sqrt{20}]$ and $[0, \sqrt{30}]$, respectively. We further observe that for large missingness, the correlation matrix is closer to the identity, while for low missingness the correlation between views is higher. This intuitively makes sense as less samples are being observed with paired views.

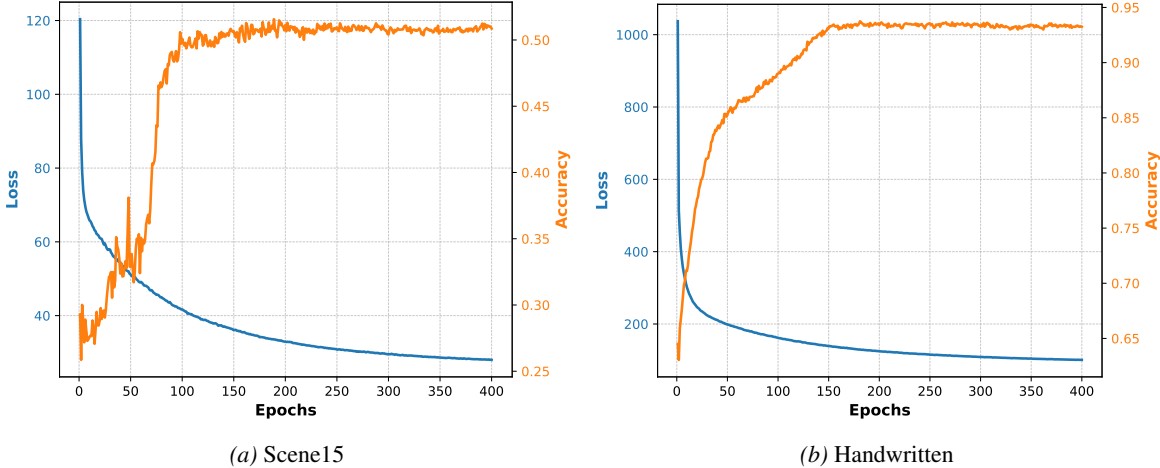

*(a)* Scene15          *(b)* Handwritten

*Figure 8.* Convergence curves on the Scene15 and Handwritten dataset.

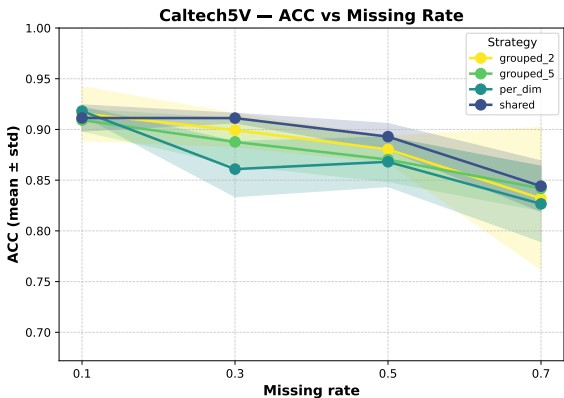

*Figure 9.* Clustering accuracy (ACC) of different strategies on the Caltech5V dataset under varying missing rates. The proposed shared correlation matrix achieves consistently strong performance compared with other strategies.

Fig. 11 and 12 further demonstrate the evolution of $\mathbf{R}$ throughout training. For the Caltech5V dataset, we observe the strongest correlation between the HOG and LBP features, which both capture local texture and edge information. This further validates that our approach learns a meaningful correlation structure.

Finally, we further empirically demonstrate that $\mathbf{R}$ is stable across runs. For this we conduct five runs with different random seeds on the Handwritten dataset and compute the variance across runs for the different elements in $\mathbf{R}$. Fig. 13 provides these results for 10% and 90% missingness, demonstrating high stability even for large missing-rates.

## D. Limitation and Future Work

While our method demonstrates effectiveness on incomplete multi-view clustering tasks by relaxing the independence assumption and adaptively modeling cross-view correlations, it has practical limitations when it comes to the computational cost of scaling to a substantially larger number of views (*e.g.*, $V > 20$) (see discussion in Appendix A.8). While most current settings leverage only a small number of views, this could potentially become a limitation in the future.

In the future, we would like to further investigate the application of our proposed mechanism in the context of imputation-based IMVC methods to further enhance performance. Furthermore, although we follow prior approaches and assume Gaussian distributions in our approach, there remains potential in exploring more expressive or non-Gaussian distributions for more complex data. Another important future direction is to establish formal convergence guarantees for multi-modal variational inference with correlation-aware posterior aggregation, extending recent theoretical advances from standard

*Table 10.* Runtime analysis with varying number of views on Caltech5V dataset.

| Number of Views ($V$) | 2 | 3 | 4 | 5 |
|---|---|---|---|---|
| Runtime (seconds) | 0.18 ± 0.03 | 0.22 ± 0.03 | 0.26 ± 0.04 | 0.27 ± 0.04 |

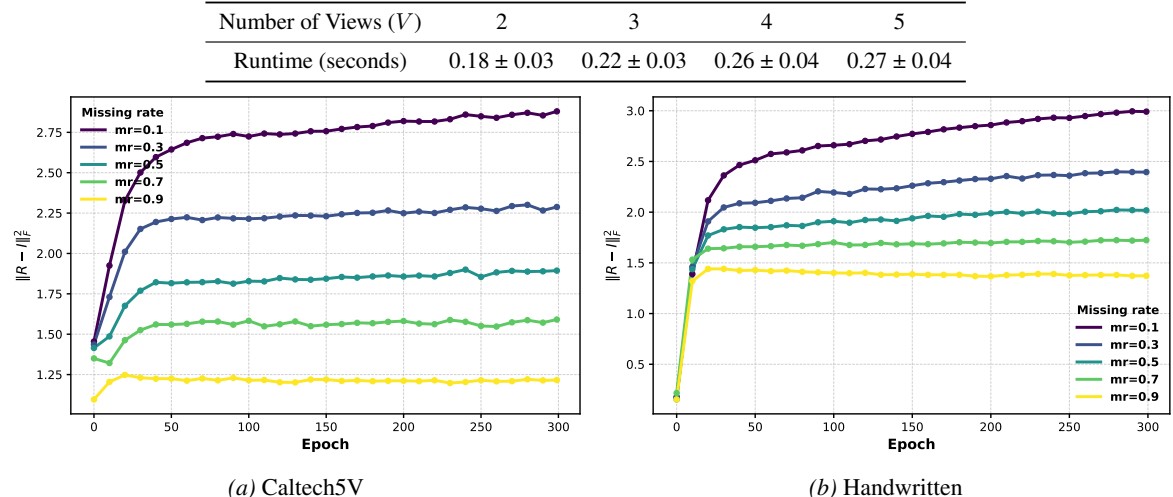

*(a)* Caltech5V          *(b)* Handwritten

*Figure 10.* Convergence curves of $\|\mathbf{R} - \mathbf{I}\|_F$ on Caltech5V and Handwritten dataset.

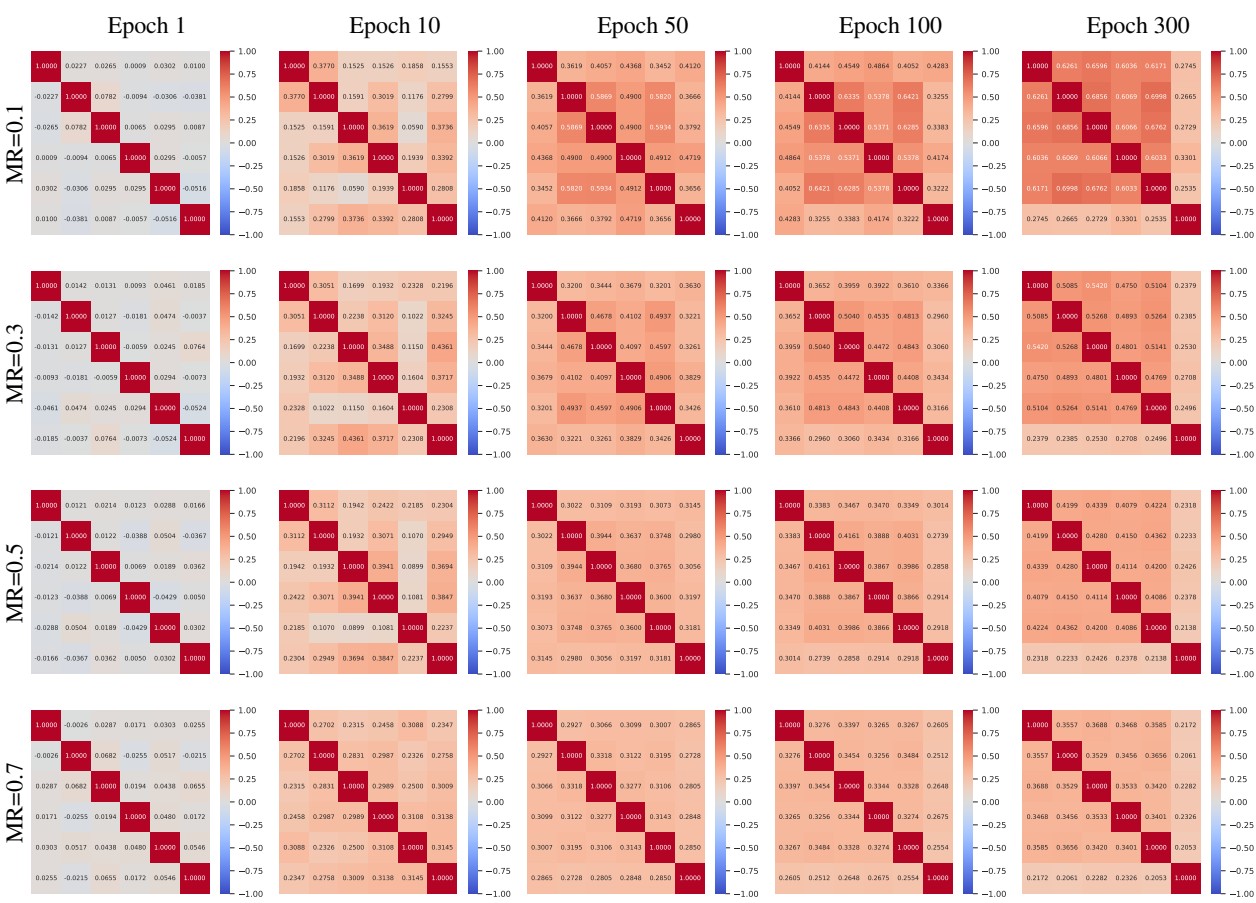

*Figure 11.* Correlation matrices on Handwritten: rows are missing rates, columns are epochs.

VAEs to incomplete multi-view settings. Finally, we aim to evaluate ACOVA on incremental-view IMVC scenarios, where new views are introduced during training. When additional views are anticipated, a straightforward strategy would be to treat them as missing during early training stages. For unexpectedly introduced views, one possible approach is to expand

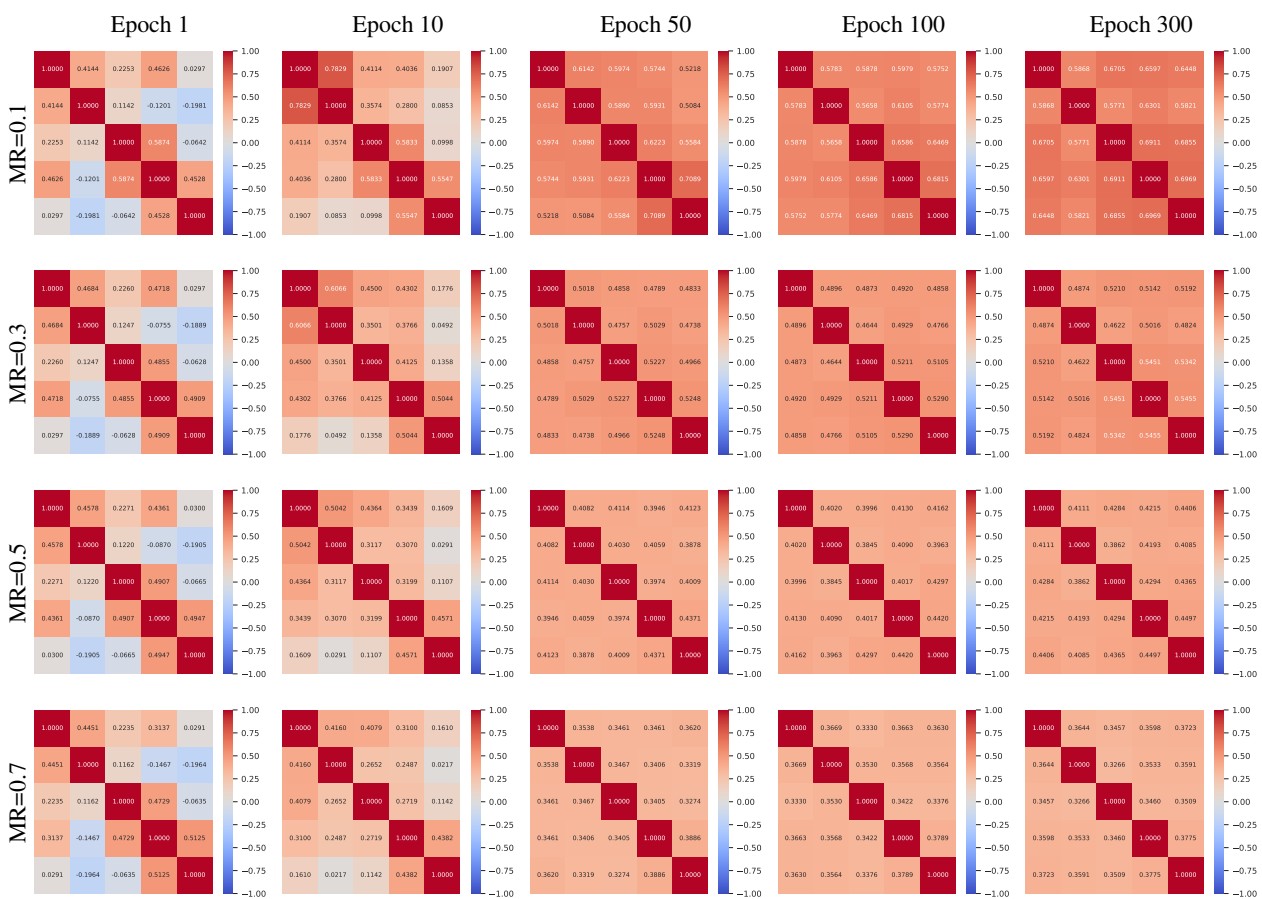

*Figure 12.* Correlation matrices on Caltech5V: rows are missing rates, columns are epochs.

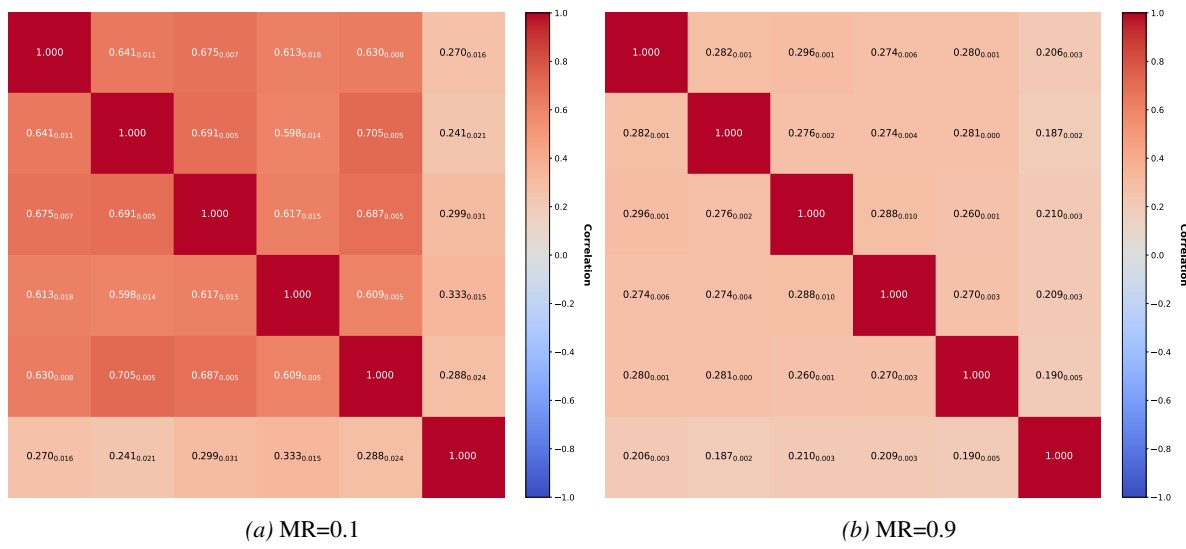

*(a)* MR=0.1        *(b)* MR=0.9

*Figure 13.* Correlation matrices for the Handwritten dataset averaged across five runs with mean and standard deviation.

the learned correlation matrix $\mathbf{R}$ through random initialization of the new entries, followed by a Cholesky decomposition to obtain an updated $\mathbf{L}$, allowing training to continue from this initialization. Investigating principled strategies for dynamically expanding and adapting the learned correlation structure in such settings constitutes an interesting direction for future work.

