# OpenReview forum: "Beyond Independence: Learning Correlated Views for Variational Incomplete Multi-View Clustering"
_ICML.cc/2026/Conference — ICML 2026 regular_

### Official Review · Reviewer_AwyB · 2026-03-04

**Soundness:** 3
**Presentation:** 3
**Significance:** 3
**Originality:** 3
**Overall Recommendation:** 4
**Confidence:** 4

**Summary:**

This paper proposes an incomplete multi-view clustering (IMVC) method within the VAE framework, named ACOVA. Different from existing methods that depend on conditional independence assumption across views, the proposed ACOVA models inter-view correlations by utilizing the covariance structure of posterior estimation errors. The correlation matrix is automatically learned. Extensive results are provided to show the effectiveness of the proposed method.

**Compliance With Llm Reviewing Policy:**

Affirmed.

**Key Questions For Authors:**

1. See the weaknesses.
2.  What is the special design of the model for the incompleteness of views?

**Limitations:**

See the weaknesses.

**Strengths And Weaknesses:**

1. This paper is well-motivated.  Different from existing methods that depend on conditional independence assumption, the proposed ACOVA models inter-view correlations by utilizing the covariance structure of posterior estimation errors.

2. The structure of the paper is well-organized and the expression is clear.

3. The complexity of the proposed method is analyzed and time comparison results are also presented.

4. Extensive experimental results on 6 datasets are provided to validate the effectiveness of the proposed method. Compared with several SOTA baselines, the proposed method shows promising results.

Weaknesses:
1. It seems that the proposed method neglects to capture complementary information of multiple views.

2. The conclusion in Theorem 4.3 seems to be trivial.

3. The results in Table 2 lack significance tests.

---

> ### Author Rebuttal · Authors · 2026-03-31
>
> Thank you for your constructive comments. Below, we address your mentioned weaknesses (W) and questions (Q).
>
> ## (W1) Capturing complementary information across views
>
> Thank you for raising this concern. We would like to explain how complementary information is considered:
>
> * *Shared latent with multi-view reconstruction*. The ELBO in Eq. (13) contains the term $\mathbb{E}{q_\phi(z\mid{x_v})}[\sum_{v\in\mathcal{V}}\log p_\theta(x_v\mid z)]$, which ensures that the latent variables sampled from the joint posterior distribution are capable of generating all observed views. Therefore, generative models encourage learning latent variables that also embed view-specific information, eliminating the need to factorize the latent space.
> Under the incomplete setting, only available samples contribute via masking.
>
> * *View-specific encoders and alignment*. Each view-specific distribution $q_{\phi_v}(z\mid x_v)$ estimates the parameters ($\mu_v$,$\sigma_v^2$). The alignment objective in Eq. (14) then minimizes the divergence between the joint posterior distribution ($q_\phi(z\mid{x_v})$) and the view-specific distribution. This can be considered as a way of preventing the joint posterior distribution from disregarding the information encoded by the view-specific distributions.
>
> * *Correlation learning*. Learning R in $\Sigma=DRD$ models dependence among estimation errors across views after conditioning on the shared z (Sec.4.1). It does not remove complementary information in ${x_v}$ but calibrates uncertainty-aware fusion so that correlated errors are not treated as independent pieces of evidence (as in PoE-style independence).
>
> We will add a short clarification in Sec.4 to make it clearer for readers.
>
>
> ## (W2) Role of Theorem 4.3 (Frobenius bound)
> Thank you for the comment. We agree that the proof of Theorem 4.3 is straightforward. We include it for completeness and to keep the paper self‑contained, as it underpins our interpretation of the learned correlation matrix in Appendix C.7. The result provides the relevant upper bound against which we contextualize the empirical correlations. If preferred, we can move the theorem to the appendix without affecting the main narrative.
>
> ## (W3) Statistical significance tests for ablation (Table 2)
>
> Thank you for the suggestion. As done for the main Table (Table 1), we will add the full version of Table 2 with standard deviation in the appendix of the revised version.
> Following your suggestion, we conduct Friedman tests on each evaluation metric across the 2 datasets and 4 missing-rate settings (i.e., 8 cases per metric with 9 variants). For ACC, NMI, ARI, and PUR, the obtained ($F_F$) values (21.429, 17.260, 19.186, and 20.654) all exceed the critical value 2.10. Therefore, the null hypothesis of equivalent performance is rejected at the 0.05 significance level.
>
> Since the Friedman tests are significant, we further perform a Bonferroni-Dunn post-hoc analysis using a unified critical difference (CD) computed over the pooled benchmark cases. Specifically, we treat each metric–dataset combination as a benchmark case, resulting in 32 cases in total.
> As shown in the Table below, under the Bonferroni-Dunn test (CD$\approx$1.71), all ablated variants are significantly worse than our "Learn $\mathbf{R}$ (w/ corr.)", validating the effectiveness of our proposed method.
> | Method                     | Avg rank |
> |----------------------------|-------------------|
> | Learn $\mathbf{R}$ (w/ corr.) | 1.1875         |
> | fixed $\rho=0.1$           | 3.7031            |
> | fixed $\rho=0.7$           | 3.8750            |
> | fixed $\rho=0.3$           | 4.1094            |
> | fixed $\rho=0.5$           | 4.5938            |
> | w/o corr.                  | 4.8438            |
> | fixed $\rho=0.9$           | 5.7813            |
> | w/o $\mathcal{L}_{\text{align}}$ | 7.9063      |
> | w/o $\mathcal{L}_{\text{ELBO}}$  | 9.0000      |
>
> ## (Q) Special design of the model for handling incomplete views
>
> Thank you for the question. Handling incomplete views follows the standard imputation-free principle in incomplete MVC: we mask out missing views so that they do not contribute to the fused posterior. Thus, the incomplete-view mechanism is primarily the mask-aware aggregation shared across training and inference. Our method-specific extension is not the mask itself, but learning cross-view error correlation inside this masked fusion. We further empirically verify in Figure 6 that our approach also performs well in the fully observed (complete-view) setting, showing that our model further generalizes to complete multi-view scenarios.

---

> > ### Author Rebuttal · Reviewer_AwyB · 2026-04-03
> >
> > In terms of the last question, if there is no special design for the incompleteness of views, how to avoid the problem of information imbalance caused by the absence of views?

---

> > > ### Author Response · Authors · 2026-04-03
> > >
> > > Thank you for the follow-up. We are glad most concerns have been addressed and appreciate the clarification on your last point. We will in the following expand on how the proposed mechanism leverages the masking to directly address information imbalance within our aggregation. In particular, we will first consider information imbalance when samples have varying number of views and then also consider the setting when there is some information imbalance with respect to noise/redundancy.
> > >
> > > In line with standard VAE training under partial observations, we learn a shared latent z for each sample using only its observed views. Missing modalities are not fed to encoders and are not treated as observed evidence in the ELBO. More specifically, the approximate posterior $q\_\phi(\mathbf{z} \mid \lbrace\mathbf{x}\_{v}\rbrace\_{v \in \mathcal{V}})$ is obtained through an aggregation mechanism combining the *available* view-specific posteriors $q\_{\phi\_{v}}(\mathbf{z} \mid \mathbf{x}\_{v})$. This paradigm is shared by IMVC methods such as DVIMC, which in this way learn a unified representation despite missing views.
> > >
> > > Let us look more concretely at how the missing-view mask $\mathbf{M}$ impacts our proposed aggregation through $q\_\phi(\mathbf{z} \mid \lbrace\mathbf{x}\_{v}\rbrace\_{v \in \mathcal{V}}) \sim \mathcal{N}(\tilde{\mu},\tilde{\sigma}^2) = \mathcal{N}(\mathbf{A}\_{\mathbf{M}}^{-1} \mathbf{B}\_{\mathbf{M}}, \mathbf{A}\_{\mathbf{M}}^{-1})$ where $\mathbf{A}\_{\mathbf{M}} = \mathbb{1}^\top (\mathbf{\Sigma}^{-1} \odot \mathbf{M}\mathbf{M}^\top) \mathbb{1}$ and $\mathbf{B}\_{\mathbf{M}} = \mathbb{1}^\top (\mathbf{\Sigma}^{-1} \odot \mathbf{M}\mathbf{M}^\top) (\mu \odot \mathbf{M})$ (Eq. 9-11). For simplicity, let’s consider a single latent dimension, where this reduces to the quadratic form $\mathbf{A}\_{\mathbf{M}} = \mathbf{M}^\top \boldsymbol{\Sigma}^{-1}\mathbf{M}$. Fewer observed views (more zeros in $\mathbf{M}$) strictly remove rows/columns from this quadratic form, so $\mathbf{A}\_{\mathbf{M}}$ cannot increase when views are missing. This means that if there are fewer views, the missing-view mask ensures that $\mathbf{A}\_{\mathbf{M}}$ will be smaller, leading to larger posterior variance, explicitly preventing overconfidence when evidence is scarce. This effectively addresses the problem that samples with fewer views can lead to overconfidence and ensures that the information imbalance is taken into account (more views, more information, more confident).
> > >
> > > Another factor of information-imbalance is related to the quality of the views (how redundant/noisy). To address this, each encoder produces $\sigma_v^2$, such that a view with higher uncertainty contributes less to $\mathbf{A}\_{\mathbf{M}}$ and $\mathbf{B}\_{\mathbf{M}}$ as $\boldsymbol\Sigma = \mathbf{D}\mathbf{R}\mathbf{D}$, with $\mathbf{D}=\mathrm{diag}(\sigma_1,\ldots,\sigma_V)$. This reduces the influence of weak/noisy views and mitigates imbalance caused by heterogeneous view quality. Finally, the learned cross-view correlations address one key problem of PoE, where PoE sums precisions and becomes overconfident (briefly discussed in Lines 128-134). We instead avoid this problem, where similar or duplicate views “outvote” single distinct views by accounting for dependency, yielding calibrated results.
> > >
> > > Let us also here look at a simple example in which we have two views $x_1$ and $x_2$, and the true joint posterior variable is $z =8$. The two view-specific encoders estimate $\mu_1=4$ and $\mu_2=8$, with uncertainties $\sigma^2_1=3$ and $\sigma^2_2=1$, and correlation $\rho_{12} = 0.55$. Winkler (1981) shows that the expectation of the joint posterior distribution (the estimate of $z$) can be written as the weighted average of the view-specific estimates $\tilde{\mu}=\omega_1\mu_1+\omega_2\mu_2$, where the weights are $\omega_1 = (\sigma_2^2-\rho_{12}\sigma_1\sigma_2)/(\sigma_1^2+\sigma_2^2-2\rho_{12}\sigma_1\sigma_2)$ and $\omega_2 = (\sigma_1^2-\rho_{12}\sigma_1\sigma_2)/(\sigma_1^2+\sigma_2^2-2\rho_{12}\sigma_1\sigma_2)$. Therefore, the estimate of the joint posterior distribution according to our proposed aggregation method is $\tilde{\mu}=0.02\times4+0.98\times8$, leaning towards the relatively more accurate and less uncertain estimate. In contrast, neglecting the correlation between view-specific encoders underestimates the true joint posterior variable as $\tilde{\mu}=0.25\times4+0.75\times8$.
> > >
> > > Empirically, we observe this in Fig. 3, where we clearly see that the learned correlations adapt with noise/missingness, indicating that the mechanism calibrates confidence as redundancy or scarcity changes, effectively handling information imbalance.
> > >
> > > We will clarify these points in the paper.

---

### Official Review · Reviewer_et6d · 2026-03-07

**Soundness:** 2
**Presentation:** 3
**Significance:** 2
**Originality:** 3
**Overall Recommendation:** 4
**Confidence:** 3

**Summary:**

To address the oversimplified assumptions in incomplete multiview clustering (IMVC), this paper presents ACOVA, a variational framework that relaxes the conventional conditional independence assumption across views during posterior aggregation. The key idea is to model inter-view dependencies through the covariance structure of estimation errors between view-specific posteriors and the true latent variable. Experimental results on six datasets across different missing rate settings show the effectiveness of the proposed method.

**Compliance With Llm Reviewing Policy:**

Affirmed.

**Final Justification:**

My concerns have been addressed, and I have updated my scores.

**Key Questions For Authors:**

Please refer to the aforementioned weakness points and carefully address them.

**Limitations:**

Yes.

**Strengths And Weaknesses:**

Strengths:
1. The paper identifies the limitation of existing variational IMVC methods, i.e., the conditional independence assumption across view in posterior aggregation. The connection to related methods is well discussed.
2. Comprehensive experiments on six datasets with different missing rates show the effectiveness of the proposed method. Both quantitative and qualitative results are provided.
3. The visual analysis on the Handwritten dataset further verifies that the learned correlation matrix captures semantically meaningful structure rather than noise.

Weaknesses:
1. The empirical gains over DVIMC are often modest. This raises the question of whether the independence assumption is truly the actual bottleneck in practice. It is suggested to include a more detailed discussion of when correlation modelling matters most.
2. The proposed method has a complexity of $O(NDV^3)$ (reducible to $O(NDV^2)). For settings with many views or where views are added incrementally, the framework's behaviour and scalability are not explored.
3. All six datasets are relatively small-scale and with mostly image features. Only CUB dataset is the truly multimodal case with images and texts but with only 600 samples. A more comprehensive evaluation on larger-scale or more heterogeneous multimodal datasets is suggested.
4. Why are the CPSPAN and CONCAT having the exact same results on Fashion dataset across all missing rates?

---

> ### Author Rebuttal · Authors · 2026-03-31
>
> Thank you for your constructive feedback.
>
> ## (Q1) Importance of correlation modeling and consistent improvements
>
> Thank you for the comment. While improvements for some datasets/missing-rates can appear modest, we demonstrate that by replacing the independence-assuming PoE with our correlation-aware aggregation approach, consistent and significant improvements can be obtained (ACC improvements of 4.5\% on average across datasets/missing rates). Notably, the added complexity is minimal: 3 additional parameters for 2-view datasets and 21 for 6-view datasets, relative to a DVMIC model with more than 6 (16) million parameters for 2 (6) views, with minimal runtime overhead (Appendix C.4). This means that these consistent improvements are obtained essentially for free, which we believe demonstrates the importance of correlation modeling.
>
> ## (Q2) Scalability when views are added incrementally
>
> Thank you for raising these interesting points.
>
> **Incremental views:** While this is not a setting that is usually considered in IMVC works, we agree that there could be practical settings where new views are added incrementally at later stages of training. For cases when we know that this will happen, the solution would be straight-forward to treat the view as missing in the beginning of training. If this happens unexpectedly, we would suggest the user to extend the current R by randomly initializing the new elements, performing a Cholesky decomposition to obtain a new L and then continue training from this initialization. We will add this recipe in the revision.
>
> **Number of views:** We evaluate the performance of our model on the standard datasets that are used by the IMVC community, containing two to six views, demonstrating its effectiveness. Within the IMVC domain and for most practical settings, datasets with six views are considered datasets with a large number of views, given that most work currently is still only considering two or three views. This also means that common benchmarks in the community are limited when it comes to even larger number of views. Similarly, we can extend the traditional interpretation of “views” to “modalities”, which is straightforward given that most modern IMVC approaches work in the latent space. We did this to show additional generalization, but also then the majority of works are considering settings with less views, such as image-text, image-video-text, etc.
>
> However, note that when it comes to the number of additional parameters, these are minimal when contrasted to the full DVIMC model. Even if we would scale up to a considerably larger number of views, the number of added parameters would be insignificant compared to the number of parameters in the view-specific encoders/decoders of DVIMC. For instance 20 views would only add 210 parameters, while the 6 view model of DVIMC already contains more than 16 million parameters. Further, we have empirically looked at the runtime when increasing the number of views in Table 8. Here we observe that increase in runtime is minimal as views are added, especially when considering that this time also includes the addition of the new modality-specific encoders and not just the cost of our proposed mechanism.
>
> ## (Q3) Larger-scale dataset
>
> We thank the reviewer for this suggestion. In this work, we follow the standard evaluation protocol used in recent state-of-the-art variational IMVC methods (including DVIMC), enabling direct and fair comparison. The six datasets are selected to span diverse regimes (small/large sample sizes, few/many views, raw pixels vs. extracted features), and we observe consistent improvements across all settings.
>
> Importantly, our method operates in latent space and is agnostic to modality. To demonstrate this beyond vision-centric benchmarks (the common focus in IMVC), we include the multimodal CUB dataset (image–text), where we observe the same trends. Moreover, the proposed aggregation is inherently mini-batch based, making it scalable to larger datasets without modification. Finally, our evaluation already includes datasets with up to 70,000 samples, where gains remain consistent, supporting the scalability of our approach. We will emphasize this complementary of the datasets in the revision.
>
> ## (Q4)  CPSPAN Fashion dataset results
>
> Thank you for pointing this out. We will replace the CPSPAN results with the correct ones for the Fashion dataset. Note, all conclusions remain intact. The CPSPAN results are:
>
> Missing| ACC        | NMI        | ARI        | PUR        |
> -| ---------- | ---------- | ---------- | ---------- |
> 0.1| 0.7130±0.0547 | 0.7566±0.0345 | 0.7548±0.0517 | 0.6303±0.0415 |
> 0.3| 0.6981±0.0457 | 0.7524±0.0232 | 0.7446±0.0431 | 0.6179±0.0342 |
> 0.5| 0.6886±0.0784 | 0.7527±0.0378 | 0.7436±0.0686 | 0.6237±0.0473 |
> 0.7| 0.6672±0.0811 | 0.7416±0.0392 | 0.7273±0.0713 | 0.6043±0.0498 |

---

> > ### Author Rebuttal · Reviewer_et6d · 2026-04-02
> >
> > Thank you for your rebuttal. I will consider raising my score.

---

> > > ### Author Response · Authors · 2026-04-05
> > >
> > > Thank you for carefully considering our rebuttal, marking your concerns as fully resolved, and indicating a willingness to raise your original Weak Reject score.

---

### Official Review · Reviewer_3PKb · 2026-03-11

**Soundness:** 3
**Presentation:** 3
**Significance:** 3
**Originality:** 2
**Overall Recommendation:** 3
**Confidence:** 4

**Summary:**

This paper addresses the problem of incomplete multi-view clustering (IMVC) by proposing ACOVA (Adaptive Correlation-aware Variational Aggregation), a variational framework that relaxes the conventional independence assumption across views. The key innovation is modeling inter-view dependencies through a learnable correlation matrix embedded in the error covariance structure of view-specific posterior estimates. The authors provide theoretical analysis including an identifiability proof and a deviation bound, and demonstrate strong empirical results across six benchmark datasets under various missing rates. However, the key innovation lies in cross-view modeling, which is considered a modest extension of the existing variational IMVC framework (e.g. VAE and deep IMVC) rather than a fundamentally new paradigm. This level of innovation does not meet the high standards expected by ICML.

**Compliance With Llm Reviewing Policy:**

Affirmed.

**Key Questions For Authors:**

1. It would be beneficial for the authors to clarify how correlated data generation is modeled in the proposed framework.

2. The performance improvements achieved by the proposed ACOVA framework appear to be relatively limited compared with existing methods.

3. While empirical convergence curves are provided, the manuscript does not discuss theoretical convergence guarantees for the joint optimization of the network parameters and the correlation matrix.

**Limitations:**

yes

**Strengths And Weaknesses:**

Strengths:
1. The paper identifies a fundamental limitation in existing variational IMVC methods—the unrealistic independence assumption that leads to underestimated posterior variance. The proposed correlation-aware aggregation directly addresses this issue.
2.The authors establish their conclusions through rigorous derivations, backed by solid theoretical guarantees.
Weakness:
1.The paper does not actually model correlated data generation; instead, it only models correlated estimation errors in the inference network. Consequently, the title ("Beyond Independence") and abstract may mislead readers into thinking the independence assumption in the generative model has been relaxed.
2. The proposed ACOVA framework offers only marginal improvements when compared to existing methods.
Relative to Mancisidor et al. (2025), the main distinction is the application to incomplete multi-view data, which is a domain shift rather than algorithmic innovation.
Relative to DVMIC (Xu et al., 2024), the change is essentially replacing PoE with a correlated aggregation — again, an incremental modification.
3.The empirical convergence curves are provided, but theoretical convergence guarantees for the joint optimization of network parameters and correlation matrix are not discussed.

---

> ### Author Rebuttal · Authors · 2026-03-31
>
> Thank you for your constructive comments and thorough review. Below, we address your mentioned weaknesses (W) and questions (Q).
>
> ## (W1 and  Q1) Independence relaxed in inference, not generation
>
> Thank you for pointing this out. Our work does indeed not model correlated data generation, and the generative model remains conditionally independent across views, which is standard in the VAE framework. Given the focus on IMVC, we target the independence assumption in the variational posterior aggregation, as it is the primary mechanism that handles missingness and as well-calibrated aggregation is essential to facilitate robust clustering. We will make this more explicit in the revision of the abstract. Note, introducing correlations in the generative model is complementary to our proposed approach. However, it requires learning the covariance matrix of each view, which requires 2,767,128 learnable parameters for small color images of size 28x28. Therefore, this approach is less common in IMVC.
>
> ## (W2 and  Q2) Novelty and improvements
>
> Thank you for the comment. While our experiments include incomplete multi-view data, the key distinction from Mancisidor et al. (2025) lies elsewhere. In Mancisidor et al., the correlation parameters are selected via cross-validation, which has two practical limitations: 1) It requires labels, restricting the method to supervised settings and making it inapplicable when labeled data are unavailable. 2) It is limited to a very simple correlation structure to make cross-validation feasible.
>
> Our contribution is to learn the correlation matrix directly within the training objective in a principled manner, removing the reliance on labels and enabling richer correlation structures. To isolate the effect of correlation modeling, we take DVMIC and replace only its independence-assuming PoE aggregator with our correlation-aware aggregation. This single change yields consistent and significant improvements across datasets (ACC improvements of 4.5\% on average across datasets/missing rates), indicating that explicitly modeling inter-view correlations is the driver of the gains. Notably, the added complexity is minimal: 3 additional learnable parameters for 2-view datasets and 21 for 6-view datasets, relative to a DVMIC model with more than 6 (16) million parameters for 2 (6) views, with minimal runtime overhead (Appendix C.4). We will make this distinction even more explicit.
>
>
> ## (W3 and  Q3) Theoretical convergence guarantees
> Thank you for raising this point. We agree that theoretical insights and guarantees are beneficial and have therefore included theoretical results that show the identifiability of our covariance parameterization (A.3) as well as have chosen to define R through the Cholesky parameterization to ensure positive definiteness. However, given that convergence guarantees for standard variational autoencoders are just emerging (see for instance [1]), we consider their extension to multi-modal VAEs with our proposed correlation mechanism to be out of scope for this work and have instead focused on an extensive empirical analysis. We do however, agree that this would be an interesting follow-up work and will discuss it in the future work section.
>
> [1] Sobihan Surendran, Antoine Godichon-Baggioni, Sylvain Le Corff. Theoretical Convergence Guarantees for Variational Autoencoders. AISTATS 2025.

---

### Official Review · Reviewer_7ZPA · 2026-03-13

**Soundness:** 2
**Presentation:** 3
**Significance:** 3
**Originality:** 2
**Overall Recommendation:** 4
**Confidence:** 3

**Summary:**

The paper studies the problem of incomplete multi-view clustering, where data instances are described by multiple views, but some views may be missing. Existing variational approaches typically aggregate view-specific posterior distributions under a conditional independence assumption, commonly using product-of-experts (PoE) aggregation. The authors argue that this assumption is often unrealistic because different views may produce correlated estimation errors. To address this limitation, the paper proposes a variational framework that explicitly models correlations between view-specific posterior estimates. The method introduces a learnable correlation matrix that captures dependencies between the estimation errors of different views, parameterized through a normalized Cholesky decomposition to ensure positive definiteness. The resulting correlated aggregation mechanism generalizes the PoE formulation and allows the model to adaptively account for inter-view dependencies. Experiments on several benchmark datasets demonstrate consistent improvements over existing incomplete multi-view clustering methods under different missing-view settings.

**Compliance With Llm Reviewing Policy:**

Affirmed.

**Final Justification:**

The paper presents a way to incorporate dependence among views. While the methodological novelty is somewhat limited, I believe that the incorporation of dependence among views is done in a principled way and is shown to result in empirical gains. I tend to agree with the authors that this is a simple but fundamental change.

**Key Questions For Authors:**

1. In Sec.4.1 (p.5) the mask is defined as $M \in {0,1}$, while in the Appendix it appears as $M \in {0,1}^V$. However, in Eqs.(10)--(11) the operations $\Sigma^{-1} \odot MM^\top$ and $\mu \odot M$ require dimensions compatible with $\mu \in \mathbb{R}^{VD\times 1}$ and $\Sigma \in \mathbb{R}^{VD\times VD}$, which would seem to require a mask defined at the level of view-latent-dimension pairs (e.g., $M\in {0,1}^{VD\times 1}$). Relatedly, the dataset contains $N$ instances, but it is not fully clear from the notation how the sample dimension $N$ enters the formulation. I suspect this may mainly be a notational/readability issue, but clarifying the exact dimensionality of $M$ and whether the method is presented per sample would improve the readability and consistency of the derivations.

2. The proposed method learns a correlation matrix $R$ between view-specific estimation errors. Could the authors provide additional insight into how the learned correlations should be interpreted in practice? For example, do the learned correlations correspond to semantic similarity between views, redundancy in feature representations, or shared noise patterns? Some qualitative discussion or analysis across datasets would help clarify what structure the model is actually capturing.

3. Since the method learns a full $V\times V$ correlation matrix between views (for each latent dimension), how does the approach scale as the number of views increases? Have the authors evaluated the method in scenarios with a larger number of views, and does the estimation of the correlation structure remain stable in such settings?

4. The model learns a full correlation matrix $R$ between view-specific estimation errors. Are there mechanisms to regularize or constrain this matrix during training to avoid overfitting, particularly when the number of views or latent dimensions is large? It would be interesting to understand whether simpler parameterizations (e.g., low-rank or shared correlations across dimensions) were considered.

5. The method models inter-view dependencies through correlations in the estimation errors of the view-specific posteriors. Could the authors elaborate on why this formulation is preferable to modeling correlations directly in the latent representations or encoder outputs?

**Limitations:**

yes

**Strengths And Weaknesses:**

The proposed method extends DVIMC by introducing a learnable correlation structure in the posterior aggregation step. While the modification appears conceptually incremental, the empirical results consistently improve over DVIMC across datasets and missing-view settings.

The motivation for modeling correlated estimation errors is reasonable. The method is technically straightforward and the training objective remains largely aligned with the standard VAE-based formulation. The experimental evaluation appears sound and shows consistent improvements over the DVIMC baseline and other methods across several datasets and missing-view settings. One minor concern is that some aspects of the notation (e.g., the dimensionality of the mask $M$ and how the sample dimension $N$ enters the formulation) are not entirely clear and would benefit from clarification.

Overall the paper is reasonably well organized and the main ideas are understandable. The motivation for modeling correlations between views is clearly explained and the empirical results are easy to follow.

The proposed method can be seen as an extension of DVIMC. While this idea is conceptually reasonable and leads to consistent empirical improvements, the methodological change itself is relatively incremental since the overall architecture, objective, and training procedure remain largely unchanged from prior work. Nevertheless, the empirical results suggest that modeling cross-view correlations can provide measurable gains in incomplete multi-view clustering scenarios, which indicates that the proposed modification may have practical value.

---

> ### Author Rebuttal · Authors · 2026-03-31
>
> Thank you for the comments.
> ## (Q1) Notation
> We apologize for the notation issues and will fix them. Thank you for highlighting them.
> (1) For a single sample, $M$ is defined as $M\in${0,1}$^V$ (a column vector indicating view availability). For $N$ samples, the mask is $\mathcal{M}\in${0,1}$^{N\times V}$.
> (2) For $\mu\in\mathbb{R}^{VD\times1}$ and $\Sigma\in\mathbb{R}^{VD\times VD}$, we adapt the per-sample $M\in ${0,1}$^V$ by repeating the $V$-dimensional indicator $D$ times. This results in an adapted mask $M\in${0,1}$^{VD\times1}$
> (3) Regarding the sample dimension $N$: for simplicity most symbols (e.g., $\mu$, $\Sigma$, $M$) are defined for individual samples in the derivations. We will clarify this and explicitly include the average over all $N$ samples in the ELBO and alignment losses to avoid ambiguity.
> Note that the correlation $R$ is shared across all samples.
>
> ## (Q2) R Interpretation
> Thank you for the question. Each view-specific variational distribution provides an estimate of the joint posterior and provide a measure of uncertainty in their estimate. If the latent variable z is univariate and there are two view-specific variational distributions estimating its value, the correlation provides a measure of the agreement/disagreement between the two view-specific distributions' estimates. In our model, the latent variable learns shared information from the views as it does not impose any type of factorization on the latent variable z. Thus, if two view-specific assessments are positively correlated, this reflects semantic similarity between views. Negative correlation implies a discrepancy between the views. Noisy shared patterns will be reflected in the covariance matrix ${\Sigma}^d$ via high standard deviations in the view-specific assessments.
>
> Qualitative results (Fig. 4d) show weak correlations between morphological and intensity/frequency views, while profile features and pixel averages are strongly correlated, reflecting redundancy in intensity statistics. Similarly, on Caltech5V (C.7), the strongest correlations arise between HOG and LBP (both capturing local texture/edges). The controlled noise study (Sec. 5.4) further shows that increasing noise or missingness reduces off-diagonal correlations, indicating that R adapts to heterogeneity.
>
> R calibrates aggregation by down-weighting dependent experts and leveraging complementary ones, improving variance calibration and clustering performance.
>
> ## (Q3, Q4) Scalability of correlation mechanism
> Thank you for the questions. We first clarify a potential misunderstanding: although our method can learn a full (VxV) correlation matrix per latent dimension (C.6), all experiments use a *single shared* (VxV) matrix R. Combined with the Cholesky parameterization, this keeps the overhead negligible. For a 6-view dataset (larger #views than in most practical settings), only 21 parameters are added to the more than 16 million parameters of DVIMC. This design also yields strong stability, as supported by convergence and across-seed analyses (C.5, C.7).
>
> We also explore more expressive variants in C.6 (e.g., per-dimension or group-wise correlation matrices). However, in the unsupervised clustering setting, increased flexibility does not consistently improve performance and can lead to degenerate solutions. The shared R provides a strong balance of accuracy, stability, and complexity. Further regularization arises from the positive-definite constraint via Cholesky and minibatch estimation of R. Additional options, such as shrinkage toward identity (e.g., $\lambda |R - I|_F$), could also be incorporated.
>
> ## (Q5) Benefit of estimation errors
> This choice stems from a fundamental consideration in variational inference: the mean‑field assumption. VAEs rely on this to keep the ELBO tractable. Directly modeling correlations between encoder outputs would break it, making the subsequent derivation more complex and potentially losing a closed‑form solution in the KL term.
> Instead, we treat the output of view-specific distributions as estimates of the true joint posterior distribution, modeling their correlation via estimation errors. This preserves the mean‑field structure while capturing cross‑view dependencies. We will add this in the introduction.
>
> ## Novelty
> Thank you for noting the consistent gains. The novelty lies in replacing independence-assuming aggregation with a correlation-aware alternative, a simple but fundamental change that serves as a drop-in replacement, attributing gains to modeling dependence rather than architectural tweaks. The method is principled: it relaxes independence, provides a mask-aware closed-form aggregation generalizing PoE/CoDE, and ensures positive definiteness via normalized Cholesky parameterization. Despite its simplicity, it yields consistent gains across incomplete (Tables 1/2) and complete settings (App. C), with +4.5\% ACC on average across datasets/missingess over DVIMC (Table 1) and negligible parameter/runtime overhead.

---

> > ### Author Rebuttal · Reviewer_7ZPA · 2026-04-02
> >
> > Thank you for your rebuttal.
> >
> > The authors have addressed my main concerns, and I am happy to raise my score.
> > The paper presents a way to incorporate dependence among views. While the methodological novelty is somewhat limited, I believe that the incorporation of dependence among views is done in a principled way and is shown to result in empirical gains. I tend to agree with the authors that this is a simple but fundamental change.

---

> > > ### Author Response · Authors · 2026-04-05
> > >
> > > Thank you for taking our rebuttal into account and for raising your score. We are glad to see that all your concerns have been addressed.

---

### Decision · Program_Chairs · 2026-04-30

**Decision:**

Accept (regular)

**Comment:**

The authors present a method for multi-view clustering with incomplete data by introducing a correlation structure between the views. The proposed method generalizes some previous methods in multi-view clustering. However,  there are some raised issues with the novelty of the method as it is a somewhat incremental change from previous work. The experimental section of the paper demonstrates the utility of the method and there are some modest improvements compared to previous methods. However, the overall results are not strong enough to justify the relative lack of novelty. Thus, I recommend the authors either try to further strengthen their arguments with more convincing empirical or theoretical results.